# Selective Sampling and Imitation Learning via Online Regression

**Ayush Sekhari** [* 1]   **Karthik Sridharan** [2]   **Wen Sun** [2]   **Runzhe Wu** [2]

## Abstract

We consider the problem of Imitation Learning (IL) by actively querying noisy expert for feedback. While imitation learning has been empirically successful, much of prior work assumes access to noiseless expert feedback which is not practical in many applications. In fact, when one only has access to noisy expert feedback, algorithms that rely on purely offline data (non-interactive IL) can be shown to need a prohibitively large number of samples to be successful. In contrast, in this work, we provide an interactive algorithm for IL that uses selective sampling to actively query the noisy expert for feedback. Our contributions are twofold: First, we provide a new selective sampling algorithm that works with general function classes and multiple actions, and obtains the best-known bounds for the regret and the number of queries. Next, we extend this analysis to the problem of IL with noisy expert feedback and provide a new IL algorithm that makes limited queries.

Our algorithm for selective sampling leverages function approximation, and relies on an online regression oracle w.r.t. the given model class to predict actions, and to decide whether to query the expert for its label. On the theoretical side, the regret bound of our algorithm is upper bounded by the regret of the online regression oracle, while the query complexity additionally depends on the eluder dimension of the model class. We complement this with a lower bound that demonstrates that our results are tight. We extend our selective sampling algorithm for IL with general function approximation and provide bounds on both the regret and the number of queries made to the noisy expert. A key novelty here is that our regret and query complexity bounds only depend on the

number of times the optimal policy (and not the noisy expert, or the learner) go to states that have a small margin.

## 1. Introduction

From the classic supervised learning setting to the more complex problems like interactive Imitation Learning (IL) (Ross et al., 2011), high-quality labels or supervision is often expensive and hard to obtain. Thus, one wishes to develop learning algorithms that do not require a label for every data sample presented during the learning process. Active learning or selective sampling is a learning paradigm that is designed to reduce query complexity by only querying for labels at selected data points, and has been extensively studied in both theory and in practice (Agarwal, 2013; Dekel et al., 2012; Hanneke & Yang, 2021; Zhu & Nowak, 2022; Cesa-Bianchi et al., 2005; Hanneke & Yang, 2015).

In this work, we study selective sampling and its application to interactive Imitation Learning (Ross et al., 2011). Our goal is to design algorithms that can leverage general function approximation and online regression oracles to achieve small regret on predicting the correct labels, and at the same time minimize the number of expert queries made (query complexity). Towards this goal, we first study selective sampling which is an online active learning framework, and provide regret and query complexity bounds for general function classes (used to model the experts). Our key results in selective sampling are obtained by developing a connection between the regret of the online regression oracles and the regret of predicting the correct labels. Additionally, we bound the query complexity using the eluder dimension (Russo & Van Roy, 2013) of the underlying function class used to model the expert. We complement our results with a lower bound indicating that a dependence on an eluder dimension like complexity measure is unavoidable in the query complexity in the worst case. In particular, we provide lower bounds in terms of the star number of the function class—a quantity closely related to the eluder dimension. Our new selective sampling algorithm, called SAGE, can operate under fairly general modeling assumptions, loss functions, and allows for multiple labels (i.e., multi-class classification).

---

[*] Authors are listed in alphabetical order of their last names [1]MIT [2]Cornell University. Correspondence to: Ayush Sekhari <sekhari@mit.edu>.

Appearing at *Interactive Learning with Implicit Human Feedback Workshop at ICML 2023*, Honolulu, Hawaii, USA. Copyright 2023 by the author(s).

We then extend our selective sampling algorithm to the interactive IL framework proposed by (Ross et al., 2011) to reduce the query complexity. While the DAgger algorithm proposed by (Ross et al., 2011) has been extensively used in various robotics applications (e.g., Ross et al. (2013); Pan et al. (2018)), it often requires a large number of expert queries. There have been some efforts on reducing the expert query complexity by leveraging ideas from active learning (e.g., (Laskey et al., 2016; Brantley et al., 2020)), however, these prior attempts do not have theoretical guarantees on bounding expert's query complexity. In this work, we provide the first provably correct algorithm for interactive IL with general function classes, called , which not only achieves strong regret bounds in terms of maximizing the underlying reward functions, but also enjoys a small query complexity. Furthermore, we note that operates under significantly weaker assumptions as compared to the prior works, like (Ross et al., 2011), on interactive IL. In particular, we only assume access to a noisy expert, as compared to the prior works that assume that the expert is noiseless. In fact, for the noisy setting, we show that one can not even hope to learn from purely offline expert demonstrations unless one has exponentially in horizon $H$ many samples. Such a strong separation does not hold in the noiseless setting.

Our bounds depend on the margin of the noisy expert, which intuitively quantifies the confidence level of the expert. In particular, the margin is large for states where the expert is very confident in terms of providing the correct labels, while on the other hand, the margin is small on the states where the expert is less confident and subsequently provides more noisy labels as feedback. Such kind of margin condition was missing in prior works, like (Ross et al., 2011), which assumes that the expert can provide confident labels everywhere. Additionally, we note that our margin assumption is quite mild as we only assume that the expert has a large margin under the states that could be visited by the noiseless expert (however, the states visited by the learner, or by following the noisy expert, may not have a small margin).

We then extend our results to the multiple expert setting where the learner has access to $M$ many experts/teachers who may have different expertise at different parts of the state space. In particular, there is no expert who can singlehandedly perform well on the underlying environment, but an aggregation of their policies can lead to good performance. Such an assumption holds in various applications and has been recently explored in continuous control tasks like robotics and discrete tasks like chess and Minigrid (Beliaev et al., 2022). For illustration, consider the game of chess, where we can easily find experts that have non-overlapping skills, e.g. some experts may have expertise on how to open the game, and other experts may have expertise in endgames. In this case, while no single expert can perform

well throughout the game, an aggregation of their policies can lead to a good strategy that we wish to compete with.

Similar to the single expert setting, we model the expertise of the experts in multiple expert setting using the concept of margins. Different experts have different margin functions, capturing the fact that experts may have different expertise at different parts of the state space. Prior work from (Cheng et al., 2020) also considers multiple experts in IL and provides meaningful regret bounds, however, their assumption on the experts is much stronger than us: they assume that for any state, there at least exists one expert who can achieve high reward-to-go if the expert took over the control starting from this state till the end of the episode. Furthermore, (Cheng et al., 2020) considers the setting where one can also query for the reward signals, whereas we do not require access to any reward signals.

## 2. Contributions and Overview of Results

**Selective Sampling**    Online selective sampling models the interaction between a learner and an adversary over $T$ rounds. At the beginning of each round of the interaction, the adversary presents a context $x_t$ to the learner. After receiving the context, the learner makes a prediction $\hat{y}_t \in [K]$, where $K$ denotes the number of actions. Then, the learner needs to make a choice of whether or not to query an *expert* who is assumed to have some knowledge about the true label for all the presented contexts. The experts knowledge about the true label is modeled via the ground truth modeling function $f^\star$, which is assumed to belong to a given function class $\mathcal{F}$ but is unknown to the learner. If the learner decides to query for the label, then the expert will return a noisy label $y_t$ sampled using $f^\star$. If the learner does not query, then the learner does not receive any feedback in this round. The learner makes an update based on the latest information it has, and moves on to the next round of the interaction. The goal of the learner is to compete with the expert policy $\pi^\star$, that is defined using the experts model $f^\star$. In the selective sampling setting, we care about two things: the total regret of the learner w.r.t. the policy $\pi^\star$, and the number of expert queries that the learner makes. Our key contributions are as follows:

- We provide a new selective sampling algorithm (Algorithm 1) that relies on an online regression oracle w.r.t. $\mathcal{F}$ (where $\mathcal{F}$ is the given model class) to make predictions and to decide whether to query for labels. Our algorithm can handle multiple actions, adversarial contexts, arbitrary model class $\mathcal{F}$, and fairly general modeling assumptions (that we discuss in more detail in Section 3), and enjoys the following regret bound and query complexity:

$$\text{Reg}_T = \widetilde{O}\left(\inf_\varepsilon \left\{\varepsilon T_\varepsilon + \frac{\text{Reg}(\mathcal{F}; T)}{\varepsilon}\right\}\right) \quad \text{and,}$$

$$N_T = \widetilde{O}\left(\inf_{\varepsilon}\left\{T_\varepsilon + \frac{\mathrm{Reg}(\mathcal{F};T) \cdot \mathfrak{E}(\mathcal{F},\varepsilon;f^\star)}{\varepsilon^2}\right\}\right).$$
(1)

where $\mathrm{Reg}(\mathcal{F};T)$ denotes the regret bound for the online regression oracle on $\mathcal{F}$, $\mathfrak{E}(\mathcal{F},\varepsilon;f^\star)$ denotes the eluder dimension of $\mathcal{F}$, and $T_\varepsilon$ denotes the number of rounds at which the margin of the experts predictions is smaller than $\varepsilon$ (the exact notion of margin is defined in Section 3).

- We show via a lower bound that, without additional assumptions, the dependence on the eluder dimension in the query complexity bound (1) is unavoidable if we desire a regret bound of the form (1), even when $T_\varepsilon = 0$. The details are located in Section 3.2.

- For the stochastic setting, where the context $\{x_t\}_{t \le T}$ are sampled i.i.d. from a fixed unknown distribution, we provide an alternate algorithm (Algorithm 3) that enjoys the same regret bound as (1) but whose query complexity scales with the disagreement coefficient of $\mathcal{F}$ instead of the eluder dimension (Theorem 2). Since the disagreement coefficient is always smaller than the eluder dimension, Theorem 2 yields an improvement in the query complexity.

**Imitation Learning.** We then move to the more challenging Imitation Learning (IL) setting, where the learner operates in an episodic finite horizon Markov Decision Process (MDP), and can query a noisy expert for feedback (i.e. the expert action) on the states that it visits. The interaction proceeds in $T$ episodes of length $H$ each. In episode $t$, at each time step $h \in [H]$ and on the state $x_{t,h}$, the learner chooses an action $\widehat{y}_{t,h}$ and transitions to state $x_{t,h+1}$. However, the learner does not receive any reward signal. Instead, the learner can actively choose to query an *expert* who has some knowledge about the correct action to be taken on $x_{t,h}$, and gives back noisy feedback $y_{t,h}$ about this action. Similar to the selective sampling setting, the experts knowledge about the true label is modeled via the ground truth modeling function $f_h^\star$, which is assumed to belong to a given function class $\mathcal{F}_h$ but is unknown to the learner. The goal of the learner is to compete with the optimal policy $\pi^\star$ of the (noiseless) expert. Our key contributions in IL are:

- In Section 4, we first demonstrate an exponential separation in terms of task horizon $H$ in the sample complexity, for learning via offline expert demonstration only vs interactive querying of experts, when the feedback from the expert is noisy.

- We then provide a general IL algorithm (in Algorithm 2) that relies on online regression oracles w.r.t. $\{\mathcal{F}_h\}_{h \le H}$ to predict actions, and to decide whether to query for labels. Similar to the selective sampling setting, the regret

bound for our algorithm scales with the regret of the online regression oracles, and the query complexity bound has an additional dependence on the eluder dimension. Furthermore, our algorithm can handle multiple actions, adversarially changing dynamics, arbitrary model class $\mathcal{F}$, and fairly general modeling assumptions.

- A key difference from our results in selective sampling is that the term $T_\varepsilon$ that appears in our regret and query complexity bounds in IL denote the number of time steps in which the expert policy $\pi^\star$ has a small margin (instead of the number of time steps when the learner's policy has a small margin). In fact, the learner and the expert trajectories could be completely different from each other, and we only pay in the margin term if the expert trajectory at that time step would have a low margin. See Section 4 for the exact definition of margin.

- In Section 4.1, we provide extensions to our algorithm when the learner can query $M$ experts at each round. Similar to selective sampling setting, we do not assume that any of the experts is singlehandedly optimal for the entire state space, but that there exist aggregation functions of these experts' predictions that perform well in practice, and with which we compete.

## 3. Selective Sampling

In the problem of selective sampling, on every round $t$, nature produces a context $x_t$ (possibly chosen adversarially). The learner then receives this context and predicts a label $\widehat{y}_t \in [K]$ for that context. The learner also computes a query condition $Z_t \in \{0,1\}$ for that context. If $Z_t = 1$, the learner requests for label $y_t \in [K]$ corresponding to the $x_t$, and if not, the learner receives no feedback on the label for that round. Let $\mathcal{F}$ be a model class such that each model $f \in \mathcal{F}$ maps contexts $x$ to scores $f(x) \in \mathbb{R}^K$. In this work we assume that while contexts can be chosen arbitrarily, the label $y_t$ corresponding to a context $x_t$ is drawn from a distribution over labels specified by the score $f^\star(x_t)$ where $f^\star \in \mathcal{F}$ is a fixed model unknown to the learner. We assume that a link function $\phi : \mathbb{R}^K \mapsto \Delta(K)$ maps scores to distributions and assume that the noisy label

$$y_t \sim \phi(f^\star(x_t)).$$
(2)

In this work, we assume that the link function $\phi(v) = \nabla\Phi(v)$ for some $\Phi : \mathbb{R}^K \mapsto \mathbb{R}$ (see (Agarwal, 2013) for more details) which satisfies the following assumption:

**Assumption 1.** The function $\Phi$ is $\lambda$-strongly-convex and $\gamma$-smooth, i.e. for all $u, u' \in \mathbb{R}^K$,

$$\frac{\lambda}{2}\|u' - u\|_2^2 \le \Phi(u') - \Phi(u) - \langle \nabla\Phi(u), u' - u \rangle \le \frac{\gamma}{2}\|u' - u\|_2^2.$$

Our main contribution in this section is a selective sampling algorithm that uses online non-parametric regression

w.r.t. the model class $\mathcal{F}$ as a black box. Specifically, define the loss function corresponding to the link function $\phi$ as $\ell_\phi(v, y) = \Phi(v) - v[y]$ where $v \in \mathbb{R}^K$ and $y \in [K]$. We assume that the learner has access to an online regression oracle for the loss $\ell_\phi$ (which is a convex loss) w.r.t. the class $\mathcal{F}$, that for any sequence $\{(x_1, y_1), \ldots, (x_T, y_T)\}$ guarantees the regret bound

$$\sum_{s=1}^T \ell_\phi(f_s(x_s), y_s) - \inf_{f \in \mathcal{F}} \sum_{s=1}^T \ell_\phi(f(x_s), y_s) \le \mathrm{Reg}^{\ell_\phi}(\mathcal{F}; T).$$
(3)

When $\phi$ is identity (under which the models in $\mathcal{F}$ directly map to distributions over the labels), then $\ell_\phi$ denotes the standard square loss, and we need a bound on $\mathrm{Reg}^{\mathrm{sq}}(\mathcal{F}; T)$. When $\phi$ is the Boltzman distribution mapping (given by $\Phi$ being the softmax function) then $\ell_\phi$ is the logistic loss, and we need an online logistic regression oracle for $\mathcal{F}$. Minimax rates for the regret bound in (3) are well known:

- *Square loss regression:* Rakhlin & Sridharan (2014) characterized the minimax rates for online square loss regression in terms of the offset sequential Rademacher complexity of $\mathcal{F}$, which for example, leads to regret bound $\mathrm{Reg}^{\mathrm{sq}}(\mathcal{F}; T) = O(\log|\mathcal{F}|)$ for finite function classes $\mathcal{F}$, and $\mathrm{Reg}^{\mathrm{sq}}(\mathcal{F}; T) = O(d \log(T))$ when $\mathcal{F}$ is a $d$-dimensional linear class. More examples can be found in Rakhlin & Sridharan (2014, Section 4). We refer the readers to Krishnamurthy et al. (2017); Foster et al. (2018a) for efficient implementations.

- *Logistic loss regression:* When $\mathcal{F}$ is finite, we have the regret bound $\mathrm{Reg}(\mathcal{F}; T) \le O(\log|\mathcal{F}|)$ (Cesa-Bianchi & Lugosi, 2006, Chapter 9). For learning linear predictors, there exists efficient improper learner with regret bound $\mathrm{Reg}(\mathcal{F}; T) \le O(d \log|T|)$ (Foster et al., 2018b). More examples can be found in Foster et al. (2018b, Section 7) and (Rakhlin & Sridharan, 2015).

When one deals with complex model classes $\mathcal{F}$ such that the labeling concept class corresponding to $\mathcal{F}$ could possibly have infinite VC dimension (like it is typically the case), then one needs to naturally rely on a margin-based analysis (Tsybakov, 2004; Shalev-Shwartz & Ben-David, 2014; Dekel et al., 2012). For $p \in \mathbb{R}^K$, we use the following well-known notion of margin for multiclass settings:

$$\mathtt{Margin}(p) = \phi(p)[k^\star] - \max_{k' \neq k^\star} \phi(p)[k'],$$
(4)

where $k^\star \in \mathrm{argmax}_k \phi(p)[k]$. A key quantity that appears in our results is the number of $x_t$'s that fall within an $\varepsilon$ margin region,

$$T_\varepsilon = \sum_{t=1}^T \mathbf{1}\{\mathtt{Margin}(f^\star(x_t)) \le \varepsilon\}.$$

$T_\varepsilon$ denotes the number of times where even the Bayes optimal classifier is confused about the correct label on $x_t$, and has confidence less than $\varepsilon$. The algorithm relies on an online regression oracle mentioned above to produce the predictor $f_t$ at every round. The predicted label $\widehat{y}_t = \mathtt{SelectAction}(f_t(x_t)) = \mathrm{argmax}_k \phi(f_t(x_t))[k]$ is picked based on the score $f_t(x_t)$ (where $\widehat{y}_t$ is the label with the largest score). The learner updates the regression oracle on only those rounds in which it makes a query. Our main algorithm for selective sampling is provided in Algorithm 1. Our goal in this work is twofold. Firstly, we would like

---

**Algorithm 1** Selective Sampling, Action-set $\mathcal{A} = [K]$

**Require:** Parameters $\delta, \gamma, \lambda, T$, function class $\mathcal{F}$, and online regression oracle Oracle w.r.t $\ell_\phi$.

1: Set
$\Psi_\delta^{\ell_\phi}(\mathcal{F}, T) = \frac{4}{\lambda} \mathrm{Reg}^{\ell_\phi}(\mathcal{F}; T) + \frac{112}{\lambda^2} \log(4 \log^2(T)/\delta)$

2: Compute $f_1 \leftarrow \mathtt{Oracle}_1(\varnothing)$.

3: **for** $t = 1$ to $T$ **do**

4:     Nature chooses $x_t$.

5:     Learner plays the action
$\widehat{y}_t = \mathtt{SelectAction}(f_t(x_t))$.

6:     Learner computes

$$\Delta_t(x_t) := \max_{f \in \mathcal{F}} \|f(x_t) - f_t(x_t)\|$$

$$\text{s.t.} \quad \sum_{s=1}^{t-1} Z_s \|f(x_s) - f_s(x_s)\|^2 \le \Psi_\delta^{\ell_\phi}(\mathcal{F}, T),$$
(5)

7:     Learner decides whether to query: $Z_t = \mathbf{1}\{\mathtt{Margin}(f_t(x_t)) \le 2\gamma \Delta_t(x_t)\}$.

8:     **if** $Z_t = 1$ **then**

9:         Learner queries the label $y_t$ on $x_t$.

10:         $f_{t+1} \leftarrow \mathtt{Oracle}_t(\{x_t, y_t\})$.

11:     **else**

12:         $f_{t+1} \leftarrow f_t$.

13:     **end if**

14: **end for**

---

Algorithm 2 to have a low regret w.r.t. the optimal model $f^\star$, defined as

$$\mathrm{Reg}_T = \sum_{t=1}^T \mathbf{1}\{\widehat{y}_t \neq y_t\} - \sum_{t=1}^T \mathbf{1}\{\mathtt{SelectAction}(f^\star(x_t)) \neq y_t\}$$

Simultaneously, we also aim to make as few label queries $N_T = \sum_{t=1}^T Z_t$ as possible. Before delving into our results, we first recall the following variant of eluder-dimension (Russo & Van Roy, 2013; Foster et al., 2020; Zhu & Nowak, 2022).

**Definition 1** (Scale-sensitive eluder dimension (normed version))**.** Fix any $f^\star \in \mathcal{F}$, and define $\widetilde{\mathfrak{E}}(\mathcal{F}, \beta; f^\star)$ to be the length of the longest sequence of contexts $x_1, x_2, \ldots x_m$

such that for all $i$, there exists $f_i \in \mathcal{F}$ such that

$$\|f_i(x_i) - f^\star(x_i)\| > \beta, \text{ and } \sum_{j<i}\|f_i(x_j) - f^\star(x_j)\|^2 \le \beta^2.$$

The value function eluder dimension is defined as $\mathfrak{E}(\mathcal{F}, \beta'; f^\star) = \sup_{\beta \ge \beta'} \widetilde{\mathfrak{E}}(\mathcal{F}, \beta; f^\star)$.

Bounds on the eluder dimension for various function classes are well known, e.g. when $\mathcal{F}$ is finite, $\mathfrak{E}(\mathcal{F}, \beta'; f^\star) \le |\mathcal{F}| - 1$, and when $\mathcal{F}$ is the set of $d$-dimensional function with bounded norm, then $\mathfrak{E}(\mathcal{F}, \beta'; f^\star) = O(d)$. We refer the reader to Russo & Van Roy (2013); Mou et al. (2020); Li et al. (2022) for more examples. The following theorem is our main result for selective sampling:

**Theorem 1.** Let $\delta \in (0, 1)$. Under the modeling assumptions above (in (2), (3) and (4)), with probability at least $1 - \delta$, Algorithm 1 obtains the regret bound

$$\text{Reg}_T = \widetilde{O}\left(\inf_\varepsilon\left\{\varepsilon T_\varepsilon + \frac{\gamma^2}{\lambda\varepsilon}\text{Reg}^{\ell_\phi}(\mathcal{F}; T) + \log(1/\delta)\right\}\right),$$

while simultaneously the total number of label queries made is bounded by:

$$N_T = \widetilde{O}\left(\inf_\varepsilon\left\{T_\varepsilon + \frac{\gamma^2}{\lambda\varepsilon^2} \cdot \text{Reg}^{\ell_\phi}(\mathcal{F}; T) \cdot \mathfrak{E}(\mathcal{F}, \varepsilon/4\gamma; f^\star) \right.\right.$$
$$\left.\left. + \log(1/\delta)\right\}\right).$$

A few points are in order:

- It must be noted that for most settings we consider, as an example if model class $\mathcal{F}$ is finite, one typically has that $\text{Reg}(\mathcal{F}; T) \le \log|\mathcal{F}|$. Thus, in the case where one has a hard margin condition i.e. $T_\gamma = 0$ for some $\gamma > 0$, we get $\text{Reg}_T \le O\left(\frac{\log|\mathcal{F}|}{\gamma}\right)$ and $N_T \le O\left(\frac{\mathfrak{E}(\mathcal{F}, \varepsilon; f^\star)\log|\mathcal{F}|}{\gamma^2}\right)$.

- Our regret bound does not depend on the eluder dimension. However, the query complexity bound has a dependence on eluder dimension. Thus, for function classes for which the eluder dimension is large, the regret bound is still optimal while the number of label queries may be large.

### 3.1. Stochastic Setting

So far we assumed that the contexts $\{x_t\}_{t\ge 0}$ could be chosen in a possible adversarial fashion, and thus our bound on the number of label queries scales with the eluder dimension. However, it turns out that if the contexts are drawn i.i.d. from some (unknown) distribution $\mu$, then one can improve the query complexity to scale with the value function disagreement coefficient of $\mathcal{F}$ (defined below) which is always smaller than the eluder dimension (Lemma 6).

**Definition 2** (Scale sensitive disagreement coefficient (normed version), (Foster et al., 2020)). Let $\mathcal{F} \subseteq \{\mathcal{X} \mapsto \mathbb{R}^K\}$. For any $f^\star \in \mathcal{F}$, and $\beta_0, \varepsilon_0 > 0$, the value function disagreement coefficient $\theta^{\text{val}}(\mathcal{F}, \varepsilon_0, \beta_0; f^\star)$ is defined as

$$\sup_\mu \sup_{\beta > \beta_0, \varepsilon > \varepsilon_0} \left\{\frac{\varepsilon^2}{\beta^2} \cdot \Pr_{x\sim\mu}\binom{\exists f\in\mathcal{F}\|f(x)-f^\star(x)\|>\varepsilon,}{\|f-f^\star\|_\mu \le \beta}\right\} \vee 1$$

where $\|f\|_\mu = \sqrt{\mathbb{E}_{x\sim\mu}[\|f(x)\|^2]}$.

The key idea that gives us the above improvement, of replacing the eluder dimension by disagreement coefficient in the query complexity bound, is to use epoching for the query condition, while still using an online regression oracle to make predictions. The exact algorithm is given in Algorithm 3, deferred to Appendix C.4.

**Theorem 2.** Let $\delta \in (0, 1)$, and consider the modeling assumptions in (2), (3) and (4). Furthermore, suppose that $x_t$ is sampled i.i.d. from $\mu$, where $\mu$ is a fixed distribution. Then, with probability at least $1 - \delta$, Algorithm 3 obtains the regret bound

$$\text{Reg}_T = \widetilde{O}\left(\inf_\varepsilon\left\{\varepsilon T_\varepsilon + \frac{\gamma^2}{\lambda\varepsilon}\text{Reg}^{\ell_\phi}(\mathcal{F}; T) + \log(1/\delta)\right\}\right),$$

while simultaneously the total number of label queries made is bounded by:

$$N_T = \widetilde{O}\left(\inf_\varepsilon\left\{T_\varepsilon + \log(1/\delta)\right.\right.$$
$$\left.\left. + \frac{\gamma^2}{\lambda\varepsilon^2} \cdot \text{Reg}^{\ell_\phi}(\mathcal{F}; T) \cdot \theta^{\text{val}}\left(\mathcal{F}, \varepsilon/8\gamma, \text{Reg}^{\ell_\phi}(\mathcal{F};T)/T; f^\star\right)\right\}\right).$$

We note that Algorithm 3 automatically adapts to Tsybakov noise condition with respect to $\mu$.

**Corollary 1** (Tsybakov noise condition, (Tsybakov, 2004)). Suppose there exists constants $c, \rho \ge 0$ s.t. $\Pr_{x\sim\mu}(\text{Margin}(f^\star(x)) \le \varepsilon) \le c\varepsilon^\rho$ for all $\varepsilon \in (0, 1)$, and consider the same modeling assumptions as in Theorem 2. Then, with probability at least $1 - \delta$, Algorithm 3 obtains the bound

$$\text{Reg}_T = \widetilde{O}\left(\left(\text{Reg}^{\ell_\phi}(\mathcal{F}; T)\right)^{\frac{\rho+1}{\rho+2}} \cdot (T)^{\frac{1}{\rho+2}}\right), \quad \text{and,}$$

$$N_T = \widetilde{O}\left(\left(\text{Reg}^{\ell_\phi}(\mathcal{F}; T) \cdot \theta^{\text{val}}\left(\mathcal{F}, \varepsilon/8\gamma, \text{Reg}^{\ell_\phi}(\mathcal{F};T)/T; f^\star\right)\right)^{\frac{\rho}{\rho+2}}\right.$$
$$\left. \cdot T^{\frac{2}{\rho+2}}\right).$$

A detailed comparison of our results with the relevant prior works is given in Appendix C.

## 3.2. Lower Bounds (Binary Action Case)

We supplement the above upper bound with a lower bound in terms of star number (defined below). The star number can be bounded from above by eluder dimension which appears in our lower bounds. While in general star number may not be lower bounded by eluder dimension, it is the case that for most commonly considered classes, star number seems to be the same order as eluder dimension ([Foster et al., 2020](#)). For the sake of a clean presentation, we restrict our lower bound to the binary actions case, although one can easily extend the lower bounds to the multiple actions case.

**Definition 3** (scale-sensitive star number). For any $\zeta \in (0,1)$ and $\beta \in (0, \zeta/2)$, define $\mathfrak{s}^{\mathrm{val}}(\mathcal{F}, \zeta, \beta)$ as the largest $m$ such that there exists target function $f^\star \in \mathcal{F}$ and sequence $x_1, \ldots, x_m \in \mathcal{X}$ s.t. $\forall i \in [m], |f^\star(x_i)| > \zeta, \exists f_i \in \mathcal{F}$ s.t.,

(1) $\sum_{j \neq i} (f_i(x_j) - f^\star(x_j))^2 < \beta^2$

(2) $|f_i(x_i)| > \zeta/2$ and $f_i(x_i) f^\star(x_i) < 0$

(3) $|f_i(x_i) - f^\star(x_i)| \leq 2\zeta$

The below theorem provides a lower bound on number of queries, in terms of star number for any algorithm that guarantees a non-trivial regret bound.

**Theorem 3.** Given a function class $\mathcal{F}$ and some desired margin $\zeta > 0$, define $\beta$ to be the largest $\beta$ s.t., $\beta^2 \leq \min\{\zeta^2/\mathfrak{s}^{\mathrm{val}}(\mathcal{F}, \zeta, \beta), \zeta^2/16\}$. Any algorithm that guarantees regret bound of $\mathbb{E}[\mathrm{Reg}_T] \leq C \frac{\zeta T}{\mathfrak{s}^{\mathrm{val}}(\mathcal{F}, \zeta, \beta)}$ on all instances with margin $\zeta/2$, there exists a distribution $\mu$ over $\mathcal{X}$ and a target function $f^\star \in \mathcal{F}$ with margin $\zeta$ such that, on that instance the algorithm has a lower bound of number of label queries of at least:

$$\mathbb{E}[N_T] \geq \frac{\log(2)\mathfrak{s}^{\mathrm{val}}(\mathcal{F}, \zeta, \beta)}{40\zeta^2}$$

The above lower bound demonstrates that a dependence on the a quantity like eluder dimension or star number in the number of queries required is real, thus showing that our upper bound cannot be improved (beyond the discrepancy between star number and eluder dimension).

**Corollary 2.** There exists a class $\mathcal{F}$ with $|\mathcal{F}| = \sqrt{T}$ such that any algorithm that makes less than $\sqrt{T}$ number of label queries, will have a regret of at least $\mathbb{E}[\mathrm{Reg}_T] \geq \sqrt{T}$.

## 4. Imitation Learning ($H > 1$) with Selective Queries to Expert

The problem of Imitation Learning (IL) consists of learning policies in MDPs when one has access to an expert or a teacher that can make suggestions on which actions to take at a given state. IL has enjoyed tremendous empirical success, and various different interaction models have been considered. In the simplest IL setting, studied under the umbrella of offline RL ([Levine et al., 2020](#)) or Behavior Cloning ([Ross & Bagnell, 2010](#); [Torabi et al., 2018](#)), the learner is given an offline dataset of trajectories (state and action pairs) from an expert and aims to output a well-performing policy. Here, the learner is not allowed any interaction with the expert, and can only rely on the provided dataset of expert demonstrations for learning. A much stronger IL setting is the one where the learner can interact with the expert, and rely on its feedback on states that it reaches by executing its own policies.

In their seminal work, [Ross et al. (2011)](#) proposed a framework for interactive imitation learning via reduction to online learning and classification tasks. This has been extensively studied in the IL literature (e.g., ([Ross & Bagnell, 2014](#); [Sun et al., 2017](#); [Cheng & Boots, 2018](#))). The algorithm DAgger from ([Ross et al., 2011](#)) has enjoyed great empirical success. On the theoretical side, however, performance guarantees for DAgger only hold under the assumption that, when queried, the expert makes action suggestions from a very good policy $\pi^\star$ that we would like to compete with. However, in practice, human demonstrators are far from being optimal and suggestions from experts should be modeled as noisy suggestions that only correlate with $\pi^\star$. It turns out that IL where one only has access to noisy expert suggestions is drastically different from the noiseless setting. For instance, in the sequel, we show that there can be an exponential separation in terms of the dependence on horizon $H$ in the sample complexity of learning purely from offline demonstration vs learning with online interactions.

Formally, we consider interactive IL in an episodic finite horizon Markov Decision Process (MDP), where the learner can query a noisy expert for feedback (i.e., action) on the states that it visits. The game proceeds in $T$ episodes. In each episode $t$, the nature picks the initial state $x_{t,1}$ for $h = 1$; then for every time step $h \in [H]$, the learner proposes an action $\hat{y}_{t,h} \in [K]$ given the current state $x_{t,h}$; then the system proceeds by sampling the next state $x_{t;h+1} \sim \widetilde{T}(x_{t,h}, \hat{y}_{t,h})$. The learner then decides whether to query the expert for feedback. If the learner queries, it receives a recommended action from the expert, and otherwise the learner does not receive any additional information. The game moves on to the next time step $h + 1$, and moves to the next episode $t + 1$ when it reaches to time step $H$ in the current episode. We now describe the expert model. With $f_h^\star$ being the underlying score function at time step $h$, the expert feedback is sampled from a distribution $\phi(f_h^\star(x)) \in \Delta(K)$, with $\phi : \mathbb{R}^K \mapsto \mathbb{R}^K$ being some link function (e.g., $\phi(p)[i] \propto \exp(p[i])$). The benchmark that we compare against the Bayes optimal policy given by $\pi_h^\star(x) := \mathrm{argmax}_{a \in [K]} \phi(f_h^\star(x))$. Our goal is to learn a

policy $\pi$ that is comparable to $\pi^\star$ in terms of optimizing some (unknown) reward function, under possibly adversarial transition dynamics, while at the same time, we want to minimize the number of queries to the expert. Regret for the IL problem is defined as

$$\text{Reg}_T = \sum_{t=1}^{T} \sum_{h=1}^{H} r(x_{t,h}^{\pi^\star}, \pi^\star_{\;h}(x_{t,h}^{\pi^\star})) - \sum_{t=1}^{T} \sum_{h=1}^{T} r(x_{t,h}, \hat{y}_{t,h})$$

where $x_{t,h}^{\pi^\star}$ denotes the states that would have been generated if we executed $\pi^\star$ from the beginning of the episode, i.e., we consider a counterfactual regret. The query complexity $N_T$ is the total number of queries to expert across all H steps in $T$ episodes.

Given the selective sampling results we provided in the earlier section, one may be tempted to apply them to the imitation learning problem. However, there is a caveat. A key to the reduction in Ross et al. (2011) is to apply Performance Difference Lemma (PDL) to reduce the problem of IL to online classification under the sequence of state distributions induced by the policies played by the learning algorithm. Hence, if one blindly applied this reduction, then in the margin term, one would need to account for the states that the learner visits (which could be arbitrary). Thus, for DAgger to have meaningful bounds, we would require a large margin over the entire state space. This is too much to ask for in practical applications. Consider the example of learning autonomous driving from a human driver as the expert. It is reasonable to believe that human drivers can confidently provide the right actions when they are driving themselves or are faced with situations they are more familiar with. However, assuming that the human driver is going to be confident in an unfamiliar situation (e.g., an emergency situation that is not often encountered by the human driver), is a strong assumption. Towards that end, we make a significantly weaker, and much more realistic, margin assumption that the expert has a large margin only on the state distribution induced by $\pi^\star$, and not on the state distribution of the learner or the noisy expert. In particular, we define $T_{\varepsilon,h}$ to denote the total number of episodes where the comparator policy $\pi^\star$ visits a state with low margin at time step $h$, i.e., $T_{\varepsilon,h} = \sum_{t=1}^{T} \mathbf{1}\{\text{Margin}(f_h^\star(x_{t,h}^{\pi^\star})) \le \varepsilon\}$.

We now proceed to our main results in this section. Learning from a noisy expert is indeed very challenging. In fact, learning from noisy expert feedback may even be statistically intractable in the non-interactive IL setting, where the learner is only limited to accessing offline noisy expert demonstrations for learning, e.g. in offline RL, Behavior Cloning, etc. The following lower bound formalizes this. In fact, the same lower bound also shows that AggreVaTe (Ross & Bagnell, 2014) style algorithms would not succeed under noisy expert feedback, AggreVaTe relies on roll-outs obtained by running the (noisy) expert suggestions.

**Proposition 1** (Lower bound for learning from non-interactive noisy demonstrations)**.** There exists an MDP, for every $h \le H$, a function class $\mathcal{F}_h$ with $|\mathcal{F}_h| \le 2^H$, a noisy expert whose optimal policy $\pi^\star(x) = \text{argmax}_a(f_h^\star(x)[a])$ for some $f_h^\star \in \mathcal{F}_h$ with $T_{\varepsilon,h} = 0$ for any $\varepsilon \le 1/4$, such than any non-interactive algorithm needs $\Omega(2^H)$ many noisy expert trajectory demonstrations to learn, with probability at least $3/4$, a policy $\hat{\pi}$ that is $1/8$-suboptimal w.r.t. $\pi^\star$.

The above Proposition 1 suggests that in order to learn with a reasonable sample complexity (that is polynomial in $H$), a learner must be able to interactively query the expert. In Algorithm 2, we provide an interactive imitation learning algorithm (with selective querying) that can learn from noisy expert feedback. A key to obtaining our result is a modified version of PDL, that we provide in Lemma 18 in the appendix, that allows us to only have the margin under the state distribution of $\pi^\star$. Our result extends to the setting where transitions are picked adversarially, i.e., at time step $h$ and episode $t$, after seeing $\hat{y}_{t,h}$ proposed by the learner, the nature can select $T_{t,h}$ which deterministically generates $x_{t,h+1}$ given $x_{t,h}, \hat{y}_{t,h}$. The regret bound and query complexity bounds for Algorithm 2 are:

**Theorem 4.** Let $\delta \in (0,1)$. Under the modeling assumptions above, with probability at least $1 - \delta$, Algorithm 2 obtains:

$$\text{Reg}_T = \widetilde{O}\left( \inf_\varepsilon \left\{ H \sum_{h=1}^{H} T_{\varepsilon,h} \right.\right.$$
$$\left.\left. + \frac{H\gamma^2}{\lambda\varepsilon^2} \sum_{h=1}^{H} \text{Reg}^{\ell_\phi}(\mathcal{F}_h; T) + \log(1/\delta) \right\} \right),$$

and,

$$N_T = \widetilde{O}\left( \inf_\varepsilon \left\{ H \sum_{h=1}^{H} T_{\varepsilon,h} + \log(1/\delta) \right.\right.$$
$$\left.\left. + \frac{H\gamma^2}{\lambda\varepsilon^2} \sum_{h=1}^{H} \text{Reg}^{\ell_\phi}(\mathcal{F}_h; T) \cdot \mathfrak{E}(\mathcal{F}_h, {}^{\varepsilon}\!/\!{}_{8\gamma}; f_h^\star) \right\} \right).$$

In the stochastic setting, where the transition dynamic is fixed during interaction, one can hope to replace eluder dimension in the query complexity by disagreement coefficient of the corresponding function classes, by using epoching techniques similar to Section 3.1. We leave this for future work.

### 4.1. Learning from Multiple Teachers

In (Dekel et al., 2012), the problem of selective sampling from multiple teachers is considered with the main motivation being that we can consider each teacher as being an expert in certain contexts or scenarios, and we would like to

**Algorithm 2** Imitation Learning with Expert Feedback, $\mathcal{A} = \{1, 2, \ldots, K\}$

**Require:** Parameters $\delta, \gamma, \lambda, T$, function class $\{\mathcal{F}_h\}_{h \le H}$, online oracle $\mathsf{Oracle}_h$ w.r.t. $\ell_\phi$ for $h \in [H]$.

1: Set $\Psi_\delta^{\ell_\phi}(\mathcal{F}_h, T) \quad = \quad \frac{4}{\lambda} \mathrm{Reg}^{\ell_\phi}(\mathcal{F}_h; T) \quad +$ $\frac{112B}{\lambda^2} \log(4H \log^2(T)/\delta)$.
2: Compute $f_{1,h} = \mathsf{Oracle}_{1,h}(\varnothing)$ for $h \in [H]$.
3: **for** $t = 1$ to $T$ **do**
4: $\quad$ Nature chooses the state $x_{t,1}$.
5: $\quad$ **for** $h = 1$ to $H$ **do**
6: $\quad\quad$ Learner plays $\widehat{y}_{t,h} = \mathsf{SelectAction}(f_{t,h}(x_{t,h}))$
7: $\quad\quad$ Learner transitions to the next state in this round $x_{t,h+1} \leftarrow \mathbb{T}_{t,h}(x_{t,h}, \widehat{y}_{t,h})$.
8: $\quad\quad$ Learner computes

$$\Delta_{t,h} := \max_{f \in \mathcal{F}_h} \|f(x_{t,h}) - f_{t,h}(x_{t,h})\|$$
$$\text{s.t.} \sum_{s=1}^{t-1} Z_{s,h} \|f(x_{s,h}) - f_{s,h}(x_{s,h})\|^2 \le \Psi_\delta^{\ell_\phi}(\mathcal{F}_h, T).$$
$$\text{(6)}$$

9: $\quad\quad$ Learner decides whether to query: $Z_{t,h} = \mathbf{1}\{\mathsf{Margin}(f_{t,h}(x_{t,h})) \le 2\gamma \Delta_{t,h}\}$.
10: $\quad\quad$ **if** $Z_{t,h} = 1$ **then**
11: $\quad\quad\quad$ Learner queries the label $y_{t,h}$ for $x_{t,h}$.
12: $\quad\quad\quad$ $f_{t+1,h} \leftarrow \mathsf{Oracle}_{t+1,h}(\{x_{t,h}, y_{t,h}\})$
13: $\quad\quad$ **else**
14: $\quad\quad\quad$ $f_{t+1,h} \leftarrow f_{t,h}$
15: $\quad\quad$ **end if**
16: $\quad$ **end for**
17: **end for**

learn from their joint feedback. The goal there is to perform not only as well as the best of them individually but even as well as the best combination of them. This motivation is even more lucrative for the IL setting, as we can hope to get policies that perform much better than any single teacher. Continuing with the example of learning to drive from human demonstrations, we might have one human demonstrator who is an expert in highway driving, another human who is an expert in city driving, and the third one in off-road conditions. Each expert is confident in their own terrain, but we would like to learn a policy that can perform well in all terrains.

The formal model is similar to the single-teacher case, but now we have $M$ teachers. For every time step $h \le H$, the $m$-th teacher has an underlying ground truth model $f_h^{\star,m} \in \mathcal{F}_h^m$ that it uses to produce its label, i.e. given a context $x_h$ it draws its label $y_h^m \sim \phi(f^{\star,m}(x_h))$, where $\phi$ is the link function. On rounds in which the learner queries for feedback, it gets back a label from each of the $M$ experts, i.e. $\{y_h^1, \ldots, y_h^M\}$. We next describe the policy $\pi^\star$ that we wish to compete with. Let $\mathscr{A}$ be a mapping, known to

the learner, that combines the recommendation of the $M$ experts to obtain a ground truth label for the corresponding states, i.e. the ground truth label on the context $x_h$ we define $y_h \sim \mathscr{A}(\phi(f^{\star,1}(x_h)), \ldots, \phi(f^{\star,M}(x_h)))$. For example, $\mathscr{A}$ could simply choose the majority action proposed by the experts on $x_h$. Under this ground truth process for labels, the Bayes optimal predictor is simply $\pi^\star(x_h) = \mathsf{SelectAction}(\mathscr{A}(\phi(f^{\star,1}(x_h)), \ldots, \phi(f^{\star,M}(x_h))))$, which is what we wish to compete with. Our main theorem below bounds the number of label queries to teachers, and regret with respect to this $\pi^\star$. Similar to the other results in this paper, our bounds here depend on a margin term $T_{\varepsilon,h}$, that captures the number of rounds in which the Bayes optimal predictor $\pi^\star$ can flip its label if our estimates of the $M$ teachers are off by at most $\varepsilon$ (in $\ell_\infty$ norm). Similar to the single teacher case, we only pay for margin w.r.t. the state distributions induced by $\pi^\star$.

**Theorem 5.** Let $\delta \in (0, 1)$. Under the modeling assumptions above for the multiple experts setting, with probability at least $1 - \delta$, the selective sampling Algorithm 4 (given in the appendix) obtains:

$$\mathrm{Reg}_T = \widetilde{O}\left( \inf_\varepsilon \left\{ H \sum_{h=1}^H T_{\varepsilon,h} \right. \right.$$
$$\left. \left. + \frac{H}{\lambda \varepsilon^2} \sum_{m=1}^M \sum_{h=1}^H \mathrm{Reg}^{\ell_\phi}(\mathcal{F}_h^m; T) + \log(1/\delta) \right\} \right),$$

and,

$$N_T = \widetilde{O}\left( \inf_\varepsilon \left\{ H \sum_{h=1}^H T_{\varepsilon,h} + \log(1/\delta) \right. \right.$$
$$\left. \left. + \frac{H}{\lambda \varepsilon^2} \sum_{h=1}^H \sum_{m=1}^M \mathrm{Reg}^{\ell_\phi}(\mathcal{F}_h^m; T) \cdot \mathfrak{E}(\mathcal{F}_h^m, \varepsilon/8; f_h^{\star,m}) \right\} \right).$$

In Appendix A, we evaluate our IL algorithm on the Cartpole environment, with single and multiple experts. We found that our algorithm can match the performance of passive querying algorithms while making a significantly lesser number of expert queries. Finally, also note that setting $H = 1$ in the above result, recovers an algorithm, and a bound on the regret and query complexity for selective sampling with multiple teachers.

ACKNOWLEDGEMENTS

AS thanks Sasha Rakhlin and Dylan Foster for helpful discussions. AS acknowledges support from the Simons Foundation and NSF through award DMS-2031883, as well as from the DOE through award DE-SC0022199. WS acknowledges support from NSF grant IIS-2154711. KS acknowledges support from NSF CAREER Award 1750575, and LinkedIn-Cornell grant.

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

# A. Experiments

We conduct experiments to verify our theory. To this end, we first introduce the simulator, *Cart Pole* (Barto et al., 1983; Brockman et al., 2016), and then explain the implementation of our algorithm and the baselines. Finally, we present the results.

**Cart Pole.** Cart Pole is a classical control problem, in which a pole is attached by an un-actuated joint to a cart. The goal is to balance the pole by applying force to the cart either towards the left or towards the right (so binary action). The episode is terminated once either the pole is out of balance or the cart deviates too far from the origin. A reward of 1 is obtained in each time step (however, the algorithm does not get any reward signal). The observations are four-dimensional, with the values representing the cart's position, velocity, the pole's angle, and angular velocity. The action is binary, indicating the force is either to the left or to the right.

**Expert policies generation.** We first generate an optimal policy $\pi^\star$ (that attains the maximum possible reward of 500) by policy gradient. We notice that when running the optimal policy $\pi^\star$, the absolute value of the cart's position only lies in $[0, 2]$. Hence, to generate $M$ experts, we first divide this interval into $M$ sub-intervals $[a_0, a_1], [a_1, a_2], \ldots, [a_{n-1}, a_M]$ ($a_0 = 0$ and $a_M = 2$) by geometric progression. For the $i$-th expert, it plays the same action as $\pi^\star$ when the absolute value of the cart's position is in the interval $[a_{i-1}, a_i]$ and plays uniformly at random outside of this interval. We find that using such generation, each expert individually cannot achieve a good performance (when $M > 1$), while a proper combination of them can still be as strong as $\pi^\star$. We conduct experiment for $M = 1, 2, 3$, and $5$, respectively. Given this design of expert generation, when the cart is in the sub-interval $[a_{i-1}, a_i]$, the only expert with non-zero margin is exactly the $i$-th expert.

**Implementation.** The algorithm is similar to Algorithm 4 but with some modification for practical purpose. First, we use a neural network (single hidden layer neural network, with 4 neurons in the hidden layer) as our function class $\{\mathcal{F}_h^m\}_{h \leq H, m \leq M}$. Second, we specify SelectAction() to pick the action of the most confident expert, i.e.,

$$\texttt{SelectAction}(f_{t,h}^1(x), \ldots, f_{t,h}^M(x)) \coloneqq \text{sign}(f_{t,h}^{\hat{i}}(x)) \quad \text{where} \quad \hat{i} = \operatorname*{arg\,max}_{i \in [M]} |f_{t,h}^i(x)|.$$

Since we are considering binary action, we assume $f_{t,h}^i(x) \in [-1, 1]$, and the action space is $\{-1, 1\}$. Third, to compute $\Delta_{t,h}^m$ efficiently, we apply the Lagrange multiplier to (57) to arrive at the following equivalent problem:

$$\Delta_{t,h}^m(x_{t,h}) \coloneqq \min_{f \in \mathcal{F}_h^m} \max_{\alpha \geq 0} -\|f(x_{t,h}) - f_{t,h}^m(x_{t,h})\|$$

$$+ \alpha \left( \sum_{s=1}^{t-1} Z_{s,h} \|f(x_{s,h}) - f_{s,h}^m(x_{s,h})\|^2 - \Psi_\delta^{\ell_\phi}(\mathcal{F}_h^m, T) \right).$$

Then we treat the Lagrange multiplier $\alpha$ as a constant, which converts the problem into the following:

$$\Delta_{t,h}^m(x_{t,h}) \coloneqq \min_{f \in \mathcal{F}_h^m} -\|f(x_{t,h}) - f_{t,h}^m(x_{t,h})\| + \alpha \sum_{s=1}^{t-1} Z_{s,h} \|f(x_{s,h}) - f_{s,h}^m(x_{s,h})\|^2. \tag{7}$$

The study of varying $\alpha$ is shown in Figure 1. We found that small values (e.g., $\alpha = 1$) mostly lead to poor performance, while the results are fairly similar for large values. In our key experiments, we choose $\alpha = 50$ when the number of experts is 1, 2 or 3, and choose 200 for 5-expert experiments. We note that since computing (7) for each time step involves repetitively fitting neural networks, which is time-consuming, we do a warm start at each round. In particular, we set the initial weights for the neural network of each round to be the weights of the trained network from the previous round. We also implemented *early stopping* that stops the iteration if the loss does not significantly decrease for multiple consecutive iterations. The online regression oracle Oracle() is instantiated as applying gradient descent for certain steps on the mean squared loss over all data collected so far, using warm start for speedup as well.

We first conduct experiments on a single expert setting. In Figure 2 we plot the curves of return and number of queries with respect to iterations for our method, and compare to DAgger (which passively makes queries at every time step; (Ross & Bagnell, 2014)). We note that while our algorithm does not converge to the optimal value as fast as DAgger, the number of queries made by our algorithm is significantly fewer, which means that our method is indeed balancing the speed of learning and the number of queries.

In additional to DAgger, we also compare to the following baselines:

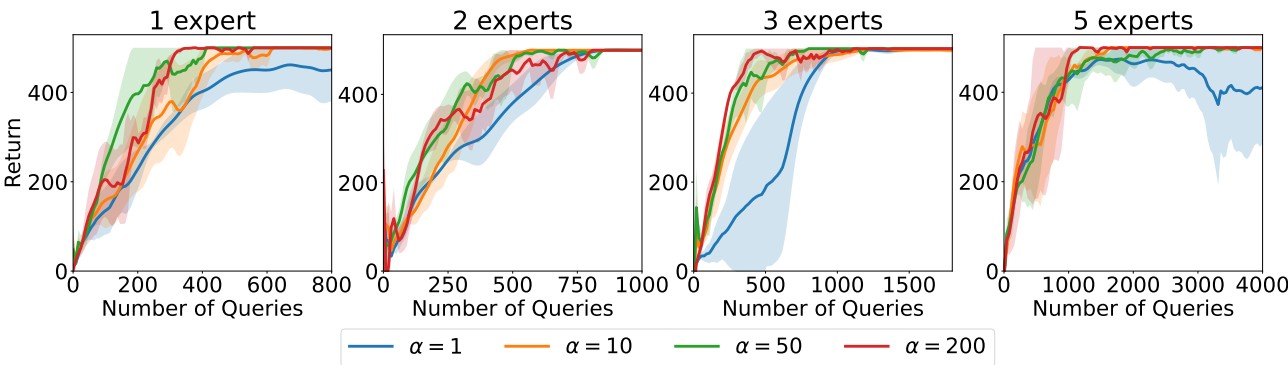

*Figure 1.* Learning curves of return with respect to the number of queries for different values of $\alpha$ and different numbers of experts.

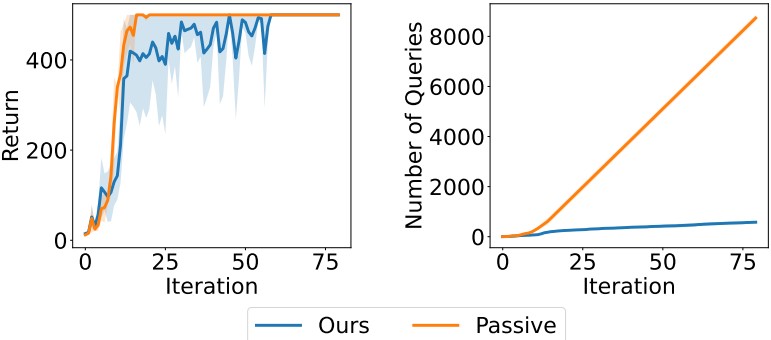

*Figure 2.* Learning curves of the return and the number of queries for 1 expert.

- **Passive learning.** By passive learning, we mean running our algorithms with $Z_{t,h} = 1$, i.e., making queries whenever possible. Based on different styles of expert feedback, we divide the passive learning baselines into two: *noisy experts* and *noiseless experts*. For the former we get the noisy label $y_{t,h}^m$ for $x_{t,h}$ (generated by $y_{t,h}^m \sim \phi(f_h^{\star,m}(x_{t,h}))$), and for the latter we directly get the action of the optimal policy (i.e. the action $\pi_h^\star(x_{t,h})$). Intuitively, noiseless feedback is more helpful than the noisy one.

- **MAMBA.** We compare our algorithm with (a slight variant of) MAMBA (Cheng et al., 2020). At each time step, it creates copies of the environment and run each expert policy on these copies, and then it selects the action of the expert policy with the highest return. For simplicity, we refer to this algorithm as MAMBA. Note that MAMBA assumes that one has access to the underlying reward function. Thus this baseline is using significantly more information than our approach.

- **Best expert.** We also compared our algorithm with the best expert policy.

The main results are shown in Figure 3. We first noticed that our algorithm outperforms passive learning with noisy experts in all settings. Moreover, we beat the noiseless version when there is only one expert. Intuitively, getting feedback from noiseless experts is a very strong assumption and it is not surprising to see that the performance is improved with this stronger feedback. Note that our algorithm is only getting noisy labels as feedback. We also note that, despite the fact that MAMBA achieves better results than the best expert policy (in terms of the value function), it is still worse than our algorithm. Indeed, MAMBA does not even learn a policy that can solve the task when $M \geq 2$. This is because by our construction of experts, there is no single expert that is capable of solving the task alone. Note that MAMBA performs well in the one expert case because in that case, the (single) expert can reliably solve the control task.

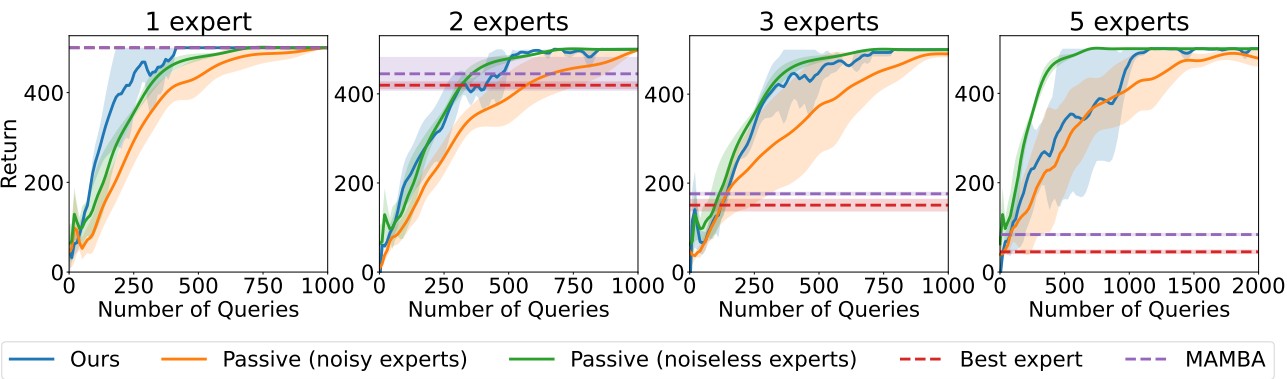

*Figure 3.* Learning curves of return with respect to the number of queries for different algorithms and numbers of experts.

# B. Useful Tools and Notation

**Additional notation.** Throughout the paper, we assume that the ties are broken arbitrarily but consistently. Vector-valued variables are denoted with small alphabets like $u, v$, etc, and matrix-valued variables are denoted with capital alphabets like $F, G$, etc. Throughout the paper, we assume that for any $f \in \mathcal{F}$, $\|f(x)\| \le B \le 1$.

The following lemma is used throughout the appendix, and whose proof is a tautology.

**Lemma 1.** Let $\mathcal{E}_1$ and $\mathcal{E}_2$ be any two events such that $\mathcal{E}_1 \implies \mathcal{E}_2$ then $\mathbf{1}\{\mathcal{E}_1\} \le \mathbf{1}\{\mathcal{E}_2\}$.

## B.1. Basic Probabilistic Tools

**Lemma 2** (Theorem 1 in Srebro et al. (2010)). Let $T > 0$, and let $\mathcal{F} = \{\mathcal{X} \times \mathcal{Y}\}$ be an arbitrary function class, and $\ell$ be an $\gamma$-smooth and non-negative loss such that $|\ell(f(x), y)| \le B$ for all $x \in \mathcal{X}, y \in \mathcal{Y}, f \in \mathcal{F}$. For any $\delta > 0$, we have with probability at least $1 - \delta$ over a random sample of size $T$, for any $f \in \mathcal{F}$,

$$T \, \mathbb{E}_{(x,y)\sim\mu}[\ell(f(x), y)] \le 2 \sum_{t=1}^{T} \ell(f(x_t), y_t) + c_1\big(HT \log^3(T)\mathcal{R}_T^2(\mathcal{F}) + B \log(1/\delta)\big)$$

where $c_1 < 10^5$ is a numeric constant.

The precise value of the numeric constant $c_1$ in the above can be derived from Srebro et al. (2010) and Mendelson (2002).

The following inequalities are well-known; we use the version stated in Zhu & Nowak (2022).

**Lemma 3** (Freedman's inequality). Let $\{X_t\}_{t\le T}$ be a real-valued martingale different sequence adapted to the filtration $\mathfrak{F}_t$, and let $\mathbb{E}_t[\cdot] := \mathbb{E}[\cdot \mid \mathfrak{F}_{t-1}]$. If $|X_t| \le B$ almost surely, then for any $\eta \in (0, 1/B)$, the following holds with probability at least $1 - \delta$:

$$\sum_{t=1}^{T} X_t \le \eta \sum_{t=1}^{T} \mathbb{E}_t[X_t^2] + \frac{B \log(1/\delta)}{\eta}.$$

**Lemma 4.** Let $\{X_t\}_{t\le T}$ be a sequence of positive valued random variables adapted to the filtration $\mathfrak{F}_t$, and and let $\mathbb{E}_t[\cdot] := \mathbb{E}[\cdot \mid \mathfrak{F}_{t-1}]$. If $X_t \le B$ almost surely, then with probability at least $1 - \delta$,

$$\sum_{t=1}^{T} X_t \le \frac{3}{2} \sum_{t=1}^{T} \mathbb{E}_t[X_t] + 4B \log(2/\delta),$$

and

$$\sum_{t=1}^{T} \mathbb{E}_t[X_t] \le 2 \sum_{t=1}^{T} X_t + 8B \log(2/\delta).$$

## B.2. Online Learning

**Lemma 5.** Suppose that the labels are generated according to the (2) where the link function satisfies Assumption 1. Additionally, assume that the regression oracle satisfies the guarantee (3). Then, for any $\delta \leq 1/e$ and $T \geq 3$, with probability at least $1 - \delta$, we have for all $t \leq T$,

$$\sum_{s=1}^{t} \|f_s(x_s) - f^\star(x_s)\|^2 \leq \Psi_\delta^{\ell_\phi}(\mathcal{F}, T) := \frac{4}{\lambda} \mathrm{Reg}^{\ell_\phi}(\mathcal{F}; T) + \frac{112B}{\lambda^2} \log(4 \log^2(T)/\delta),$$

where $B$ is defined such that $\sup_x f(x) \leq B$.

*Proof.* Using Agarwal (2013, Lemma 2) along with an Union bound implies that for all $t \leq T$,

$$\sum_{s=1}^{t} \|f_s(x_s) - f^\star(x_s)\|^2 \leq \frac{4}{\lambda} \sum_{s=1}^{t} (\ell_\phi(f_s(x_s), y_s) - \ell_\phi(f^\star(x_s), y_s)) + \frac{112B}{\lambda^2} \log(4 \log^2(T)/\delta).$$

Plugging in the regret bound (3) in the above, we get that

$$\sum_{s=1}^{t} \|f_s(x_s) - f^\star(x_s)\|^2 \leq \sum_{s=1}^{T} \|f_s(x_s) - f^\star(x_s)\|^2 \leq \frac{4}{\lambda} \mathrm{Reg}^{\ell_\phi}(\mathcal{F}; T) + \frac{112B}{\lambda^2} \log(4 \log^2(T)/\delta).$$

$\square$

## B.3. Eluder Dimension, Disagreement Coefficient, and Star Number

For the sake of completeness, we recall the scalar versions of scale-sensitive eluder dimension, and disagreement coefficient introduced in Russo & Van Roy (2013); Foster et al. (2020), which is defined for scalar valued function class $\mathcal{F} \subseteq \{\mathcal{X} \mapsto \mathbb{R}\}$.

**Definition 4** (Scale-sensitive eluder dimension (scalar version), Russo & Van Roy (2013); Foster et al. (2020)). Let $\mathcal{F} \subseteq \{\mathcal{X} \mapsto \mathbb{R}\}$. Fix any $f^\star \in \mathcal{F}$, and define $\mathfrak{E}'(\mathcal{F}, \beta; f^\star)$ to be the length of the longest sequence of contexts $x_1, x_2, \ldots x_m$ such that for all $i$, there exists $f_i \in \mathcal{F}$ such that

$$|f_i(x_i) - f^\star(x_i)| > \beta, \quad \text{and} \quad \sum_{j<i} (f_i(x_j) - f^\star(x_j))^2 \leq \beta^2.$$

We define the scale-sensitive eluder dimension as $\mathfrak{E}(\mathcal{F}, \beta_0; f^\star) := \sup_{\beta_0 \geq \beta} \mathfrak{E}'(\mathcal{F}, \beta; f^\star)$.

**Definition 5** (Scale-sensitive disagreement coefficient (scalar version), (Foster et al., 2020)). Let $\mathcal{F} \subseteq \{\mathcal{X} \mapsto \mathbb{R}\}$. For any $f^\star \in \mathcal{F}$, and $\gamma_0, \varepsilon_0 > 0$, the value function disagreement coefficient $\theta^{\mathrm{val}}(\mathcal{F}, \varepsilon_0, \gamma_0; f^\star)$ is defined as

$$\sup_{\mu} \sup_{\gamma > \gamma_0, \varepsilon > \varepsilon_0} \left\{ \frac{\varepsilon^2}{\gamma^2} \cdot \Pr_{x \sim \mu}(\exists f \in \mathcal{F} \mid |f(x) - f^\star(x)| > \varepsilon, \|f - f^\star\|_\mu \leq \gamma) \right\} \vee 1$$

where $\|f\| = \sqrt{\mathbb{E}_{x \sim \mu}[f^2(x)]}$.

While Definition 4 and 5 define the eluder dimension and disagreement coefficient for scalar valued functions, in this work we often deal with vector-valued functions. In the following, we extend the above definitions to vector-valued functions. We first provide the normed versions which are direct extensions of the corresponding definitions (mentioned above) for the scalar case as introduced in Russo & Van Roy (2013); Foster et al. (2020). We first defined the normed version of eluder dimension.

**Definition 6** (Scale-sensitive eluder dimension (normed version)). Let $\mathcal{F} \subseteq \{\mathcal{X} \mapsto \mathbb{R}^K\}$. Fix any $f^\star \in \mathcal{F}$, and define $\widetilde{\mathfrak{E}}(\mathcal{F}, \beta; f^\star)$ to be the length of the longest sequence of contexts $x_1, x_2, \ldots x_m$ such that for all $i$, there exists $f_i \in \mathcal{F}$ such that

$$\|f_i(x_i) - f^\star(x_i)\| > \beta, \quad \text{and} \quad \sum_{j<i} \|f_i(x_j) - f^\star(x_j)\|^2 \leq \beta^2.$$

The value function eluder dimension is defined as $\mathfrak{E}(\mathcal{F}, \beta'; f^\star) = \sup_{\beta \geq \beta'} \widetilde{\mathfrak{E}}(\mathcal{F}, \beta; f^\star)$.

We next define the normed version of disagreement coefficient for vector-valued functions.

**Definition 7** (Scale sensitive disagreement coefficient (normed version), (Foster et al., 2020)). Let $\mathcal{F} \subseteq \{\mathcal{X} \mapsto \mathbb{R}^K\}$. For any $f^\star \in \mathcal{F}$, and $\beta_0, \varepsilon_0 > 0$, the value function disagreement coefficient $\theta^{\mathrm{val}}(\mathcal{F}, \varepsilon_0, \beta_0; f^\star)$ is defined as

$$\sup_\mu \sup_{\beta > \beta_0, \varepsilon > \varepsilon_0} \left\{ \frac{\varepsilon^2}{\beta^2} \cdot \mathrm{Pr}_{x \sim \mu}(\exists f \in \mathcal{F} \mid \|f(x) - f^\star(x)\| > \varepsilon, \|f - f^\star\|_\mu \le \beta) \right\} \vee 1$$

where $\|f\|_\mu = \sqrt{\mathbb{E}_{x \sim \mu}[\|f(x)\|^2]}$.

We additionally also define the following mixed version of eluder dimension for vector valued functions.

**Definition 8** (Scale-sensitive eluder dimension (mixed version)). Let $\mathcal{F} \subseteq \{\mathcal{X} \mapsto \mathbb{R}^K\}$. Fix any $f^\star \in \mathcal{F}$, and define $\check{\mathfrak{E}}'(\mathcal{F}, \beta; f^\star)$ to be the length of the longest sequence of contexts and actions $(x_1, y_1), (x_2, y_2) \dots (x_m, y_m)$ such that for all $i$, there exists $f_i \in \mathcal{F}$ such that

$$|f_i(x_i)[y_i] - f^\star(x_i)[y_i]| > \beta, \quad \text{and} \quad \sum_{j < i} (f_i(x_j)[y_j] - f^\star(x_j)[y_j])^2 \le \beta^2.$$

We define the scale sensitive eluder dimension (mixed version) as $\check{\mathfrak{E}}(\mathcal{F}, \beta; f^\star) := \sup_{\beta \ge \beta_0} \check{\mathfrak{E}}'(\mathcal{F}, \beta_0; f^\star)$.

The star number is bounded by the eluder dimension from above: $\mathfrak{s}^{\mathrm{val}}(\mathcal{F}, \beta; f^\star) \le \mathfrak{E}(\mathcal{F}, \beta; f^\star)$. The following lemma shows that the disagreement coefficient can be always bounded by the squared star number and eluder dimension.

**Lemma 6** (Proposition 6, Foster et al. (2020)). Suppose $\mathcal{F} \subseteq \{\mathcal{X} \mapsto \mathbb{R}^K\}$ is a uniform Glivenko-Cantelli class. For any $f^\star \in \mathcal{F}$ and $\gamma, \varepsilon > 0$, we have $\theta^{\mathrm{val}}(\mathcal{F}, \varepsilon, \gamma; f^\star) \le 4(\mathfrak{s}^{\mathrm{val}}(\mathcal{F}, \gamma; f^\star))^2$ and $\theta^{\mathrm{val}}(\mathcal{F}, \varepsilon, \gamma; f^\star) \le 4\mathfrak{E}(\mathcal{F}, \gamma; f^\star)$.

We refer the reader to (Foster et al., 2020) for bounds on eluder dimension, disagreement coefficient and star number for various function classes.

We next provide various technical results that are useful in bounding the total number of queries made by our algorithm. We first provide a technical result which bounds the number of times we can find a function $f'$ in a refinement $\mathcal{F}_t$ of $\mathcal{F}$, such that $f'$ is sufficiently far away from $f^\star \in \mathcal{F}$. The following lemma is a variant of Russo & Van Roy (2013, Lemma 3), and first appears in Foster et al. (2020, Lemma E.4).

**Lemma 7.** Let $\{(x_t, y_t), Z_t\}_{t=1}^T$ be sequence of tuples, where $x_t \in \mathcal{X}$ and $Z_t \in \{0, 1\}$. Fix any $f^\star \in \mathcal{F}$, and define the set $\mathcal{F}_t = \{f \in \mathcal{F} \mid \sum_{s=1}^{t-1} Z_s (f(x_s)[y_s] - f^\star(x_s)[y_s])^2 \le \beta^2\}$. Then, for any $\zeta > 0$,

$$\sum_{t=1}^T Z_t \mathbf{1}\{\exists f' \in \mathcal{F}_t : \|f'(x_t) - f^\star(x_t)\| \ge \zeta\} \le \left( \frac{\beta^2}{\zeta^2} + 1 \right) \check{\mathfrak{E}}(\mathcal{F}, \zeta; f^\star).$$

*Proof.* We first note that we can always remove a tuple $\{(x_t, y_t), Z_t\}$ whenever $Z_t = 0$ without any effect on the conclusion. Hence, we can assume $Z_t = 1$ for all $t \in [T]$ without loss of generality. Then the rest of the proof essentialy follows from Foster et al. (2020, Lemma E.4). For completeness, we state the full proof here.

For simplicity of presentation, we say $(x_t, y_t)$ is $\zeta$-independent of $(x_1, y_1), \dots, (x_{t-1}, y_{t-1})$ if there exists $f \in \mathcal{F}$ such that $|f(x_t)[y_t] - f^\star(x_t)[y_t]| \ge \zeta$ and $\sum_{s=1}^{t-1}(f(x_s)[y_s] - f^\star(x_s)[y_s])^2 \le \zeta^2$. Otherwise, we say $x$ is $\zeta$-dependent. The proof consists of the following two claims.

First, we claim that for any $t \in [T]$, if there exists $f \in \mathcal{F}_t$ such that $|f(x_t)[y_t] - f^\star(x_t)[y_t]| \ge \zeta$, then $x_t$ is $\zeta$-dependent on at most $\beta^2/\zeta^2$ disjoint sequences of $(x_1, y_1), \dots, (x_{t-1}, y_{t-1})$. To show this, let's say $x_t$ is $\zeta$-dependent on a particular subsequence $(x_{i_1}, y_{i_1}), \dots, (x_{i_k}, y_{i_k})$ while $|f(x)[y] - f^\star(x)[y]| \ge \zeta$. Then it must holds that

$$\sum_{j=1}^k \left( f(x_{i_j})[y_{i_j}] - f^\star(x_{i_j})[y_{i_j}] \right)^2 \ge \zeta^2.$$

If there are $M$ such disjoint subsequence, then we can add them up and obtain the following:

$$\sum_{s=1}^{t-1} \left( f(x_s)[y_s] - f^\star(x_s)[y_s] \right)^2 \ge M\zeta^2.$$

By the construction of $\mathcal{F}_t$, the left-hand side above is at most $\beta^2$. Hence we conclude that $\beta^2 \geq M\zeta^2$, which implies $M \leq \beta^2/\zeta^2$.

Second, we claim that for any $k$ and any sequence $(x_1, y_1), \ldots, (x_k, y_k)$, there exists $j \leq k$ such that $x_j$ is $\zeta$-dependent on at least $N \coloneqq \lfloor k / \check{\mathfrak{E}}(\mathcal{F}, \zeta; f^\star) \rfloor$ disjoint subsequences of $(x_1, y_1), \ldots, (x_{j-1}, y_{j-1})$. This can be proved by construction. Let $B_1, \ldots, B_N$ be $N$ subsequences of $(x_1, y_1), \ldots, (x_k, y_k)$ and are initialized with $B_i = \{(x_i, y_i)\}$. Then we repeat the following process for $j = N+1, N+2 \ldots, k$.

1. We first check if $x_j$ is $\zeta$-dependent on $B_i$ for all $i \in [N]$. If so, we are done.

2. Otherwise, pick an arbitrary $i \in [N]$ for which $x_j$ is $\zeta$-independent of $B_i$ and append $(x_j, y_j)$ to $B_i$, i.e., $B_i \leftarrow B_i \cup \{(x_j, y_j)\}$.

If we don't reach any $j$ while running the above process for which the first statement above is satisfied, we should end up with $\sum_{i=1}^N |B_i| = k \geq N \cdot \check{\mathfrak{E}}(\mathcal{F}, \zeta; f^\star)$. We note that by construction $|B_i| \leq \check{\mathfrak{E}}(\mathcal{F}, \zeta; f^\star)$ and thus $|B_i| = \check{\mathfrak{E}}(\mathcal{F}, \zeta; f^\star)$ for all $i \in [N]$, which implies $x_k$ must be $\zeta$-dependent on all $B_i$.

Finally, let $x_{i_1}, \ldots, x_{i_k}$ be the subsequence where, for all $s \in [k]$, there exists $f \in \mathcal{F}_{i_s}$ such that $|f(x_{i_s})[y_{i_s}] - f^\star(x_{i_s})[y_{i_s}]| \geq \zeta$. By our first claim we know each element of this subsequence is $\zeta$-dependent on at most $\beta^2/\zeta^2$ disjoint subsequences. By the second claim, we know that there exists an element that is $\zeta$-dependent on at least $\lfloor k / \check{\mathfrak{E}}(\mathcal{F}, \zeta; f^\star) \rfloor$ disjoint subsequences. So we must have $\lfloor k / \check{\mathfrak{E}}(\mathcal{F}, \zeta; f^\star) \rfloor \leq \beta^2/\zeta^2$. Hence, $k \leq (\beta^2/\zeta^2 + 1) \cdot \check{\mathfrak{E}}(\mathcal{F}, \zeta; f^\star)$. $\qquad\square$

The following lemma is an extension of Lemma 7 that holds for the normed version of eluder dimension given in Definition 1. The proof is essentially the same as that of Lemma 7, so we skip it for conciseness.

**Lemma 8.** Let $\{x_t, Z_t\}_{t=1}^T$ be sequence of tuples, where $x_t \in \mathcal{X}$ and $Z_t \in \{0, 1\}$. Fix any $f^\star \in \mathcal{F}$, and define the set $\mathcal{F}_t = \{f \in \mathcal{F} \mid \sum_{s=1}^{t-1} Z_s \|f(x_s) - f^\star(x_s)\|^2 \leq \beta^2\}$. Then, for any $\zeta > 0$,

$$\sum_{t=1}^T Z_t \mathbf{1}\{\exists f' \in \mathcal{F}_t : \|f'(x_t) - f^\star(x_t)\| \geq \zeta\} \leq \left(\frac{\beta^2}{\zeta^2} + 1\right) \mathfrak{E}(\mathcal{F}, \zeta; f^\star).$$

# C. Selective Sampling

## C.1. Comparison to Related Works

There is a large bank of both theoretical and empirical work for active learning and selective sampling. Perhaps the work closest to ours that we will now compare to is Zhu & Nowak (2022). In this paper, the authors consider binary classification problem and provide bounds on number of queries and bound on excess risk in the active learning framework. Their algorithm also relies on regression oracle. However, there are many key differences. First, their guarantees for regret for selective sampling problem (see for instance Theorem 10 on page 28 of Zhu & Nowak (2022)) has a dependence on disagreement coefficient in the regret bound as well as number of queries. As we show in our work, one only needs to pay for eluder dimension or disagreement coefficient in query complexity and not in regret bound. Further, we supplement our result with lower bound showing that unless one has label complexity that depends on star number (and hence can be also related to worst case disagreement coefficient), one cant get a small enough regret bound. So the separation between regret bound (that is independent of eluder dimension/star number/disagreement coefficient) and query complexity (that depends on those quantities) is real. Additionally the results in (Zhu & Nowak, 2022) dont automatically adapt to the margin region and in general there is no way to estimate the parameters of Tsybakov's noise condition. Finally their regret bounds depend on pseudo dimension and is generally suboptimal for complex $\mathcal{F}$.

## C.2. Proof Sketch for Selective Sampling and Binary Labels

To illustrate our modeling assumptions, let us start with the simpler setting of binary actions $\mathcal{A} = \{0, 1\}$. In this case, we assue that for any context $x$, the label $y$ is drawn as $y_t \sim \text{Ber}\left(\frac{1+f^\star(x)}{2}\right)$ where $f^\star \in \mathcal{F}$ for a given model class $\mathcal{F} \subseteq [-1, 1]^{\mathcal{X}}$. While we defined $\mathcal{F}$ to be a real-valued function class for the ease of notation, we note that for any $f \in \mathcal{F}$, the score on the context $x$ can be obtained as $\frac{1}{2}(1 + f(x), 1 - f(x))^\top \in [0, 1]^2$, and thus the Bayes optimal predictor that chooses the action with the larger score is given by $\texttt{SelectAction}(f^\star(x)) = \text{sign}(f^\star(x))$. Furthermore, for the binary actions setting, the natural notion of margin is defined as $\texttt{Margin}(f^\star(x)) = |\Pr(y = 1 \mid x) - \Pr(y = 0 \mid x)| = |f^\star(x)|$, which implies that for any $\varepsilon > 0$ and the observed context sequence $\{x_t\}_{t \le T}$, we set $T_\varepsilon = \sum_{t=1}^T \mathbf{1}\{|f^\star(x_t)| \le \varepsilon\}$.

For this simple model, we assume that we have access to an online square loss regression oracle w.r.t. $\mathcal{F}$ that for any sequence $\{(x_1, y_1), \ldots, (x_T, y_T)\}$ guarantees the regret bound

$$\sum_{s=1}^T (f_s(x_s) - y_s)^2 - \inf_{f \in \mathcal{F}} \sum_{s=1}^T (f(x_s) - y_s)^2 \le \text{Reg}^{\text{sq}}(\mathcal{F}; T). \tag{8}$$

Before we go to the main result for binary actions and the proof sketch, we first remark that, by now, we have a relatively complete characterization of what regret bounds $\text{Reg}^{\text{sq}}(\mathcal{F}; T)$ are possible for general function classes $\mathcal{F}$. (Rakhlin & Sridharan, 2014) characterize the minimax rates for online square loss regression in terms of the offset sequential Rademacher complexity, thus giving favorable bounds for $\text{Reg}^{\text{sq}}(\mathcal{F}; T)$. For instance, for finite classes $\mathcal{F}$ we have $\text{Reg}^{\text{sq}}(\mathcal{F}; T) \le O(\log|\mathcal{F}|)$ and for $d$ dimensional linear classes it is give by $\text{Reg}^{\text{sq}}(\mathcal{F}; T) \le O(d \log(T))$. We also refer the reader to (Krishnamurthy et al., 2017; Foster et al., 2018a; Foster & Rakhlin, 2020) for discussions on efficient implementations.

**Theorem 1** (Specialization for binary actions)**.** Let $\delta \in (0, 1)$. Under the modeling assumptions above, with probability at least $1 - \delta$, Algorithm 1 run with $\mathcal{A} = \{0, 1\}$ obtains the regret bound:

$$\text{Reg}_T = \widetilde{O}\left(\inf_\varepsilon \left\{\varepsilon T_\varepsilon + \frac{\text{Reg}^{\text{sq}}(\mathcal{F}; T)}{\varepsilon} + \log(T/\delta)\right\}\right),$$

while simultaneously the total number of label queries made is bounded by:

$$N_T = \widetilde{O}\left(\inf_\varepsilon \left\{T_\varepsilon + \frac{\mathfrak{E}(\mathcal{F}, \varepsilon; f^\star) \cdot \text{Reg}^{\text{sq}}(\mathcal{F}; T)}{\varepsilon^2} + \log(T/\delta)\right\}\right)$$

In the following, we provide an informal proof sketch for Theorem 1 (obtained by running Algorithm 1 with $\mathcal{A} = \{0, 1\}$ and $\gamma = \lambda = 1$). We first note that the regret bound in (8) for the online regression Oracle implies that:

$$\sum_{s=1}^t Z_s(f_s(x_s) - f^\star(x_s))^2 \lesssim \text{Reg}^{\text{sq}}(\mathcal{F}; T) + \log(T/\delta), \tag{9}$$

The above implies that $f^\star$ satisfies the constraints in (5) and thus (See Lemma 10)

$$|f_t(x_t) - f^\star(x_t)| \le \Delta_t(x_t). \tag{10}$$

Now since our query condition was $Z_t = \mathbf{1}\{|f_t(x_t)| \le \Delta_t(x_t)\}$, we have that if $Z_t = 0$, then sign of $f^\star(x_t)$ and that of $f_t(x_t)$ is the same and so our predictor matches $\pi^\star$. In other words, when we don't query (ie. when $\bar{Z}_t = 1$, $\pi^\star(x_t) := \text{sign}(f^\star(x_t)) = \text{sign}(f_t(x_t))$).

**Regret bound.** Using the fact that $y_t \sim \text{Ber}\left(\frac{1+f^\star(x_t)}{2}\right), \widehat{y}_t = \text{sign}(f_t(x_t))$, we note that

$$\text{Reg}_T = \sum_{t=1}^{T} \Pr(\widehat{y}_t \ne y_t) - \Pr(\text{sign}(f^\star(x_t)) \ne y_t)$$

$$\le \sum_{t=1}^{T} \mathbf{1}\{\text{sign}(f_t(x_t)) \ne \text{sign}(f^\star(x_t))\} \cdot |2\Pr(y_t = 1) - 1|$$

$$= \sum_{t=1}^{T} \mathbf{1}\{\text{sign}(f_t(x_t)) \ne \text{sign}(f^\star(x_t))\} \cdot |f^\star(x_t)|$$

One can further split the right hand side above and upper bound via the following three terms:

$$\text{Reg}_T \le \varepsilon \sum_{t=1}^{T} \mathbf{1}\{|f^\star(x_t)| \le \varepsilon\} + \sum_{t=1}^{T} Z_t \mathbf{1}\{\text{sign}(f_t(x_t)) \ne \text{sign}(f^\star(x_t)), |f^\star(x_t)| > \varepsilon\} \cdot |f^\star(x_t)|$$

$$+ \sum_{t=1}^{T} \bar{Z}_t \mathbf{1}\{\text{sign}(f_t(x_t)) \ne \text{sign}(f^\star(x_t))\} \cdot |f^\star(x_t)|.$$

$$= \varepsilon T_\varepsilon + \underbrace{\sum_{t=1}^{T} Z_t \mathbf{1}\{\text{sign}(f_t(x_t)) \ne \text{sign}(f^\star(x_t)), |f^\star(x_t)| > \varepsilon\} \cdot |f^\star(x_t)|}_{:=T_A}$$

- $T_A$ denotes the regret for the rounds in which the learner queries for the label and the margin for $f^\star(x_t)$ is larger than $\varepsilon$. We note that

$$T_A = \sum_{t=1}^{T} Z_t \mathbf{1}\{\text{sign}(f^\star(x_t)) \ne \text{sign}(f_t(x_t)), |f^\star(x_t)| > \varepsilon\} \cdot |f^\star(x_t)|$$

$$\le \sum_{t=1}^{T} Z_t \mathbf{1}\{|f^\star(x_t) - f_t(x_t)| > \varepsilon\} \cdot |f^\star(x_t) - f_t(x_t)|$$

where the last line just plugs in the fact that $|f^\star(x_t) - f_t(x_t)| \ge |f^\star(x_t)|$ since they have opposite signs. Using the fact that $\mathbf{1}\{a \ge b\} \le a/b$ for all $a, b \ge 0$ in the above, we get

$$T_A \le \frac{1}{\varepsilon} \sum_{t=1}^{T} Z_t (f^\star(x_t) - f_t(x_t))^2 \lesssim \text{Reg}^{\text{sq}}(\mathcal{F}; T) + \log(T/\delta),$$

where the second inequality follows from (9).

Gathering the bounds above completes the proof for $\text{Reg}_T$.

**Bound on $N_T$.** Plugging in the query rule, and splitting as in the regret bound, we get

$$N_T = \sum_{t=1}^{T} Z_t = \sum_{t=1}^{T} \mathbf{1}\{|f_t(x_t)| \le \Delta_t(x_t)\}$$

$$\le \underbrace{\sum_{t=1}^{T} \mathbf{1}\{|f^\star(x_t)| \le \varepsilon\}}_{=T_\varepsilon} + \underbrace{\sum_{t=1}^{T} \mathbf{1}\{|f_t(x_t)| \le \Delta_t(x_t), |f^\star(x_t)| > \varepsilon, \Delta_t(x_t) \le \varepsilon/3\}}_{:=T_C}$$

$$+ \underbrace{\sum_{t=1}^{T} \mathbf{1}\{|f_t(x_t)| \le \Delta_t(x_t), |f^\star(x_t)| > \varepsilon, \Delta_t(x_t) > \varepsilon/3\}}_{:=\mathrm{T}_D}$$

- $\mathrm{T}_C$ denotes the rounds in which we make a query, $\Delta_t(x_t) \le \varepsilon/3$, and the margin for $f^\star(x_t)$ is larger than $\varepsilon$. From (10), note that $|f_t(x_t) - f^\star(x_t)| \le \Delta_t(x_t)$, which further implies that

$$|f^\star(x_t)| \le |f_t(x_t) - f^\star(x_t)| + |f_t(x_t)| \le \Delta_t(x_t) + |f_t(x_t)|.$$

  Thus,

$$\mathrm{T}_C \le \sum_{t=1}^{T} \mathbf{1}\{|f^\star(x_t)| \le 2\Delta_t(x_t), |f^\star(x_t)| > \varepsilon, \Delta_t(x_t) \le \varepsilon/3\} = 0.$$

- $\mathrm{T}_D$ is bounded by the number of rounds for which we make a query and $\Delta_t(x_t) \ge \varepsilon/3$. Using the standard eluder dimension machinery, we get that

$$\mathrm{T}_D \le \sum_{t=1}^{T} Z_t \mathbf{1}\{\Delta_t(x_t) \ge \varepsilon/3\} \lesssim \frac{1}{\varepsilon^2} \mathrm{Reg}^{\mathrm{sq}}(\mathcal{F}; T) \cdot \mathfrak{E}(\mathcal{F}, \varepsilon/6; f^\star) + \log(1/\delta).$$

Gathering the above bounds completes the bound on $N_T$.

### C.3. Proof of Theorem 1

Before delving into the proof, we recall the relevant notation. In Algorithm 1,

- The label $y_t \sim \phi(f^\star(x_t))$, where $\phi$ denotes the link-function given in (2).

- The function $\texttt{SelectAction}(f_t(x_t)) := \mathrm{argmax}_k \phi(f_t(x_t))[k]$.

- For any vector $v \in \mathbb{R}^K$, the margin is given by the gap between the value at the largest and the second largest coordinate, i.e.

$$\texttt{Margin}(v) = \phi(v)[k^\star] - \max_{k \ne k^\star} \phi(v)[k],$$

  where $k^\star \in \mathrm{argmax}_{k \in [K]} \phi(v)[k]$.

- We also define $T_\varepsilon = \sum_{t=1}^{T} \mathbf{1}\{\texttt{Margin}(f^\star(x_t)) \le \varepsilon\}$ to denote the number of samples within $T$ rounds of interaction for which the margin w.r.t. $f^\star$ is smaller than $\varepsilon$.

- We define the function $\mathrm{Gap} : \mathbb{R}^K \times [K] \mapsto \mathbb{R}^+$ as

$$\mathrm{Gap}(v, k) = \max_{k'} \phi(v)[k'] - \phi(v)[k], \tag{11}$$

  to denote the gap between the largest and the $k$-th coordinate of $v$.

#### C.3.1. SUPPORTING TECHNICAL RESULTS

The following lemma is immediate.

**Lemma 9.** For any $u$ and $k' \ne \mathrm{argmax}_k \phi(u)[k]$,

$$\texttt{Margin}(u) \le \mathrm{Gap}(u, k').$$

*Proof.* Let $k^\star = \mathrm{argmax}_k \phi(u)[k]$. By definition,

$$\begin{aligned}
\mathrm{Gap}(u, k') &= \phi(u)[k^\star] - \phi(u)[k'] \\
&\ge \phi(u)[k^\star] - \max_{k' \ne k} \phi(u)[k'] = \texttt{Margin}(u).
\end{aligned}$$

$\square$

The following technical result establishes a certain favorable property for the function $f^\star$, whose proof follows from the regret bound of the online oracle used in Algorithm 1.

**Lemma 10.** With probability at least $1 - \delta$, the function $f^\star \in \mathcal{F}$ satisfies the following for all $t \le T$:

$$\sum_{s=1}^{t} Z_s \|f^\star(x_s) - f_s(x_s)\|^2 \le \Psi_\delta^{\ell_\phi}(\mathcal{F}, T),$$

where $\Psi_\delta^{\ell_\phi}(\mathcal{F}, T) := \frac{4}{\lambda} \mathrm{Reg}^{\ell_\phi}(\mathcal{F}; T) + \frac{112}{\lambda^2} \log(4 \log^2(T)/\delta)$.

*Proof.* The desired result follows from an application of Lemma 5, where we note that we do not query oracle when $Z_s = 0$, and thus do not count the time steps for which $Z_s = 0$. $\qquad\square$

Throughout the proof, we condition on the $1 - \delta$ probability event that Lemma 10 holds. The next technical lemma allows us to bound the number of times when we query for the label and $\Delta_t(x_t) \ge \zeta$ in terms of the eluder dimension (normed version) of the function class $\mathcal{F}$. Note that Lemma 11 holds even if the sequence $\{x_t\}_{t \le T}$ could be adversarially generated.

**Lemma 11.** Let $f^\star$ satisfy Lemma 10, and let $\Delta_t(x_t)$ be defined in (5) in Algorithm 1. Then, for any $\zeta > 0$, with probability at least $1 - \delta$,

$$\sum_{t=1}^{T} Z_t \mathbf{1}\{\Delta_t(x_t) \ge \zeta\} \le \widetilde{O}\left(\frac{\Psi_\delta^{\ell_\phi}(\mathcal{F}, T)}{\zeta^2} \cdot \mathfrak{E}(\mathcal{F}, \zeta/2; f^\star)\right).$$

where $\mathfrak{E}$ denotes the eluder dimension is given in Definition 1.

*Proof.* Let $f_t^\star$ denote the maximizer of (5) at round $t$ on point $x_t$. Thus,

$$\Delta_t(x_t) = \|f_t^\star(x_t) - f_t(x_t)\|, \qquad \text{and} \qquad \sum_{s=1}^{t-1} Z_s \|f_t^\star(x_s) - f_s(x_s)\|^2 \le \Psi_\delta^{\ell_\phi}(\mathcal{F}, T). \tag{12}$$

However, recall that Lemma 10 implies that, with probability at least $1 - \delta$, the function $f^\star$ satisfies the bound

$$\sum_{s=1}^{t-1} Z_s \|f^\star(x_s) - f_s(x_s)\|^2 \le \Psi_\delta^{\ell_\phi}(\mathcal{F}, T). \tag{13}$$

Using (12), (13) and Triangle inequality, we get that

$$\sum_{s=1}^{t-1} Z_s \|f_t^\star(x_s) - f^\star(x_s)\|^2 \le 2 \sum_{s=1}^{t-1} Z_s \|f_t^\star(x_t) - f_s(x_s)\|^2 + 2 \sum_{s=1}^{t-1} Z_s \|f^\star(x_t) - f_s(x_s)\|^2$$

$$\le 4 \Psi_\delta^{\ell_\phi}(\mathcal{F}, T). \tag{14}$$

Next, note that, an application of Triangle inequality implies that $\|f_t^\star(x_t) - f_t(x_t)\| \le \|f_t^\star(x_t) - f^\star(x_t)\| + \|f^\star(x_t) - f_t(x_t)\|$. Thus,

$$\sum_{t=1}^{T} Z_t \mathbf{1}\{\Delta_t(x_t) \ge \zeta\} = \sum_{t=1}^{T} Z_t \mathbf{1}\{\|f_t^\star(x_t) - f_t(x_t)\| \ge \zeta\}$$

$$\le \sum_{t=1}^{T} Z_t \mathbf{1}\{\|f_t^\star(x_t) - f^\star(x_t)\| + \|f^\star(x_t) - f_t(x_t)\| \ge \zeta\}$$

$$\le \sum_{t=1}^{T} Z_t \mathbf{1}\left\{\|f_t^\star(x_t) - f^\star(x_t)\| \ge \frac{\zeta}{2}\right\} + \sum_{t=1}^{T} Z_t \mathbf{1}\left\{\|f_t(x_t) - f^\star(x_t)\| \ge \frac{\zeta}{2}\right\}$$

$$\le \sum_{t=1}^{T} Z_t \mathbf{1}\left\{\|f_t^\star(x_t) - f^\star(x_t)\| \ge \frac{\zeta}{2}\right\} + \frac{4}{\zeta^2} \sum_{t=1}^{T} Z_t (f_t(x_t) - f^\star(x_t))^2$$

$$\le \sum_{t=1}^{T} Z_t \mathbf{1}\left\{\|f_t^\star(x_t) - f^\star(x_t)\| \ge \frac{\zeta}{2}\right\} + \frac{4 \Psi_\delta^{\ell_\phi}(\mathcal{F}, T)}{\zeta^2}, \tag{15}$$

where in the last line we used Lemma 10 to bound the second term. In the following, we show how to bound the first term. Recall that for any $t \leq T$, the function $f_t^\star$ satisfies (14). Thus, we wish to bound

$$\sum_{t=1}^{T} Z_t \mathbf{1}\left\{\|f_t^\star(x_t) - f^\star(x_t)\| \geq \frac{\varsigma}{2}\right\} \quad \text{s.t.} \quad \sum_{s=1}^{t-1} Z_s (f_t^\star(x_s) - f^\star(x_s))^2 \leq 4\Psi_\delta^{\ell_\phi}(\mathcal{F}, T),$$

for all $t \leq T$. An application of Lemma 8 in the above implies that

$$\sum_{t=1}^{T} Z_t \mathbf{1}\left\{\|f_t^\star(x_t) - f^\star(x_t)\| \geq \frac{\varsigma}{2}\right\} \leq \frac{17\Psi_\delta^{\ell_\phi}(\mathcal{F}, T)}{\varsigma^2} \cdot \mathfrak{E}(\mathcal{F}, \varsigma/2; f^\star). \tag{16}$$

where in the last line, we used the fact that $\Psi_\delta^{\ell_\phi}(\mathcal{F},T)/\varsigma^2 \geq 1$, for our parameter setting.

Plugging in the bound (16) in (15), and using the fact that $\mathfrak{E}(\mathcal{F}, \varsigma/2; f^\star) \geq 1$, we get that

$$\sum_{t=1}^{T} Z_t \mathbf{1}\{\Delta_t(x_t) \geq \varsigma\} \leq \frac{20\Psi_\delta^{\ell_\phi}(\mathcal{F}, T)}{\varsigma^2} \cdot \mathfrak{E}(\mathcal{F}, \varsigma/2; f^\star).$$

$\square$

The next two technical lemma's relate the margin to the gap between functions, and are useful in the analysis for regret / total number of queries.

**Lemma 12.** Suppose the functions $\pi_1$ and $\pi_2$ are defined such that $\pi_i(x) = \operatorname{argmax}_{k \in [K]} \phi(f_i(x))[k]$. Then, for any $x$ for which $\pi_1(x) \neq \pi_2(x)$, we have

$$\texttt{Margin}(f_1(x)) \leq \phi(f_1(x))[\pi_1(x)] - \phi(f_1(x))[\pi_2(x)] \leq 2\gamma\|f_1(x) - f_2(x)\|_2,$$

where $\gamma$-denotes the Lipschitz parameter of the link function $\phi$.

*Proof.* First note $\phi(f_2(x))[\pi_2(x)] \geq \phi(f_2(x))[\pi_1(x)]$ by the definition of $\pi_2$. Thus,

$$\begin{aligned}
\phi(f_1(x))&[\pi_1(x)] - \phi(f_1(x))[\pi_2(x)] \\
&\leq \phi(f_1(x))[\pi_1(x)] - \phi(f_2(x))[\pi_1(x)] + \phi(f_2(x))[\pi_2(x)] - \phi(f_1(x))[\pi_2(x)] \\
&\leq 2\|\phi(f_1(x)) - \phi(f_2(x))\|_\infty \\
&\leq 2\|\phi(f_1(x)) - \phi(f_2(x))\|_2.
\end{aligned}$$

Using the fact that $\phi$ is $\gamma$-Lipschitz, we immediately get that

$$\phi(f_1(x))[\pi_1(x)] - \phi(f_1(x))[\pi_2(x)] \leq 2\gamma\|f_1(x) - f_2(x)\|_2.$$

$\square$

**Lemma 13.** For any two function $f_1, f_2 \in \mathcal{F}$, and $x \in \mathcal{X}$,

$$\texttt{Margin}(f_1(x)) - \texttt{Margin}(f_2(x)) \leq 2\gamma\|f_1(x) - f_2(x)\|.$$

*Proof.* For the ease of notation, define

$$k_1 = \operatorname*{argmax}_{k \in [k]} \phi(f_1(x))[k] \qquad \text{and} \qquad k_1' = \operatorname*{argmax}_{k' \neq k_1} \phi(f_1(x))[k'],$$

where ties are broken arbitrarily but consistently. Similarly, we define

$$k_2 = \operatorname*{argmax}_{k \in [k]} \phi(f_2(x))[k] \qquad \text{and} \qquad k_2' = \operatorname*{argmax}_{k' \neq k_2} \phi(f_2(x))[k']. \tag{17}$$

Thus, we have that

$$\texttt{Margin}(f_1(x)) = \phi(f_1(x))[k_1] - \phi(f_1(x))[k_1'],$$

and

$$\texttt{Margin}(f_2(x)) = \phi(f_2(x))[k_2] - \phi(f_2(x))[k_2']. \tag{18}$$

Finally, also note that for any coordinate $k$,

$$\phi(f_1(x))[k] - \phi(f_2(x))[k] \le \|\phi(f_2(x)) - \phi(f_1(x))\|. \tag{19}$$

We now proceed with the proof. Plugging in the form in (18), we get that

$$\begin{aligned}
\texttt{Margin}&(f_1(x)) - \texttt{Margin}(f_2(x)) \\
&= \phi(f_1(x))[k_1] - \phi(f_1(x))[k_1'] - (\phi(f_2(x))[k_2] - \phi(f_2(x))[k_2']) \\
&= (\phi(f_1(x))[k_1] - \phi(f_2(x))[k_2]) + (\phi(f_2(x))[k_2'] - \phi(f_1(x))[k_1']) \\
&\le (\phi(f_1(x))[k_1] - \phi(f_2(x))[k_1]) + (\phi(f_2(x))[k_2'] - \phi(f_1(x))[k_1']) \\
&\le \|\phi(f_2(x)) - \phi(f_1(x))\| + (\phi(f_2(x))[k_2'] - \phi(f_1(x))[k_1']),
\end{aligned}$$

where the first inequality uses the fact that $k_2$ is the maximizer coordinate of $\phi(f_2(x))$ and the last inequality uses (19). In the following, we bound the second term in the right hand side above under the following three cases:

- *Case 1: $k_2' \ne k_1$:* Since $k_2' \ne k_1$, we note that replacing $k_1'$ by $k_2'$ in the second term will only increase the value (see the definition in (17)). Thus,

$$\begin{aligned}
\phi(f_2(x))[k_2'] - \phi(f_1(x))[k_1'] &\le \phi(f_2(x))[k_2'] - \phi(f_1(x))[k_2'] \\
&\le \|\phi(f_2(x)) - \phi(f_1(x))\|,
\end{aligned}$$

  where the last line uses (19).

- *Case 2a: $k_2' = k_1, k_2 = k_1'$:* Using definition of $k_2$ in (17), we note that

$$\begin{aligned}
\phi(f_2(x))[k_2'] - \phi(f_1(x))[k_1'] &= \phi(f_2(x))[k_2'] - \phi(f_1(x))[k_2] \\
&\le \phi(f_2(x))[k_2] - \phi(f_1(x))[k_2] \\
&\le \|\phi(f_2(x)) - \phi(f_1(x))\|,
\end{aligned}$$

  where the last line uses (19).

- *Case 2b: $k_2' = k_1, k_2 \ne k_1'$:* Using the fact that $k_2' = k_1$ and that $k_2 \ne k_2'$, we get that $k_2 \ne k_1$. Thus using the definition of $k_1'$ along with the fact that $k_2 \ne k_1$, we get that

$$\begin{aligned}
\phi(f_2(x))[k_2'] - \phi(f_1(x))[k_1'] &\le \phi(f_2(x))[k_2'] - \phi(f_1(x))[k_2] \\
&\le \phi(f_2(x))[k_2] - \phi(f_1(x))[k_2] \\
&\le \|\phi(f_2(x)) - \phi(f_1(x))\|,
\end{aligned}$$

  where the second last line uses definition of $k_2$ and the last line uses (19).

Combining all the above bounds together implies that

$$\texttt{Margin}(f_1(x)) - \texttt{Margin}(f_2(x)) \le 2\|\phi(f_2(x)) - \phi(f_1(x))\|.$$

The final statement follows since $\phi$ is $\gamma$-Lipschitz. $\qquad\square$

### C.3.2. REGRET BOUND

For the ease of notation, for the rest of the proof in this section we define the function $\pi^\star$ such that

$$\pi^\star(x) = \operatorname*{argmax}_k \phi(f^\star(x))[k].$$

Additionally, we recall that for any time $t$, $\widehat{y}_t = \texttt{SelectAction}(f_t(x_t)) = \operatorname{argmax}_k \phi(f_t(x))[k]$. Starting from the definition of the regret, we have

$$
\begin{aligned}
\operatorname{Reg}_T &= \sum_{t=1}^T \Pr(\widehat{y}_t \neq y_t) - \Pr(\pi^\star(x_t) \neq y_t) \\
&= \sum_{t=1}^T \mathbf{1}\{\widehat{y}_t \neq \pi^\star(x_t)\} \cdot |\Pr(y_t = \pi^\star(x_t)) - \Pr(y_t = \widehat{y}_t)| \\
&= \sum_{t=1}^T \mathbf{1}\{\widehat{y}_t \neq \pi^\star(x_t)\} \cdot |\phi(f^\star(x_t))[\pi^\star(x_t)] - \phi(f^\star(x_t))[\widehat{y}_t]| \\
&\leq \sum_{t=1}^T \mathbf{1}\{\widehat{y}_t \neq \pi^\star(x_t)\} \cdot \operatorname{Gap}(f^\star(x_t), \widehat{y}_t),
\end{aligned}
$$

where the second last line uses the probabilistic model from which labels are generated, and the last inequality plugs in the definition of $\operatorname{Gap}$ from (42). We can decompose the above regret bound further as:

$$
\operatorname{Reg}_T \leq \sum_{t=1}^T \mathbf{1}\{\widehat{y}_t \neq \pi^\star(x_t), \operatorname{Gap}(f^\star(x_t), \widehat{y}_t) \leq \varepsilon\} \cdot \operatorname{Gap}(f^\star(x_t), \widehat{y}_t)
$$

$$
+ \sum_{t=1}^T \mathbf{1}\{\widehat{y}_t \neq \pi^\star(x_t), \operatorname{Gap}(f^\star(x_t), \widehat{y}_t) > \varepsilon\} \cdot \operatorname{Gap}(f^\star(x_t), \widehat{y}_t)
$$

Using the fact that $\mathsf{y}_t(x_t) = \operatorname{argmax}_{k \in [K]} \phi(f_t(x))[k]$ along with the definition of $\operatorname{Gap}$ and Lemma 12, we get that

$$
\operatorname{Reg}_T \leq \sum_{t=1}^T \mathbf{1}\{\widehat{y}_t \neq \pi^\star(x_t), \operatorname{Gap}(f^\star(x_t), \widehat{y}_t) \leq \varepsilon\} \cdot \varepsilon
$$

$$
+ 2\gamma \sum_{t=1}^T \mathbf{1}\{\widehat{y}_t \neq \pi^\star(x_t), \operatorname{Gap}(f^\star(x_t), \widehat{y}_t) > \varepsilon\} \cdot \|f^\star(x_t) - f_t(x_t)\|
$$

$$
\leq \sum_{t=1}^T \mathbf{1}\{\texttt{Margin}(f^\star(x_t)) \leq \varepsilon\} \cdot \varepsilon
$$

$$
+ 2\gamma \sum_{t=1}^T \mathbf{1}\{\widehat{y}_t \neq \pi^\star(x_t), \operatorname{Gap}(f^\star(x_t), \widehat{y}_t) > \varepsilon\} \cdot \|f^\star(x_t) - f_t(x_t)\|
$$

$$
\leq \sum_{t=1}^T \mathbf{1}\{\texttt{Margin}(f^\star(x_t)) \leq \varepsilon\} \cdot \varepsilon
$$

$$
+ 2\gamma \sum_{t=1}^T Z_t \mathbf{1}\{\widehat{y}_t \neq \pi^\star(x_t), \operatorname{Gap}(f^\star(x_t), \widehat{y}_t) > \varepsilon\} \cdot \|f^\star(x_t) - f_t(x_t)\|
$$

$$
+ 2\gamma \sum_{t=1}^T \bar{Z}_t \mathbf{1}\{\widehat{y}_t \neq \pi^\star(x_t)\} \cdot \|f^\star(x_t) - f_t(x_t)\| \tag{20}
$$

$$
= T_\varepsilon \cdot \varepsilon + 2\gamma \cdot T_A + 2\gamma \cdot T_B \cdot \|f^\star(x_t) - f_t(x_t)\|,
$$

where the second inequality holds because $\operatorname{Gap}(f^\star(x_t), \widehat{y}_t) \leq \varepsilon$ implies that $\texttt{Margin}(f^\star(x_t)) \leq \varepsilon$ whenever $\widehat{y}_t \neq \pi^\star(x_t)$. In the last line above, we plugged in the definition of $T_\varepsilon$, and defined $T_A$ and $T_B$ as the second term and the last term respectively (upto constants). We bound them separately below:

- *Bound on $T_A$:* We note that

$$
T_A = \sum_{t=1}^T Z_t \mathbf{1}\{\widehat{y}_t \neq \pi^\star(x_t), \operatorname{Gap}(f^\star(x_t), \widehat{y}_t) > \varepsilon\} \cdot \|f^\star(x_t) - f_t(x_t)\|
$$

$$\leq \sum_{t=1}^{T} Z_t \mathbf{1}\{\|f^\star(x_t) - f_t(x_t)\| > \varepsilon/2\gamma\} \cdot \|f^\star(x_t) - f_t(x_t)\|$$

where the second line follows from Lemma 12 and because $\widehat{y}_t \neq \pi^\star(x_t)$. Using the fact that $\mathbf{1}\{a \geq b\} \leq a/b$ for all $a, b \geq 0$, we get that

$$\mathsf{T}_A \leq 4\gamma \sum_{t=1}^{T} Z_t \frac{\|f^\star(x_t) - f_t(x_t)\|^2}{\varepsilon}. \tag{21}$$

- *Bound on* $\mathsf{T}_B$: Fix any $t \leq T$, and note that Lemma 10 implies that $\sum_{s=1}^{t} \|f^\star(x_t) - f_t(x_t)\|^2 \leq \Psi_\delta^{\ell_\phi}(\mathcal{F}, T)$. Thus $f^\star$ satisfies the constraint in the definition of $\Delta_t$ in (5) and we must have that

$$\|f^\star(x_t) - f_t(x_t)\| \leq \Delta_t(x_t). \tag{22}$$

Plugging in the definition of $Z_t$, we note that

$$\mathsf{T}_B = \sum_{t=1}^{T} \mathbf{1}\{\mathrm{Margin}(f_t(x_t)) > 2\gamma\Delta_t(x_t), \widehat{y}_t \neq \pi^\star(x_t)\}$$

$$\leq \sum_{t=1}^{T} \mathbf{1}\{\|f_t(x_t) - f^\star(x_t)\| > \Delta_t(x_t)\},$$

where the second inequality is due Lemma 12. However, note that the term inside the indicator contradicts (22) (which always holds). Thus,

$$\mathsf{T}_B = 0. \tag{23}$$

Combining the bounds (21) and (23), we get that

$$\mathrm{Reg}_T \leq \varepsilon T_\varepsilon + 8\gamma^2 \sum_{t=1}^{T} Z_t \frac{\|f_t(x_t) - f^\star(x_t)\|^2}{\varepsilon}$$

$$\leq \varepsilon T_\varepsilon + \frac{8\gamma^2}{\varepsilon} \Psi_\delta^{\ell_\phi}(\mathcal{F}, T),$$

where the last inequality is due to Lemma 10.

Since $\varepsilon$ is a free parameter above, the final bound follows by choosing the best parameter $\varepsilon$, and by plugging in the form of $\Psi_\delta^{\ell_\phi}(\mathcal{F}, T)$.

### C.3.3. TOTAL NUMBER OF QUERIES

We use the notation $N_T$ to denote the total number of expert queries made by the learner within $T$ rounds of interactions. Using the definition of $Z_t$, we have that

$$N_T = \sum_{t=1}^{T} Z_t$$

$$= \sum_{t=1}^{T} \mathbf{1}\{\mathrm{Margin}(f_t(x_t)) \leq 2\gamma\Delta_t(x_t)\}$$

$$= \sum_{t=1}^{T} \mathbf{1}\{\mathrm{Margin}(f_t(x_t)) \leq 2\gamma\Delta_t(x_t), \mathrm{Margin}(f^\star(x_t)) \leq \varepsilon\}$$

$$+ \sum_{t=1}^{T} \mathbf{1}\{\mathrm{Margin}(f_t(x_t)) \leq 2\gamma\Delta_t(x_t), \mathrm{Margin}(f^\star(x_t)) > \varepsilon\}$$

$$\leq \sum_{t=1}^{T} \mathbf{1}\{\mathrm{Margin}(f^\star(x_t)) \leq \varepsilon\}$$

$$+ \sum_{t=1}^{T} \mathbf{1}\{\texttt{Margin}(f_t(x_t)) \le 2\gamma\Delta_t(x_t), \texttt{Margin}(f^\star(x_t)) > \varepsilon, \Delta_t(x_t) \le \varepsilon/4\gamma\}$$

$$+ \sum_{t=1}^{T} \mathbf{1}\{\texttt{Margin}(f_t(x_t)) \le 2\gamma\Delta_t(x_t), \texttt{Margin}(f^\star(x_t)) > \varepsilon, \Delta_t(x_t) > \varepsilon/4\gamma\} \qquad (24)$$

$$= T_\varepsilon + \texttt{T}_D + \texttt{T}_E,$$

where in the last line we use the definition of $T_\varepsilon$, and defined $\texttt{T}_D$ and $\texttt{T}_E$ respectively. We bound them separately below:

- *Bound on the term* $\texttt{T}_D$. Recall (22) which implies that $f^\star$ satisfies the bound $\|f_t(x_t) - f^\star(x_t)\| \le \Delta_t(x_t)$. Thus, using Lemma 13, we get that

$$\texttt{Margin}(f^\star(x_t)) \le 2\gamma\|f_t(x_t) - f^\star(x_t)\| + t\texttt{Margin}(f_t(x_t)) \le 2\gamma\Delta_t(x_t) + \texttt{Margin}(f_t(x_t)).$$

The above implies that

$$\texttt{T}_D = \sum_{t=1}^{T} \mathbf{1}\{\texttt{Margin}(f_t(x_t)) \le 2\gamma\Delta_t(x_t), \texttt{Margin}(f^\star(x_t)) > \varepsilon, \Delta_t(x_t) \le \varepsilon/4\gamma\}$$

$$\le \sum_{t=1}^{T} \mathbf{1}\{\texttt{Margin}(f^\star(x_t)) \le 4\gamma\Delta_t(x_t), \texttt{Margin}(f^\star(x_t)) > \varepsilon, \Delta_t(x_t) \le \varepsilon/4\gamma\}$$

$$\le 0,$$

where the last line follows from the fact that all the conditions inside the indictor can not hold simultaneously.

- *Bound on the term* $\texttt{T}_E$. We note that

$$\texttt{T}_E = \sum_{t=1}^{T} \mathbf{1}\{\texttt{Margin}(f_t(x_t)) \le 2\gamma\Delta_t(x_t), \texttt{Margin}(f^\star(x_t)) > \varepsilon, \Delta_t(x_t) > \varepsilon/4\gamma\}$$

$$\le \sum_{t=1}^{T} Z_t \mathbf{1}\{\Delta_t(x_t) \ge \varepsilon/4\gamma\}$$

$$\le \frac{320\Psi_\delta^{\ell_\phi}(\mathcal{F}, T)}{\varepsilon^2} \cdot \mathfrak{E}(\mathcal{F}, \varepsilon/4\gamma; f^\star).$$

where the last line follows from setting $\zeta = \varepsilon/4\gamma$ in Lemma 11.

Gathering the bounds above, we get that

$$N_T \le T_\varepsilon + \frac{640\gamma^2\Psi_\delta^{\ell_\phi}(\mathcal{F}, T)}{\varepsilon^2} \cdot \mathfrak{E}(\mathcal{F}, \varepsilon/4\gamma; f^\star).$$

Since $\varepsilon$ is a free parameter above, the final bound follows by choosing the best parameter $\varepsilon$, and by plugging in the form of $\Psi_\delta^{\ell_\phi}(\mathcal{F}, T)$.

### C.4. Proof of Theorem 2

Before delving into the proof, we recall the relevant notation. In Algorithm 3,

- The label $y_t \sim \phi(f^\star(x_t))$, where $\phi$ denotes the link-function given in (2).

- The function $\texttt{SelectAction}(f_t(x_t)) \coloneqq \text{argmax}_k \phi(f_t(x_t))[k]$.

- For any vector $v \in \mathbb{R}^K$, the margin is given by the gap between the value at the largest and the second largest coordinate, i.e.

$$\texttt{Margin}(v) = \phi(v)[k^\star] - \max_{k \ne k^\star} \phi(v)[k],$$

where $k^\star \in \text{argmax}_{k \in [K]} \phi(v)[k]$.

---

**Algorithm 3** Selective Sampling for Stochastic Setting, Action Set $\mathcal{A} = [K]$

---

**Require:** Parameters $\delta, \gamma, \lambda, T$, function class $\mathcal{F}$, and online regression oracle Oracle w.r.t $\ell_\phi$.

1: Set $\Psi_\delta^{\ell_\phi}(\mathcal{F}, T) = \frac{4}{\lambda} \mathrm{Reg}^{\ell_\phi}(\mathcal{F}; T) + \frac{112}{\lambda^2} \log(4 \log^2(T)/\delta)$, Compute $f_1 \leftarrow \mathrm{Oracle}_1(\varnothing)$.

2: Set $E = \lceil \log(T) \rceil$ and $\tau_e = 2^{e-1}$ for $e \le E$.

3: **for** $e = 1, \ldots, E - 1$ **do**

4:     Learner constructs the feasible set of optimal functions $\mathcal{F}_e$ as

$$\mathcal{F}_e = \left\{ f \in \mathcal{F} \mid \sum_{s=1}^{\tau_e - 1} Z_s \| f(x_s) - f_s(x_s) \|^2 \le \Psi_\delta^{\ell_\phi}(\mathcal{F}, T) \right\}. \tag{25}$$

5:     **for** $t \leftarrow \tau_e$ to $\tau_{e+1} - 1$ **do**

6:         Nature samples $x_t$ from an (unknown) distribution $\mu$.

7:         Learner computes

$$g_t \in \underset{g \in \mathcal{F}_e}{\mathrm{argmin}} \ \mathrm{Margin}(g(x_t)), \quad \text{and,} \quad \Delta_e(x_t) := \max_{f, f' \in \mathcal{F}_e} \| f(x_t) - f'(x_t) \|. \tag{26}$$

8:         Learner decides whether to query: $Z_t = \mathbf{1}\{\mathrm{Margin}(g_t(x_t)) \le 2\gamma \Delta_e(x_t)\}$.

9:         **if** $Z_t = 1$ **then**

10:             Learner plays the action $\widehat{y}_t = \mathrm{SelectAction}(f_t(x_t))$.

11:             Learner queries the label $y_t$ on $x_t$.

12:             $f_{t+1} \leftarrow \mathrm{Oracle}_t(\{x_t, y_t\})$.

13:         **else**

14:             Learner plays the action $\widehat{y}_t = \mathrm{SelectAction}(g_t(x_t))$.

15:             $f_{t+1} \leftarrow f_t$.

16:         **end if**

17:     **end for**

18: **end for**

---

- We also define $T_\varepsilon = \sum_{t=1}^T \mathbf{1}\{\mathrm{Margin}(f^\star(x_t)) \le \varepsilon\}$ to denote the number of samples within $T$ rounds of interaction for which the margin w.r.t. $f^\star$ is smaller than $\varepsilon$.

- We define the function $\mathrm{Gap} : \mathbb{R}^K \times [K] \mapsto \mathbb{R}^+$ as

$$\mathrm{Gap}(v, k) = \max_{k'} \phi(v)[k'] - \phi(v)[k], \tag{27}$$

  to denote the gap between the largest and the $k$-th coordinate of $v$. Recall that Lemma 9 holds.

- Additionally, we define the function $Z^e$ to denote the query condition

$$Z^e(x) = \mathbf{1}\{\inf_{g \in \mathcal{F}_e} \mathrm{Margin}(g(x)) \le 2\gamma \sup_{f, f' \in \mathcal{F}_e} \| f(x) - f'(x) \|\}. \tag{28}$$

  The definition in (28) suggests that for all $t \in [\tau_e, \tau_{e+1})$, $Z_t = Z^e(x_t)$, for all $e \le E - 1$.

**Intuition for epoching.** We next provide intuition on why epoching in needed in Algorithm 3 to get the improved query complexity bound. From the proof sketch in Section C.2, the term $\sum_{t=1}^T Z_t \mathbf{1}\{\Delta_t(x_t) \ge \varepsilon\}$ appearing in the query complexity bound is handled using the eluder dimension of $\mathcal{F}$. When $x_t$ is sampled i.i.d. we wish to bound this using disagreement-coefficient instead. However, note that in Algorithm 1 the query condition $Z_t$ depends on the samples $\{x_s\}_{s<t}$ drawn in all previous time steps and the corresponding query conditions $\{Z_s\}_{s<t}$. This introduces a bias, and thus the terms $Z_t \mathbf{1}\{\Delta_t(x_t) \ge \varepsilon\}$ are no longer independent to each other. Thus, we can not directly used distributional properties like the disagreement coefficient to bound the query complexity. Algorithm 3 fixes this issue by defining epochs of doubling length such that the query condition in epoch $e$ only depends on the samples presented to the learner at time steps before this epoch (i.e. in time steps $1 \le t \le \tau_e - 1$). Thus, the terms $Z_t \mathbf{1}\{\Delta_t(x_t) \ge \varepsilon\}$ for $\tau_e \le t < \tau_{e+1}$ are i.i.d. allowing us to get bounds in terms of distributional properties like the disagreement coefficient of $\mathcal{F}$.

However, note that whenever we query in Algorithm 3, we still choose the labels according to the estimate from the online regression oracle and thus the regret bound remains unchanged.

C.4.1. SUPPORTING TECHNICAL RESULTS

The following lemma establishes useful technical properties of the function $f^\star$ and the sets $\mathcal{F}_e$.

**Lemma 14.** *Suppose* Algorithm 3 *is run on the sequence* $\{x_t\}_{t \leq T}$ *drawn i.i.d. from the unknown distribution* $\mu$. *Then, with probability at least* $1 - \delta$, *each of the following holds:*

$(a)$ *For all* $t \leq T$, *the function* $f^\star \in \mathcal{F}$ *satisfies*

$$\sum_{s=1}^{t} Z_s \|f^\star(x_s) - f_s(x_s)\|^2 \leq \Psi_\delta^{\ell_\phi}(\mathcal{F}, T),$$

*where* $\Psi_\delta^{\ell_\phi}(\mathcal{F}, T) = \frac{4}{\lambda}\mathrm{Reg}^{\ell_\phi}(\mathcal{F}; T) + \frac{112}{\lambda^2}\log(4\log^2(T)/\delta)$.

*Thus,* $f^\star \in \mathcal{F}_e$ *for all* $e \leq E - 1$, *and* $\|f^\star(x_t) - g_t(x_t)\| \leq \Delta_e(x_t)$ *for all* $\tau_e \leq t \leq \tau_{e+1} - 1$.

$(b)$ *For any function* $f \in \mathcal{F}_e$, *we have*

$$\mathbb{E}\left[\sum_{s=1}^{\tau_e - 1} Z_s \|f(x_s) - f^\star(x_s)\|^2\right] \leq \widehat{\Psi}_\delta^{\ell_\phi}(\mathcal{F}; T),$$

*where* $\widehat{\Psi}_\delta^{\ell_\phi}(\mathcal{F}; T) := 2\Psi_\delta^{\ell_\phi}(\mathcal{F}, T) + 4c_2\left(\log^4(T)\sup_{\tau \leq T}\left(\tau \mathcal{R}_\tau^2(\mathcal{F})\right) + 2\log(T)\log(E/\delta)\right)$.

$(c)$ *For any* $e \leq E$, *and any function* $f \in \mathcal{F}_e$, *we have*

$$\|f(x_s) - f^\star(x_s)\|_{\bar{\nu}_e} \leq \sqrt{\frac{\widehat{\Psi}_\delta^{\ell_\phi}(\mathcal{F}; T)}{\tau_e - 1}}$$

*where the sub-distributions* $\bar{\mu}_{\bar{e}}(x) := Z^{\bar{e}}(x)\mu(x)$ *and* $\bar{\nu}_e := \frac{1}{\tau_e - \tau_1}\sum_{\bar{e}=1}^{e-1}(\tau_{\bar{e}+1} - \tau_{\bar{e}})\bar{\mu}_{\bar{e}}$.

$(d)$ *For any* $\bar{e} < e$, *the corresponding sets* $\mathcal{F}_e$ *and* $\mathcal{F}_{\bar{e}}$ *satisfy the relation* $\mathcal{F}_e \subseteq \mathcal{F}_{\bar{e}}$.

$(e)$ *For any* $\bar{e} \leq e$, *we have* $\bar{\mu}_e \preccurlyeq \bar{\mu}_{\bar{e}}$.

*Proof.* We prove each part separately below:

$(a)$ An application of Lemma 5, where we note that we do not query oracle when $Z_s = 0$, and thus do not count the time steps for which $Z_s = 0$, implies that

$$\sum_{s=1}^{t} Z_s \|f^\star(x_s) - f_s(x_s)\|^2 \leq \frac{4}{\lambda}\mathrm{Reg}^{\ell_\phi}(\mathcal{F}; T) + \frac{112}{\lambda^2}\log(4\log^2(T)/\delta) =: \Psi_\delta^{\ell_\phi}(\mathcal{F}, T)$$

for all $t \leq T$ with probability at least $1 - \delta$. Using the above for $t = \tau_{e+1} - 1$ implies that $f^\star \in \mathcal{F}_e$ for all $e \leq E - 1$. Since, we also have that $g_t \in \mathcal{F}_e$ (by construction) for all $\tau_e \leq t \leq \tau_{e+1} - 1$, plugging in the definition of $\Delta_e(x_t)$, we immediately get that $\|f^\star(x_t) - g_t(x_t)\| \leq \Delta_e(x_t)$.

$(b)$ Fix any epoch number $\bar{e} \leq E - 1$, and consider the time steps $\tau_{\bar{e}} \leq t < \tau_{\bar{e}+1}$. Define the loss function

$$\ell_{\bar{e}}(f(x), f^\star(x)) = Z^{\bar{e}}(x)\|f(x) - f^\star(x)\|^2$$

where $Z^{\bar{e}}$ denotes the query conditions at epoch $\bar{e}$ (defined in (28)), and recall that $Z^{\bar{e}}$ does not depend on any samples that are drawn at epoch $\bar{e}$ (by definition). Furthermore, note that $\ell_{\bar{e}}$ is 2-smooth w.r.t. $f$ and satisfies $\ell_{\bar{e}}(f(x), f^\star(x)) \leq 4$ for all $f, f^\star \in \mathcal{F}$ and $x$. Thus using Lemma 2, we get that for any $f \in \mathcal{F}_e$, with probability at least $1 - \delta/E$,

$$\mathbb{E}\left[\sum_{s=\tau_{\bar{e}}}^{\tau_{\bar{e}+1}-1} Z^{\bar{e}}(x_s)\|f(x_s) - f^\star(x_s)\|^2\right]$$

$$\leq 2\sum_{s=\tau_{\bar{e}}}^{\tau_{\bar{e}+1}-1} Z^{\bar{e}}(x_s)\|f(x_s) - f^\star(x_s)\|^2 + c_2\left(2\tau_e \log^3(\tau_e)\mathcal{R}_{\tau_e}^2(\mathcal{F}_e) + 4\log(E/\delta)\right)$$

$$\leq 2 \sum_{s=\tau_{\bar{e}}}^{\tau_{\bar{e}+1}-1} Z^{\bar{e}}(x_s)\|f(x_s) - f^\star(x_s)\|^2 + 2c_2 \left( \log^3(T) \sup_{\tau \leq T} \left( \tau \mathcal{R}_\tau^2(\mathcal{F}) \right) + 2\log(E/\delta) \right),$$

where in the last line we used the fact that $\mathcal{F}_e \subseteq \mathcal{F}$. Summing the above for all $\bar{e} \leq e - 1$, we get that for any $f \in \mathcal{F}_e$,

$$\mathbb{E}\left[ \sum_{t=1}^{\tau_e - 1} Z^e(x_s)\|f(x_s) - f^\star(x_s)\|^2 \right]$$

$$\leq 2 \sum_{t=1}^{\tau_e - 1} Z^e(x_s)\|f(x_s) - f^\star(x_s)\|^2 + 4c_2 E\left( \log^3(T) \sup_{\tau \leq T} \left( \tau \mathcal{R}_\tau^2(\mathcal{F}) \right) + 2\log(E/\delta) \right)$$

$$\leq 2\Psi_\delta^{\ell_\phi}(\mathcal{F}, T) + 4c_2 \left( \log^4(T) \sup_{\tau \leq T} \left( \tau \mathcal{R}_\tau^2(\mathcal{F}) \right) + 2\log(T)\log(E/\delta) \right),$$

where the last line follows by using the definition of $\mathcal{F}_e$, and that $E \leq \lceil \log(T) \rceil$, and that $\mathcal{F}_e \subseteq \mathcal{F}$.

$(c)$ Starting from part-$(b)$, we first note that

$$\mathbb{E}\left[ \sum_{s=1}^{\tau_e - 1} Z_s \|f(x_s) - f^\star(x_s)\|^2 \right] \leq \widehat{\Psi}_\delta^{\ell_\phi}(\mathcal{F}; T). \tag{29}$$

Additionally, also note that

$$\mathbb{E}\left[ \sum_{s=1}^{\tau_e - 1} Z_s \|f(x_s) - f^\star(x_s)\|^2 \right] = \mathbb{E}\left[ \sum_{\bar{e}=1}^{e-1} \sum_{s=\tau_{\bar{e}}}^{\tau_{\bar{e}+1}-1} Z_s \|f(x_s) - f^\star(x_s)\|^2 \right]$$

$$= \mathbb{E}\left[ \sum_{\bar{e}=1}^{e-1} \sum_{s=\tau_{\bar{e}}}^{\tau_{\bar{e}+1}-1} Z^{\bar{e}}(x_s) \|f(x_s) - f^\star(x_s)\|^2 \right]$$

$$= \sum_{\bar{e}=1}^{e-1} \sum_{s=\tau_{\bar{e}}}^{\tau_{\bar{e}+1}-1} \mathbb{E}_{x_s \sim \mu}\left[ Z^{\bar{e}}(x_s) \|f(x_s) - f^\star(x_s)\|^2 \right]$$

$$= \sum_{\bar{e}=1}^{e-1} (\tau_{\bar{e}+1} - \tau_{\bar{e}}) \, \mathbb{E}_{x \sim \mu}\left[ Z^{\bar{e}}(x) \|f(x) - f^\star(x)\|^2 \right],$$

where the last two lines use the fact that the query condition $Z^{\bar{e}}$ does not depend on the samples from rounds $t = \tau_{\bar{e}}$ to $\tau_{\bar{e}+1} - 1$. Plugging in the definition of $\bar{\mu}_{\bar{e}}$ in the above, we get that

$$\mathbb{E}\left[ \sum_{s=1}^{\tau_e - 1} Z_s \|f(x_s) - f^\star(x_s)\|^2 \right] = \sum_{\bar{e}=1}^{e-1} (\tau_{\bar{e}+1} - \tau_{\bar{e}}) \cdot \mathbb{E}_{x \sim \bar{\mu}_{\bar{e}}}\left[ \|f(x_s) - f^\star(x_s)\|^2 \right]$$

$$= (\tau_e - \tau_1) \cdot \mathbb{E}_{x \sim \bar{\nu}_e}\left[ \|f(x_s) - f^\star(x_s)\|^2 \right]$$

$$= (\tau_e - \tau_1) \cdot \|f(x_s) - f^\star(x_s)\|_{\bar{\nu}_e}^2, \tag{30}$$

where in the second line, we used the fact that the sub-distribution $\bar{\nu}_e := \frac{1}{\tau_e - \tau_1} \sum_{\bar{e}=1}^{e-1} (\tau_{\bar{e}+1} - \tau_{\bar{e}}) \bar{\mu}_{\bar{e}}$.

Combining (29) and (30), we get that

$$\|f(x_s) - f^\star(x_s)\|_{\bar{\nu}_e} \leq \sqrt{\frac{\widehat{\Psi}_\delta^{\ell_\phi}(\mathcal{F}; T)}{\tau_e - 1}}$$

$(d)$ The argument follows from the definition of the set $\mathcal{F}_e$ as any function $f \in \mathcal{F}_e$ that satisfies

$$\sum_{s=1}^{\tau_e - 1} Z_s \|f(x_s) - f_s(x_s)\|^2 \leq \Psi_\delta^{\ell_\phi}(\mathcal{F}, T),$$

also satisfies the constraint

$$\sum_{s=1}^{\tau_{\bar{e}} - 1} Z_s \|f(x_s) - f_s(x_s)\|^2 \leq \Psi_\delta^{\ell_\phi}(\mathcal{F}, T),$$

for any $\bar{e} \leq e$, since the left hand side consists of lesser number of terms and all terms are non-negative. Thus, $\mathcal{F}_e \subseteq \mathcal{F}_{\bar{e}}$.

(e) Recall that for any $e \leq E - 1$, the sub-probability measure $\bar{\mu}_e(x) := Z^e(x)\mu(x)$ where $Z^e(x) = \mathbf{1}\{\min_{g \in \mathcal{F}_e} \|g(x)\| \leq \Delta_e(x)\}$, and $\Delta_e(x) = \max_{f', f \in \mathcal{F}_e} \|f(x) - f'(x)\|$. First note that for any $\bar{e} \leq e$,

$$\Delta_e(x) = \max_{f', f \in \mathcal{F}_e} \|f(x) - f'(x)\| \leq \max_{f', f \in \mathcal{F}_{\bar{e}}} \|f(x) - f'(x)\| = \Delta_{\bar{e}}(x),$$

where the inequality above holds because $\mathcal{F}_e \subseteq \mathcal{F}_{\bar{e}}$ due to part-(d) above. Furthermore,

$$\min_{g \in \mathcal{F}_e} \|g(x)\| \geq \min_{g \in \mathcal{F}_{\bar{e}}} \|g(x)\|,$$

again because $\mathcal{F}_e \subseteq \mathcal{F}_{\bar{e}}$. Thus,

$$Z^e(x) = \mathbf{1}\{\min_{g \in \mathcal{F}_e} \|g(x)\| \leq \Delta_e(x)\} \leq \mathbf{1}\{\min_{g \in \mathcal{F}_{\bar{e}}} \|g(x)\| \leq \Delta_{\bar{e}}(x)\} \leq Z^{\bar{e}}(x).$$

The above implies that $\bar{\mu}_e \leq \bar{\mu}_{\bar{e}}$.

$\square$

**Lemma 15.** Let $\varepsilon_0, \gamma_0 \geq 0$, and $f^\star \in \mathcal{F}$. Then, for any sub distribution $\bar{\mu}$ such that $\mathbb{E}_{x \sim \bar{\mu}}[\mathbf{1}\{x \in \mathcal{X}\}] > 0$, $\varepsilon \geq \varepsilon_0$ and $\gamma \geq \gamma_0$,

$$\frac{\varepsilon^2}{\gamma^2} \Pr_{x \sim \bar{\mu}}\left(\exists f \in \mathcal{F} : \|f(x) - f^\star(x)\| > \varepsilon, \|f(x_s) - f^\star(x_s)\|_{\bar{\mu}} \leq \gamma\right) \leq \theta^{\mathrm{val}}(\mathcal{F}, \varepsilon_0, \gamma_0; f^\star).$$

*Proof.* The key idea in the proof is to go from sub distributions to distributions, and then invoking the definition of $\theta$ from Definition 2. Define $\kappa = \mathbb{E}_{x \sim \bar{\mu}}[\mathbf{1}\{x \in \mathcal{X}\}]$. Since $0 < \kappa \leq 1$, we can define a probability measure $\mu$ such that $\mu(x) = \bar{\mu}(x)/\kappa$. Thus, for any $\varepsilon \geq \varepsilon_0$ and $\gamma \geq \gamma_0$,

$$\frac{\varepsilon^2}{\gamma^2} \Pr_{x \sim \bar{\mu}}\left(\exists f \in \mathcal{F} : \|f(x) - f^\star(x)\| > \varepsilon, \|f(x_s) - f^\star(x_s)\|_{\bar{\mu}} \leq \gamma\right)$$

$$= \frac{\varepsilon^2}{\gamma^2/k} \Pr_{x \sim \mu}\left(\exists f \in \mathcal{F} : \|f(x) - f^\star(x)\| > \varepsilon, \|f(x_s) - f^\star(x_s)\|_{\bar{\mu}} \leq \gamma\right)$$

$$\leq \frac{\varepsilon^2}{\gamma^2/k} \Pr_{x \sim \mu}\left(\exists f \in \mathcal{F} : \|f(x) - f^\star(x)\| > \varepsilon, \|f(x_s) - f^\star(x_s)\|_{\mu} \leq \gamma/\sqrt{k}\right)$$

$$= \frac{\varepsilon^2}{\bar{\gamma}^2} \Pr_{x \sim \mu}\left(\exists f \in \mathcal{F} : \|f(x) - f^\star(x)\| > \varepsilon, \|f(x_s) - f^\star(x_s)\|_{\mu} \leq \bar{\gamma}\right)$$

$$\leq \sup_{\varepsilon \geq \varepsilon_0, \bar{\gamma} \geq \gamma_0} \frac{\varepsilon^2}{\bar{\gamma}^2} \Pr_{x \sim \mu}\left(\exists f \in \mathcal{F} : \|f(x) - f^\star(x)\| > \varepsilon, \|f(x_s) - f^\star(x_s)\|_{\mu} \leq \bar{\gamma}\right),$$

where in the second last line, we defined $\bar{\gamma} = \gamma/\sqrt{k}$, and the last line used the fact that both $\varepsilon \geq \varepsilon_0$ and $\bar{\gamma} \geq \gamma_0$. The final statement follows by noting the fact that $\mu$ is a distribution and the definition of the disagreement coefficient $\theta^{\mathrm{val}}(;)$ from Definition 2. $\square$

The following technical result will be useful in bounding the query complexity for Algorithm 3.

**Lemma 16.** For any $t \leq T$, let $e(t)$ denotes the epoch number such that $\tau_{e(t)} \leq t < \tau_{e(t)+1}$. Let $f^\star \in \mathcal{F}$ satisfy Lemma 14, and let $\Delta_{e(t)}(x_t)$ be defined in Algorithm 3. Then, for any $\zeta > 0$, with probability at least $1 - \delta$,

$$\sum_{t=1}^{T} Z_t \mathbf{1}\{\Delta_{e(t)}(x_t) \geq \zeta\} \leq 12 \log(T) \cdot \frac{\widehat{\Psi}_\delta^{\ell_\phi}(\mathcal{F}; T)}{\zeta^2} \cdot \theta^{\mathrm{val}}\left(\mathcal{F}, \frac{\zeta}{2}, \frac{\widehat{\Psi}_\delta^{\ell_\phi}(\mathcal{F}; T)}{T}; f^\star\right) + 4 \log(2/\delta).$$

*Proof.* Recall the definition of the query rule $Z^e$ given in (28), and note that the function $Z^e$ is independent of the samples $\{x_t\}_{t=\tau_e}^{\tau_{e+1}-1}$ chosen by the nature for time steps at epoch $e$. Additionally, also recall that at every time step, $x_t$ is sampled independently from the distribution $\mu$. Thus, using the query condition $Z^e$, we can define the sub-probability measure

$$\bar{\mu}_e := \mu(x)Z^e(x), \tag{31}$$

such that $\bar{\mu}_e(x) = \mu(x)$ whenever $Z_e(x) = 1$ and is 0 otherwise. Furthermore, for any $e \in E-1$, we define the sub-probability measure $\bar{\nu}_e$ as

$$\bar{\nu}_e = \frac{1}{\tau_e - \tau_1} \sum_{\bar{e}=1}^{e-1} (\tau_{\bar{e}+1} - \tau_{\bar{e}}) \bar{\mu}_{\bar{e}}. \tag{32}$$

We now move to the main proof. First fix any epoch $e \leq E - 1$, and consider any round $t \in [\tau_e, \tau_{e+1} - 1]$. Using the definition of $\Delta_e(x_t)$ and definition of $Z^e$ from (28) in the above, we get that

$$\mathbb{E}_{x_t \sim \mu}[Z_t \mathbf{1}\{\Delta_e(x_t) > \zeta\}] = \mathbb{E}_{x \sim \mu}\left[Z^e(x_t)\mathbf{1}\left\{\sup_{f,f' \in \mathcal{F}_e} \|f(x_t) - f'(x_t)\| > \zeta\right\}\right]$$

$$\leq \mathbb{E}_{x \sim \mu}\left[Z^e(x_t)\mathbf{1}\left\{\sup_{f \in \mathcal{F}_e} \|f(x_t) - f^\star(x_t)\| > \frac{\zeta}{2}\right\}\right]$$

where the second line follows because $f^\star \in \mathcal{F}_e$, and because $\sup_{f,f' \in \mathcal{F}_e} \|f(x_t) - f'(x_t)\| \leq 2 \sup_{f \in \mathcal{F}_e} \|f(x_t) - f^\star(x_t)\|$ due to Triangle inequality. Plugging in the definition of $\bar{\mu}_e$ from (31) in the above we get that

$$\mathbb{E}_{x_t \sim \mu}[Z_t \mathbf{1}\{\Delta_e(x_t) > \zeta\}] \leq \mathbb{E}_{x \sim \bar{\mu}_e}\left[\mathbf{1}\left\{\sup_{f \in \mathcal{F}_e} \|f(x_t) - f^\star(x_t)\| > \frac{\zeta}{2}\right\}\right]$$

$$\leq \mathbb{E}_{x \sim \bar{\mu}_{\bar{e}}}\left[\mathbf{1}\left\{\sup_{f \in \mathcal{F}_e} \|f(x_t) - f^\star(x_t)\| > \frac{\zeta}{2}\right\}\right]. \tag{33}$$

for all $\bar{e} \leq e$, where the last inequality follows from Lemma 14-$(e)$. Since the above holds for all $\bar{e} \leq e$, we immediately get that

$$\mathbb{E}_{x_t \sim \mu}[Z_t \mathbf{1}\{\Delta_e(x_t) > \zeta\}] \leq \mathbb{E}_{x \sim \bar{\nu}_{\bar{e}}}\left[\mathbf{1}\left\{\sup_{f \in \mathcal{F}_e} \|f(x_t) - f^\star(x_t)\| > \frac{\zeta}{2}\right\}\right]$$

$$= \mathbb{E}_{x \sim \bar{\nu}_{\bar{e}}}\left[\mathbf{1}\left\{\exists f \in \mathcal{F}_e \; : \; \|f(x) - f^\star(x)\| > \frac{\zeta}{2}\right\}\right], \tag{34}$$

where the sub-probability measure $\bar{\nu}_{\bar{e}}$ is defined in (32). Additionally, recall that Lemma 14-$(b)$ implies that with probability at least $1 - \delta$ any $f \in \mathcal{F}_e$ satisfies

$$\|f(x_s) - f^\star(x_s)\|_{\bar{\nu}_e} \leq \sqrt{\frac{\widehat{\Psi}_\delta^{\ell_\phi}(\mathcal{F}; T)}{\tau_e - 1}}. \tag{35}$$

Conditioning on the above event, and plugging it in (34), we get that

$$\mathbb{E}_{x_t \sim \mu}[Z_t \mathbf{1}\{\Delta_e(x_t) > \zeta\}]$$

$$\leq \mathbb{E}_{x \sim \bar{\nu}_{\bar{e}}}\left[\mathbf{1}\left\{\exists f \in \mathcal{F}_e \; : \; \|f(x) - f^\star(x)\| > \frac{\zeta}{2}, \|f(x_s) - f^\star(x_s)\|_{\bar{\nu}_e} \leq \sqrt{\frac{\widehat{\Psi}_\delta^{\ell_\phi}(\mathcal{F}; T)}{\tau_e - 1}}\right\}\right]$$

$$\leq 4 \cdot \frac{\widehat{\Psi}_\delta^{\ell_\phi}(\mathcal{F}; T)}{(\tau_e - 1)\zeta^2} \cdot \theta^{\mathrm{val}}\left(\mathcal{F}, \frac{\zeta}{2}, \frac{\widehat{\Psi}_\delta^{\ell_\phi}(\mathcal{F}; T)}{\tau_e - 1}; f^\star\right), \tag{36}$$

where the last inequality uses Lemma 15.

Summing up the bound in (36) for each term $t = 1$ to $T$, we get that

$$\sum_{t=1}^{T} \mathbb{E}_{x_t}[Z_t \mathbf{1}\{\Delta_e(x_t) > \zeta\}] = \sum_{e=1}^{E-1} \sum_{t=\tau_e}^{\tau_{e+1}-1} \mathbb{E}_{x_t}[Z_t \mathbf{1}\{\Delta_e(x_t) > \zeta\}]$$

$$\leq 4 \sum_{e=1}^{E-1} (\tau_{e+1} - \tau_e) \frac{\widehat{\Psi}_\delta^{\ell_\phi}(\mathcal{F}; T)}{(\tau_e - 1)\zeta^2} \cdot \theta^{\mathrm{val}}\left(\mathcal{F}, \frac{\zeta}{2}, \frac{\widehat{\Psi}_\delta^{\ell_\phi}(\mathcal{F}; T)}{\tau_e - 1}; f^\star\right)$$

$$\leq 8 \sum_{e=1}^{E-1} \frac{\widehat{\Psi}_\delta^{\ell_\phi}(\mathcal{F};T)}{\zeta^2} \cdot \theta^{\mathrm{val}}\left(\mathcal{F}, \frac{\zeta}{2}, \frac{\widehat{\Psi}_\delta^{\ell_\phi}(\mathcal{F};T)}{\tau_e - 1}; f^\star\right)$$

$$\leq 8 \sum_{e=1}^{E-1} \frac{\widehat{\Psi}_\delta^{\ell_\phi}(\mathcal{F};T)}{\zeta^2} \cdot \theta^{\mathrm{val}}\left(\mathcal{F}, \frac{\zeta}{2}, \frac{\widehat{\Psi}_\delta^{\ell_\phi}(\mathcal{F};T)}{T}; f^\star\right)$$

$$\leq 8 \log(T) \cdot \frac{\widehat{\Psi}_\delta^{\ell_\phi}(\mathcal{F};T)}{\zeta^2} \cdot \theta^{\mathrm{val}}\left(\mathcal{F}, \frac{\zeta}{2}, \frac{\widehat{\Psi}_\delta^{\ell_\phi}(\mathcal{F};T)}{T}; f^\star\right)$$

where the second inequality uses the fact that $\tau_{e+1} = 2\tau_e$ and that $\tau_1 = 1$, the third inequality holds due to monotonicity of $\theta^{\mathrm{val}}\left(\mathcal{F}, \frac{\zeta}{2}, \cdot; f^\star\right)$ and the last line simply plugs in the value of $E = \log(T)$.

Using Lemma 4 with the above bound for the sequence of random variable $X_t = Z_t \mathbf{1}\{\Delta_e(x_t) > \zeta\}$, we get that with probability at least $1 - \delta$,

$$\sum_{t=1}^{T} Z_t \mathbf{1}\{\Delta_e(x_t) > \zeta\} \leq \frac{3}{2} \sum_{t=1}^{T} \mathbb{E}_{x_t}\left[Z_t \mathbf{1}\{\Delta_e(x_t) > \zeta\}\right] + 4\log(2/\delta)$$

$$\leq 12 \log(T) \cdot \frac{\widehat{\Psi}_\delta^{\ell_\phi}(\mathcal{F};T)}{\zeta^2} \cdot \theta^{\mathrm{val}}\left(\mathcal{F}, \frac{\zeta}{2}, \frac{\widehat{\Psi}_\delta^{\ell_\phi}(\mathcal{F};T)}{T}; f^\star\right) + 4\log(2/\delta).$$

The final result follows by taking a union bound of the above and the event in (35). $\qquad \square$

### C.4.2. REGRET BOUND

For the ease of notation, through the proofs in this section we define the operators $\mathsf{y}^\star$ as

$$\mathsf{y}^\star(x) = \operatorname*{argmax}_k \phi(f^\star(x))[k].$$

Furthermore, recall that $\widehat{y}_t$ denotes the action chosen by the learner at round $t$ of interaction. Starting from the definition of the regret, we get that

$$\mathrm{Reg}_T = \sum_{t=1}^{T} \Pr(\widehat{y}_t \neq y_t) - \Pr(\mathsf{y}^\star(x_t) \neq y_t)$$

$$= \sum_{t=1}^{T} \mathbf{1}\{\widehat{y}_t \neq \mathsf{y}^\star(x_t)\} \cdot |\Pr(y_t = \mathsf{y}^\star(x_t)) - \Pr(y_t = \widehat{y}_t)|$$

$$= \sum_{t=1}^{T} \mathbf{1}\{\widehat{y}_t \neq \mathsf{y}^\star(x_t)\} \cdot |\phi(f^\star(x_t))[\mathsf{y}^\star(x_t)] - \phi(f^\star(x_t))[\widehat{y}_t]|$$

$$\leq \sum_{t=1}^{T} \mathbf{1}\{\widehat{y}_t \neq \mathsf{y}^\star(x_t)\} \cdot \mathrm{Gap}(f^\star(x_t), \widehat{y}_t),$$

where the last inequality plugs in the definition of Gap from (42). We can decompose the above regret bound further as:

$$\mathrm{Reg}_T \leq \sum_{t=1}^{T} \mathbf{1}\{\widehat{y}_t \neq \mathsf{y}^\star(x_t), \mathrm{Gap}(f^\star(x_t), \widehat{y}_t) \leq \varepsilon\} \cdot \mathrm{Gap}(f^\star(x_t), \widehat{y}_t)$$

$$+ \sum_{t=1}^{T} \mathbf{1}\{\widehat{y}_t \neq \mathsf{y}^\star(x_t), \mathrm{Gap}(f^\star(x_t), \widehat{y}_t) > \varepsilon\} \cdot \mathrm{Gap}(f^\star(x_t), \widehat{y}_t)$$

Using the fact that $\mathsf{y}_t(x_t) = \operatorname{argmax}_{k \in [K]} \phi(f_t(x))[k]$ in the above along with the definition of Gap and Lemma 12, we get that

$$\mathrm{Reg}_T \leq \sum_{t=1}^{T} \mathbf{1}\{\widehat{y}_t \neq \mathsf{y}^\star(x_t), \mathrm{Gap}(f^\star(x_t), \widehat{y}_t) \leq \varepsilon\} \cdot \varepsilon$$

$$+ 2\gamma \sum_{t=1}^{T} \mathbf{1}\{\widehat{y}_t \neq y^\star(x_t), \mathrm{Gap}(f^\star(x_t), \widehat{y}_t) > \varepsilon\} \cdot \|f^\star(x_t) - f_t(x_t)\|$$

$$\leq \sum_{t=1}^{T} \mathbf{1}\{\mathtt{Margin}(f^\star(x_t)) \leq \varepsilon\} \cdot \varepsilon$$

$$+ 2\gamma \sum_{t=1}^{T} \mathbf{1}\{\widehat{y}_t \neq y^\star(x_t), \mathrm{Gap}(f^\star(x_t), \widehat{y}_t) > \varepsilon\} \cdot \|f^\star(x_t) - f_t(x_t)\|$$

$$\leq \sum_{t=1}^{T} \mathbf{1}\{\mathtt{Margin}(f^\star(x_t)) \leq \varepsilon\} \cdot \varepsilon$$

$$+ 2\gamma \sum_{t=1}^{T} Z_t \mathbf{1}\{\widehat{y}_t \neq y^\star(x_t), \mathrm{Gap}(f^\star(x_t), \widehat{y}_t) > \varepsilon\} \cdot \|f^\star(x_t) - f_t(x_t)\|$$

$$+ 2\gamma \sum_{t=1}^{T} \bar{Z}_t \mathbf{1}\{\widehat{y}_t \neq y^\star(x_t)\} \cdot \|f^\star(x_t) - f_t(x_t)\|$$

$$= T_\varepsilon \cdot \varepsilon + 2\gamma \cdot \mathtt{T}_A + 2\gamma \cdot \mathtt{T}_B \cdot \|f^\star(x_t) - f_t(x_t)\|,$$

where the second inequality holds because $\mathrm{Gap}(f^\star(x_t), \widehat{y}_t) \leq \varepsilon$ implies that $\mathtt{Margin}(f^\star(x_t)) \leq \varepsilon$ whenever $\widehat{y}_t \neq y^\star(x_t)$. In the last line we plugged in the definition of $T_\varepsilon$, and defined $\mathtt{T}_A$ and $\mathtt{T}_B$ as the second term and the last term respectively. We bound term separately below:

- *Bound on* $\mathtt{T}_A$: Note that whenever $Z_t = 1$, we choose $\widehat{y}_t = \mathrm{argmax}_k \phi(f_t(x_t))[k]$. Thus,

$$\mathtt{T}_A = \sum_{t=1}^{T} Z_t \mathbf{1}\{\widehat{y}_t \neq y^\star(x_t), \mathrm{Gap}(f^\star(x_t), \widehat{y}_t) > \varepsilon\} \cdot \|f^\star(x_t) - f_t(x_t)\|$$

$$\leq \sum_{t=1}^{T} Z_t \mathbf{1}\{\|f^\star(x_t) - f_t(x_t)\| > \varepsilon/2\gamma\} \cdot \|f^\star(x_t) - f_t(x_t)\|$$

where the second line follows from Lemma 12 and because $\widehat{y}_t \neq y^\star(x_t)$. Using the fact that $\mathbf{1}\{a \geq b\} \leq a/b$ for all $a, b \geq 0$, we get that

$$\mathtt{T}_A \leq 4\gamma \sum_{t=1}^{T} Z_t \frac{\|f^\star(x_t) - f_t(x_t)\|^2}{\varepsilon}. \tag{37}$$

- *Bound on* $\mathtt{T}_B$: Fix any $t \leq T$, and let $e$ be such that $\tau_e \leq t < \tau_{e+1}$. Next, note that from Lemma 14, we have

$$\|f^\star(x_t) - g_t(x_t)\| \leq \Delta_e(x_t). \tag{38}$$

Plugging in the definition of $Z_t$, we note that

$$\mathtt{T}_B = \sum_{t=1}^{T} \mathbf{1}\{\mathtt{Margin}(g_t(x_t)) > 2\gamma\Delta_e(x_t), \widehat{y}_t \neq y^\star(x_t)\}$$

$$\leq \sum_{t=1}^{T} \mathbf{1}\{\|g_t(x_t) - f^\star(x_t)\| > \Delta_e(x_t), \mathtt{Margin}(f^\star(x_t)) > \varepsilon\},$$

where the second inequality is due Lemma 12 and by noting that $\widehat{y}_t \neq y^\star(x_t)$ and that when $Z_t = 0$, we choose $\widehat{y}_t = \mathrm{argmax}_k \phi(g_t(x_t))[k]$. However, note that the term inside the indicator contradicts (38) (which always holds). Thus,

$$\mathtt{T}_B = 0. \tag{39}$$

Combining the bounds (37) and (39), we get that:

$$\mathrm{Reg}_T \leq \varepsilon T_\varepsilon + 8\gamma^2 \sum_{t=1}^{T} Z_t \frac{\|f_t(x_t) - f^\star(x_t)\|^2}{\varepsilon}$$

$$\leq \varepsilon T_\varepsilon + \frac{8\gamma^2}{\varepsilon} \Psi_\delta^{\ell_\phi}(\mathcal{F}, T),$$

where the last inequality is due to Lemma 14.

Since $\varepsilon$ is a free parameter above, the final bound follows by choosing the best parameter $\varepsilon$, and by plugging in the form of $\Psi_\delta^{\ell_\phi}(\mathcal{F}, T)$.

### C.4.3. TOTAL NUMBER OF QUERIES

Let $N_T$ denote the total number of expert queries made by the learner within $T$ rounds of interactions. For the ease of notation, define $\Delta_t(x_t) = \Delta_{e(t)}(x_t)$ where $e(t)$ denotes the epoch number for which $\tau_{e(t)} \leq t < \tau_{e(t)+1}$. Thus,

$$
\begin{aligned}
N_T &= \sum_{t=1}^{T} Z_t \\
&= \sum_{t=1}^{T} \mathbf{1}\{\texttt{Margin}(g_t(x_t)) \leq 2\gamma\Delta_t(x_t)\} \\
&= \sum_{t=1}^{T} \mathbf{1}\{\texttt{Margin}(g_t(x_t)) \leq 2\gamma\Delta_t(x_t), \texttt{Margin}(f^\star(x_t)) \leq \varepsilon\} \\
&\qquad\qquad + \sum_{t=1}^{T} \mathbf{1}\{\texttt{Margin}(g_t(x_t)) \leq 2\gamma\Delta_t(x_t), \texttt{Margin}(f^\star(x_t)) > \varepsilon\} \\
&\leq \sum_{t=1}^{T} \mathbf{1}\{\texttt{Margin}(f^\star(x_t)) \leq \varepsilon\} \\
&\qquad\qquad + \sum_{t=1}^{T} \mathbf{1}\{\texttt{Margin}(g_t(x_t)) \leq 2\gamma\Delta_t(x_t), \texttt{Margin}(f^\star(x_t)) > \varepsilon, \Delta_t(x_t) \leq \varepsilon/4\gamma\} \\
&\qquad\qquad + \sum_{t=1}^{T} \mathbf{1}\{\texttt{Margin}(g_t(x_t)) \leq 2\gamma\Delta_t(x_t), \texttt{Margin}(f^\star(x_t)) > \varepsilon, \Delta_t(x_t) > \varepsilon/4\gamma\} \\
&= T_\varepsilon + \texttt{T}_D + \texttt{T}_E,
\end{aligned}
$$

where in the last line we used the definition of $T_\varepsilon$ and defined $\texttt{T}_D$ and $\texttt{T}_E$ respectively, which we bound separately below.

- *Bound on the term* $\texttt{T}_D$. From Lemma 14 recall that $\|f^\star(x_t) - g_t(x_t)\| \leq \Delta_e(x_t)$. Thus, for any $x_t$ for which $\|g_t(x_t)\| \leq \Delta_e(x_t)$, Lemma 13 implies that

$$\texttt{Margin}(f^\star(x_t)) \leq 2\gamma\|f_t(x_t) - f^\star(x_t)\| + \texttt{Margin}(g_t(x_t)) \leq 2\gamma\Delta_t(x_t) + \texttt{Margin}(f_t(x_t)).$$

The above implies that

$$
\begin{aligned}
\texttt{T}_D &= \sum_{t=1}^{T} \mathbf{1}\{\texttt{Margin}(g_t(x_t)) \leq 2\gamma\Delta_t(x_t), \texttt{Margin}(f^\star(x_t)) > \varepsilon, \Delta_t(x_t) \leq \varepsilon/4\gamma\} \\
&\leq \sum_{t=1}^{T} \mathbf{1}\{\texttt{Margin}(f^\star(x_t)) \leq 4\gamma\Delta_t(x_t), \texttt{Margin}(f^\star(x_t)) > \varepsilon, \Delta_t(x_t) \leq \varepsilon/4\gamma\} \\
&\leq 0,
\end{aligned}
$$

where the last line follows from the fact that all the conditions inside the indictor can not hold simultaneously for any $\varepsilon > 0$.

- *Bound on the term* $\texttt{T}_E$. We note that

$$
\begin{aligned}
\texttt{T}_E &= \sum_{t=1}^{T} \mathbf{1}\{\texttt{Margin}(g_t(x_t)) \leq 2\gamma\Delta_t(x_t), \texttt{Margin}(f^\star(x_t)) > \varepsilon, \Delta_t(x_t) > \varepsilon/4\gamma\} \\
&\leq \sum_{t=1}^{T} Z_t \mathbf{1}\{\Delta_t(x_t) \geq \varepsilon/4\gamma\}
\end{aligned}
$$

$$\leq \sum_{t=1}^{T} Z_t \mathbf{1}\{\Delta_{e(t)}(x_t) \geq \varepsilon/4\gamma\}.$$

Using Lemma 16 with $\zeta = \varepsilon/4\gamma$ to bound the term on the right hand side above, we get that with probability at least $1 - 2\delta$,

$$\mathtt{T}_E \leq O\left(\log(T)\gamma^2 \cdot \frac{\widehat{\Psi}_\delta^{\ell_\phi}(\mathcal{F};T)}{\varepsilon^2} \cdot \theta^{\mathrm{val}}\left(\mathcal{F}, \frac{\varepsilon}{8\gamma}, \frac{\widehat{\Psi}_\delta^{\ell_\phi}(\mathcal{F};T)}{T}; f^\star\right) + \log(2/\delta)\right).$$

Gathering the bounds above, we get that

$$N_T \leq \mathtt{T}_\varepsilon + O\left(\log(T)\gamma^2 \cdot \frac{\widehat{\Psi}_\delta^{\ell_\phi}(\mathcal{F};T)}{\varepsilon^2} \cdot \theta^{\mathrm{val}}\left(\mathcal{F}, \frac{\varepsilon}{8\gamma}, \frac{\widehat{\Psi}_\delta^{\ell_\phi}(\mathcal{F};T)}{T}; f^\star\right) + \log(2/\delta)\right).$$

Since $\varepsilon$ is a free parameter above, the final bound follows by choosing the best parameter $\varepsilon$, and by plugging in the form of $\widehat{\Psi}_\delta^{\ell_\phi}(\mathcal{F};T)$.

### C.5. Proof of Corollary 1

Note that the Tsybakov noise condition implies that there exists constants $c, \rho \geq 0$ such that:

$$\Pr_{x \sim \mu}(\mathtt{Margin}(f^\star(x_t)) \leq \varepsilon) \leq c\varepsilon^\rho.$$

Thus, using Lemma 4, we get that

$$
\begin{aligned}
T_\varepsilon &= \sum_{t=1}^{T} \mathbf{1}\{\mathtt{Margin}(f^\star(x_t)) \leq \varepsilon\} \\
&\leq \frac{3T}{2}\Pr_{x \sim \mu}(\mathtt{Margin}(f^\star(x)) \leq \varepsilon) + 4\log(2/\delta) \\
&\leq 2cT\varepsilon^\rho + 4\log(2/\delta).
\end{aligned}
$$

Using the above in the bound for Theorem 2, we get that for any $\varepsilon > 0$,

$$\mathrm{Reg}_T \lesssim cT\varepsilon^{\rho+1} + \frac{\gamma^2}{\lambda\varepsilon}\mathrm{Reg}^{\ell_\phi}(\mathcal{F};T) + \log(1/\delta),$$

Setting $\varepsilon = \left(\frac{\gamma^2}{\lambda cT}\mathrm{Reg}^{\ell_\phi}(\mathcal{F};T)\right)^{\frac{1}{\rho+2}}$ in the above implies that

$$\mathrm{Reg}_T \lesssim \left(\frac{\gamma^2}{\lambda}c^{\frac{1}{\rho+1}}\right)^{\frac{\rho+1}{\rho+2}}\left(\mathrm{Reg}^{\ell_\phi}(\mathcal{F};T)\right)^{\frac{\rho+1}{\rho+2}} \cdot (T)^{\frac{1}{\rho+2}} + \log(1/\delta).$$

Similarly, we can bound the query complexity bound for any $\varepsilon > 0$ as:

$$N_T \lesssim T\varepsilon^\rho + \frac{\gamma^2}{\lambda\varepsilon^2} \cdot \mathrm{Reg}^{\ell_\phi}(\mathcal{F};T) \cdot \theta^{\mathrm{val}}\left(\mathcal{F}, \varepsilon/8\gamma, \mathrm{Reg}^{\ell_\phi}(\mathcal{F};T)/T; f^\star\right) + \log(1/\delta).$$

Setting $\varepsilon = \left(\frac{\gamma^2}{\lambda T} \cdot \mathrm{Reg}^{\ell_\phi}(\mathcal{F};T) \cdot \theta^{\mathrm{val}}\left(\mathcal{F}, \varepsilon/8\gamma, \mathrm{Reg}^{\ell_\phi}(\mathcal{F};T)/T; f^\star\right)\right)^{\frac{1}{\rho+2}}$ in the above implies that

$$N_T \leq \left(\frac{\gamma^2}{\lambda} \cdot \mathrm{Reg}^{\ell_\phi}(\mathcal{F};T) \cdot \theta^{\mathrm{val}}\left(\mathcal{F}, \varepsilon/8\gamma, \mathrm{Reg}^{\ell_\phi}(\mathcal{F};T)/T; f^\star\right)\right)^{\frac{\rho}{\rho+2}} \cdot T^{\frac{2}{\rho+2}}.$$

## C.6. Proofs for Section 3.2

We prove below the lower bound in Theorem 3. The proof is on exactly the same lines as the lower bound in Theorem 28 of (Foster et al., 2020) with two main changes, first that we make classification instance and two the second scale $\beta$ we need in the star number for our ]q

*Proof of Theorem 3.* Given a the function class $\mathcal{F}$, assume that $m = \mathfrak{s}_{f^\star}^{\text{val}}(\mathcal{F}, \zeta, \beta_T) > 0$ (when its 0 the result is obvious). Let target function $f^\star$, $x^1, \ldots, x^m$ and $f_1, \ldots, f_m$ be the witness for star number. First, for any $i \in [m]$, we have that $|f^\star(x^i)| > \zeta$ by definition of the star number. Next note that, for each $f_i$, we have that, for any $j \neq i$,

$$|f_i(x^j)| \geq |f^\star(x^j)| - |f_i(x^j) - f^\star(x^j)| \geq \zeta - \beta_T \geq \zeta/2$$

On the other hand, from our definition of star number we have that,

$$|f_i(x^i)| \geq \zeta/2$$

Hence we are guaranteed that each $f_i$ has a margin of at least $\zeta/2$ on $x^1, \ldots, x^m$. Now consider the distribution $\mu$ over the context to be the uniform distribution over $\{x^1, \ldots, x^m\}$. Also let $P^i(y = 1|x) = \frac{1+f_i(x)}{2}$ be the conditional probability of label given context $x$. Let $D_i$ denote the joint distribution over $\mathcal{X} \times \{\pm 1\}$ given by drawing $x$'s from $\mu$ and labels from $P^i$. Let $D_0$ be given by drawing $x$'s from $\mu$ and $y$ conditioned on $x$ as $P(y = 1|x) = \frac{1+f^\star(x)}{2}$. Now let $\nu = 0$ with probability $1/2$ and with probability $1/2$, $\nu$ is drawn uniformly from $[m]$. Note that by our premise,

$$\frac{1}{2}\mathbb{E}_{D_0}[\text{Reg}_T] + \frac{1}{2}\frac{1}{m}\sum_{i=1}^m \mathbb{E}_{D_i}[\text{Reg}_T] = \mathbb{E}_\nu[\mathbb{E}_{D_\nu}[\text{Reg}_T]] \leq c\frac{\zeta T}{m}$$

$$\mathbb{E}_{D_i}[\text{Reg}_T] \geq \mathbb{E}_{D_i}\left[\sum_{t=1}^T \frac{\zeta}{2}\mathbb{E}_{x\sim\mu}\mathbb{E}_{\hat{y}\sim p_t(x)}\left[\mathbf{1}\{f_i(x)\hat{y} < 0\}\right]\right]$$

$$= \frac{T\zeta}{2}\mathbb{E}_{D_i}\left[\frac{1}{T}\sum_{t=1}^T \mathbb{E}_{x\sim\mu}P_{\hat{y}\sim p_t(x)}(\text{sign}(f_i(x)) \neq \hat{y})\right]$$

$$\geq \frac{T\zeta}{4}\mathbb{E}_{D_i}\left\|\frac{1}{T}\sum_{t=1}^T p_t - \text{sign}(f_i)\right\|_{L_1(\mu)}$$

Using this we get that

$$\frac{1}{m}\sum_{i=1}^m \mathbb{E}_{D_i}\left\|\frac{1}{T}\sum_{t=1}^T p_t - \text{sign}(f_i)\right\|_{L_1(\mu)} \leq \frac{8c}{m}$$

Hence using $c = 64$ and using Markov's inequality we have that

$$\frac{1}{m}\sum_{i=1}^m P_{D_i}\left(\left\|\frac{1}{T}\sum_{t=1}^T p_t - \text{sign}(f_i)\right\|_{L_1(\mu)} > \frac{2}{m}\right) \leq \frac{1}{16}$$

Further note that $\|\text{sign}(f_j) - \text{sign}(f_i)\|_{L_1(\mu)} > \frac{4}{m}$ and hence we can identify $\nu$ conditioned on it not being 0 with probability at least $1 - 1/16$. Hence using Fano with reference measure $D_0$ we get,

$$\log(2) \leq \frac{1}{m}\sum_{i=1}^m D_{KL}(D_0|D_i)$$

Now we are left with bounding $D_{KL}(D_0|D_i)$. To this, end we first make a simple observation that the distribution on the $x_t$'s is the same under $D_0$ and $D_i$. Hence on rounds $t$ where $Z_t = 0$ since we dont query for labels, these rounds we dont glean any new information to distinguish $D_i$ from $D_0$. In other words, we only need to consider rounds when $Z_t = 1$. Hence we have:

$$D_{KL}(D_0|D_i) = \mathbb{E}_{D_0}\left[\sum_{t=1}^T Z_t D_{KL}(P_0(y_t = 1|x_t)|P_i(y_t = 1|x_t))\right]$$

Assuming $\zeta > 1/4$ and using the bound on KL between Bernoulli variables we get,

$$D_{KL}(D_0|D_i) \leq \mathbb{E}_{D_0}\left[32N_i\zeta^2 + 8\left(\max_{j \neq i} N_j\right)\beta^2\right]$$

where $N_i = |\{t : Z_t = 1, x_t = x^i\}|$ and we used item 3 in the definition of star number for the $\zeta^2$ term and item 1 for the $\beta^2$ term. Hence we have that,

$$\begin{aligned}
\log(2) &\leq \frac{1}{m}\sum_{i=1}^m \mathbb{E}_{D_0}\left[32N_i\zeta^2 + 8\left(\max_{j \neq i} N_j\right)\beta^2\right] \\
&\leq \frac{32}{m}\mathbb{E}_{D_0}\left[\sum_{i=1}^m N_i\right]\zeta^2 + 8\mathbb{E}_{D_0}\left[\max_j N_j\right]\beta^2 \\
&\leq \frac{32}{m}\mathbb{E}_{D_0}\left[N_T\right]\zeta^2 + 8\mathbb{E}_{D_0}\left[N_T\right]\beta^2
\end{aligned}$$

Since $\beta^2 \leq \zeta^2/m$, we have that

$$\log(2) \leq \frac{40}{m}\mathbb{E}_{D_0}\left[N_T\right]\zeta^2$$

Hence we conclude that,

$$\mathbb{E}_{D_0}\left[N_T\right] \geq \frac{\log(2)m}{40\zeta^2}$$

which yields the lemma. $\qquad\square$

*Proof of Corollary 2.* Use $\mathcal{F} = \{f_0, f_1, \ldots, f_{\sqrt{T}}\}$ where $f_0(x_i) = 1/2 + \zeta$ for every $x_1, \ldots, x_{\sqrt{T}}$. Further, let $f_i(x_i) = 1/2 - \zeta$ and $f_i(x_j) = 1/2 + \zeta$ for any $j \neq i$. Note that with $f^\star = f_0$ and $f_1, \ldots, f_m$ on $x_1, \ldots, x_{\sqrt{T}}$ shows that star number for this class is $\sqrt{T}$ (and so is disagreement coeff.) Thus applying theorem 3 (using the converse) we see that if number of queries is smaller than $\sqrt{T}$, then regret bound is larger $\sqrt{T}$ $\qquad\square$

# D. Imitation Learning

**Additional notation.** A policy $\pi$ maps states $\mathcal{X}$ to actions $\mathcal{A}$. For any $h \leq H$, and random variable $\mathsf{Z}(x_h, a_h)$, we use the notation $\mathbb{E}_\pi[\mathsf{Z}(x_h, a_h)]$ to denote the expectation w.r.t. trajectories $\{x_1, a_1 \ldots, x_H, a_H\}$ sampled using the policy $\pi$.

## D.1. Imitation Learning Tools

We first recall the performance difference lemma, which is well known in the IL literature.

**Lemma 17** (Performance Difference Lemma; Kakade & Langford (2002); Ross & Bagnell (2014))**.** For any MDP $M$, and any two arbitrary stationary policies $\pi$ and $\widetilde{\pi}$, we have

$$V^\pi - V^{\widetilde{\pi}} = \sum_{h=1}^H \mathbb{E}_{x_h, a_h \sim d_h^{\widetilde{\pi}}}\left[-A_h^\pi(x_h, a_h)\right],$$

where $A^\pi$ is the advantage function of the policy $\pi$ in MDP $M$, i.e., $A_h^\pi(x, a) = Q_h^\pi(x, a) - V_h^\pi(x)$.

## D.2. Proof of Proposition 1

**MDP construction.** The underlying MDP is a binary tree of depth $H$. In particular, we construct the deterministic MDP $M = (\mathcal{X}, \mathcal{A}, P, r, x_1)$ where state space $\mathcal{X} = \cup_{h=1}^H \mathcal{X}_h$ with $\mathcal{X}_h = \{x_{h,i}\}_{i=1}^{2^{h-1}}$ (we assume that $x_1 = x_{1,1}$), action space $\mathcal{A} = \{0, 1\}$, reward $r$ is such that $r(x, a) = \text{Bern}\left(\frac{1}{2} + \frac{1}{4}\mathbf{1}\{x = x^\star\}\right)$ for some special state $x^\star \in \mathcal{X}_H$. The transition dynamics $P$ is deterministic and defines a binary tree over $\mathcal{X}$, i.e. for any $h$ and $x_{h,i}$, $P(x' \mid x_{h,i}, a) = 1$ if $x' = x_{h+1,2i-1}$ and $a = 0$, or $x' = x_{h+1,2i+1}$ and $a = 1$, else $P(x' \mid x, a) = 0$.

We next define the expert policy $\pi^\star$, expert model $f^\star$ and the class $\mathcal{F}$. First, for any path $\tau = (x_1, a_1, \ldots, x_H, a_H)$ from the root state $x_1$ to a terminal state $x_H$ at the layer $H$, define the policy $\pi_\tau$ as

$$\pi_\tau(x_h) = \begin{cases} a_h & \text{if} \quad (x_h, a_h) \in \tau \\ \bar{a}_h \leftarrow \text{Uniform}(\{0,1\}) & \text{otherwise} \end{cases}.$$

In particular, $\pi_\tau$ is defined such that for any state on the path $\tau$, we choose the corresponding action in $\tau$, and for any state outside of $\tau$, we choose an arbitrary (deterministic) action. Let $\mathcal{T}$ denote the set of all $2^H$ many paths from the root note $x_1$ to a leaf node $x_H \in \mathcal{X}_H$. We define the class $\Pi = \{\pi_\tau \mid \tau \in \mathcal{T}\}$, and $\mathcal{F} = \{f_\tau \mid \tau \in \mathcal{T}\}$, where for any $\tau$, we define $f_\tau : \mathcal{X} \mapsto \mathbb{R}^2$ as

$$f_\tau(x) = \begin{cases} (3/4, 1/4) & \text{if} \quad \pi_\tau(x) = 0 \\ (1/4, 3/4) & \text{if} \quad \pi_\tau(x) = 1 \end{cases}.$$

Next, let $\tau^\star = (x_1, a_1', x_2', \ldots, x_{H-1}', a_{H-1}', x^\star)$ be the path from the root $x_1$ to the special state $x^\star \in \mathcal{X}_H$ on the underlying binary tree. We finally define $\pi^\star = \pi_{\tau^\star}$ and $f^\star = f_{\tau^\star}$.

**Lower bound.** Given the MDP construction, the class $\mathcal{F}$, $f^\star$ and $\pi^\star$ above, we now proceed to the desired lower bound for non-interactive imitation learning. First, note that $\pi^\star(x) = \arg\max_a(f^\star(x)[a])$ for any $x \in \mathcal{X}$. Furthermore, $\text{Margin}(f^\star((x)) = \frac{1}{2}|f(x)[0] - f(x)[1]| = \frac{1}{4}$ for all $x \in \mathcal{X}$. Thus, for any $\varepsilon \leq \frac{1}{4}$, $T_{\varepsilon,h} = 0$.

Next, for any policy $\pi$, note that $V^\pi = \frac{1}{2} + \frac{1}{4}\mathbf{1}\{\pi = \pi^\star\}$. Thus, $\pi^\star$ is the unique $1/8$-suboptimal policy. Additionally, consider a noisy expert that draws its label according to (2) with the link function $\phi(z) = z$, i.e. on the state $x$, the expert draws its label from $a \sim f^\star(x)$. Now, suppose that the learner is given a dataset $\mathcal{D}$ of $m$ many trajectories drawn this noisy expert. There are two scenarios: either $\mathcal{D}$ does not contain $\tau^*$, or $\mathcal{D}$ contains the trajectory $\tau^*$.

- In the first case, the learner is restricted to finding $\pi^\star$ by eliminating all other $\pi \neq \pi^\star$ using the observations $\mathcal{D}$. Since, $|\Pi| = 2^H$ and each policy in the class is associated with a different path on the tree, we must have that $m = O(2^H)$.

- In the second case, we need $\tau^\star \in \mathcal{D}$. However, note that probability of observing the trajectory $\tau^\star$ when following the actions proposed by the noisy expert is $\Pr(\tau^\star \mid a_h \sim f^\star(x_h)) = (3/4)^H$. Thus, in order to observe $\tau^\star$ with probability at least $3/4$ in the dataset $\mathcal{D}$, we need $m = O((4/3)^H)$.

In both the scenarios above, we need to collect exponentially many samples.

### D.3. Proof of Theorem 4

Before delving into the proof, we recall the relevant notation. In Algorithm 2, for any $h \leq H$,

- The label $y_{t,h} \sim \phi(f^\star(x_{t,h}))$, , where $\phi$ denotes the link-function given in (2).

- The function $\text{SelectAction}(f_{t,h}(x_{t,h})) = \arg\max_k \phi(f_{t,h}(x_{t,h}))[k]$.

- For any vector $v \in \mathbb{R}^K$, the margin is given by the gap between the value at the largest and the second largest coordinate (under the link function $\phi$), i.e.

$$\text{Margin}(v) = \phi(v)[k^\star] - \max_{k \neq k^\star} \phi(v)[k],$$

where $k^\star \in \arg\max_{k \in [K]} \phi(v)[k]$.

- We define $T_\varepsilon = \sum_{t=1}^{T} \sum_{h=1}^{H} \mathbf{1}\{\text{Margin}(f_h^\star(x_{t,h})) \leq \varepsilon\}$ to denote the number of samples within $T$ rounds of interaction for which the margin w.r.t. $f_h^\star$ is smaller than $\varepsilon$.

- The trajectory at round $t$ is generated using the dynamics $\{\mathbb{T}_{t,h}\}_{h \leq H}$ to determine the states that the learner observes, starting from the state $x_1$.

- At round $t$, the learner collects data using the policy $\pi_t$ such that at time $h$, and state $x$, the action $\pi_t(x) = \texttt{SelectAction}(f_{t,h}(x_{t,h}))$.

- For any policy $\pi$, let $\tau_t^\pi$ denote the (counterfactual) trajectory that one would obtain by running $\pi$ on the deterministic dynamics $\{\mathbb{T}_{t,h}\}_{h \le H}$ with the start state $x_{t,1}$, i.e.

$$\tau_t^\pi = \left\{ x_{t,1}^\pi, \pi(x_{t,1}^\pi), \dots, x_{t,H}^\pi, \pi(x_{t,H}^\pi) \right\} \tag{40}$$

where $x_{t,1}^\pi = x_{t,1}$ and $x_{t,h+1}^\pi = \mathbb{T}_{t,h}(x_{t,h}^\pi, \pi(x_{t,h}^\pi))$.

- For a trajectory $\tau = \{x_1, a_1, \dots, x_H, a_H\}$, we define the total return

$$R(\tau) = \sum_{h=1}^{H} r(x_h, a_h). \tag{41}$$

- Additionally, for any policy $\pi$ and dynamics $\{\mathbb{T}_h\}_{h \le H}$, we define the trajectory obtained by running the policy $\pi$ as

$$\tau^\pi = \{x_1^\pi, \pi_1(x_1^\pi), x_2^\pi, \dots\}.$$

- We define the function $\mathrm{Gap} : \mathbb{R}^K \times [K] \mapsto \mathbb{R}^+$ as

$$\mathrm{Gap}(v, k) = \max_{k'} \phi(v)[k'] - \phi(v)[k], \tag{42}$$

to denote the gap between the largest and the $k$-th coordinate of $v$, and note that $\texttt{Margin}(v) \le \mathrm{Gap}(v, k)$ for all $k \ne k^\star$ (due to Lemma 9).

### D.3.1. Supporting Technical Results

We first define a useful technical lemma which allows us to bound the gap between the total returns for policies $\pi_1$ and $\pi_2$, under the dynamics $\{\mathbb{T}_h\}_{h \le H}$. Recall that for a policy $\pi$, we define the trajectory $\tau^\pi$ under $\{T_h\}_{h \le H}$ and the start state $x_1$ as the trajectory $\{x_1^\pi, \pi(x_1^\pi), \dots, x_H^\pi, \pi(x_H^\pi)\}$ where $x_1^\pi = x_1$, and $x_{h+1}^\pi \leftarrow \mathbb{T}_h(x_h^\pi, \pi(x_h^\pi))$.

**Lemma 18.** Let $\{\mathbb{T}_h\}_{h \le H}$ be a deterministic dynamics, and let $x_1$ be the start state. Let $\pi_1$ and $\pi_2$ be any two deterministic policies, and let $\tau^{\pi_1} = \{x_1^{\pi_1}, \pi_1(x_1^{\pi_1}), x_2^{\pi_1}, \dots\}$ and $\tau^{\pi_2} = \{x_1^{\pi_2}, \pi_1(x_1^{\pi_2}), x_2^{\pi_2}, \dots\}$ be two trajectories drawn using $\pi_1$ and $\pi_2$ on $\{\mathbb{T}_h\}_{h \le H}$ with start state $x_1$. Then, for any set $\mathtt{X} \subseteq \mathcal{X}$, the total trajectory rewards satisfy

$$R(\tau^{\pi_1}) - R(\tau^{\pi_2}) \le 2H \sum_{h=1}^{H} \mathbf{1}\{x_h^{\pi_1} \in \mathtt{X}\} + 2H \sum_{h=1}^{H} \mathbf{1}\{\pi_2(x_h^{\pi_2}) \ne \pi_1(x_h^{\pi_2}), x_h^{\pi_2} \notin \mathtt{X}\}.$$

*Proof.* Let $\mathfrak{h} \le H$ denote the first timestep at which the policies $\pi_1$ and $\pi_2$ choose different actions under $\{\mathbb{T}_h\}_{h \le H}$. Since the trajectories $\tau^{\pi_1} = \{x_1^{\pi_1}, \pi_1(x_1^{\pi_1}), x_2^{\pi_1}, \dots\}$ and $\tau^{\pi_2} = \{x_1^{\pi_2}, \pi_1(x_1^{\pi_2}), x_2^{\pi_2}, \dots\}$ are obtained by evolving through (the deterministic dynamics) $\{\mathbb{T}_h\}_{h \le H}$ using policies $\pi_1$ and $\pi_2$ respectively, and with the same state state $x_1$, we have that

$$x_h^{\pi_1} = x_h^{\pi_2} \qquad \text{for all } h \le \mathfrak{h},$$

and

$$\pi_1(x_h^{\pi_1}) = \pi_2(x_h^{\pi_2}) \qquad \text{for all } h \le \mathfrak{h} - 1. \tag{43}$$

Starting from the definition of the cumulative reward $R(\cdot)$, we have that

$$
\begin{aligned}
R(\tau^{\pi_1}) - R(\tau^{\pi_2}) &= \sum_{h=1}^{H} \left( r(x_h^{\pi_1}, \pi_1(x_h^{\pi_1})) - r(x_h^{\pi_2}, \pi_2(x_h^{\pi_2})) \right) \\
&= \sum_{h=1}^{\mathfrak{h}-1} \left( r(x_h^{\pi_1}, \pi_1(x_h^{\pi_1})) - r(x_h^{\pi_2}, \pi_2(x_h^{\pi_2})) \right) + \sum_{h=\mathfrak{h}}^{H} \left( r(x_h^{\pi_1}, \pi_1(x_h^{\pi_1})) - r(x_h^{\pi_2}, \pi_2(x_h^{\pi_2})) \right)
\end{aligned}
$$

$$= \sum_{h=\mathfrak{h}}^{H} \left( r(x_h^{\pi_1}, \pi_1(x_h^{\pi_1})) - r(x_h^{\pi_2}, \pi_2(x_h^{\pi_2})) \right),$$

where the last line uses the fact that the trajectories (and thus the rewards) $\tau^{\pi_1}$ and $\tau^{\pi_2}$ are identical for the first $\mathfrak{h} - 1$ states and actions (see (43)). Since (43) also implies that $x_\mathfrak{h}^{\pi_1} = x_\mathfrak{h}^{\pi_2}$, for the ease of notation we define $x_\mathfrak{h} = x_\mathfrak{h}^{\pi_1} = x_\mathfrak{h}^{\pi_2}$. Using the fact that $|r(x,a)| \leq 1$ and that $\pi_1(x_\mathfrak{h}) \neq \pi_2(x_\mathfrak{h})$ (by definition of $\mathfrak{h}$), we can bound the above as

$$
\begin{aligned}
R(\tau^{\pi_1}) - R(\tau^{\pi_2}) &\leq 2(H - \mathfrak{h} + 1)\mathbf{1}\{\pi_1(x_\mathfrak{h}) \neq \pi_2(x_\mathfrak{h})\} \\
&\leq 2H\mathbf{1}\{\pi_1(x_\mathfrak{h}) \neq \pi_2(x_\mathfrak{h})\} \\
&= 2H\mathbf{1}\{\pi_1(x_\mathfrak{h}) \neq \pi_2(x_\mathfrak{h}), x_\mathfrak{h} \in \mathsf{X}\} + 2H\mathbf{1}\{\pi_1(x_\mathfrak{h}) \neq \pi_2(x_\mathfrak{h}), x_\mathfrak{h} \notin \mathsf{X}\} \\
&\leq 2H\mathbf{1}\{x_\mathfrak{h} \in \mathsf{X}\} + 2H\mathbf{1}\{\pi_1(x_\mathfrak{h}) \neq \pi_2(x_\mathfrak{h}), x_\mathfrak{h} \notin \mathsf{X}\} \\
&= 2H\mathbf{1}\{x_\mathfrak{h}^{\pi_1} \in \mathsf{X}\} + 2H\mathbf{1}\{\pi_2(x_\mathfrak{h}^{\pi_2}) \neq \pi_1(x_\mathfrak{h}^{\pi_2}), x_\mathfrak{h}^{\pi_2} \notin \mathsf{X}\} \\
&\leq 2H\sum_{h=1}^{H}\mathbf{1}\{x_h^{\pi_1} \in \mathsf{X}\} + 2H\sum_{h=1}^{H}\mathbf{1}\{\pi_2(x_h^{\pi_2}) \neq \pi_1(x_h^{\pi_2}), x_h^{\pi_2} \notin \mathsf{X}\},
\end{aligned}
$$

where the equality in second last line plugs in the fact that $x_\mathfrak{h} = x_\mathfrak{h}^{\pi_1} = x_\mathfrak{h}^{\pi_2}$, and the last inequality is a straightforward upper bound. $\qquad\square$

We will also be using Lemma 9 and Lemma 13 from Appendix C.3 for bounding the $\mathtt{Margin}$ in the regret bound proofs. Finally, we note the following properties of the function $f_h^\star$.

**Lemma 19.** With probability at least $1 - \delta$, the function $f_h^\star$ satisfies for any $h \leq H$ and $t \leq T$,

$(a)$ $\sum_{s=1}^{t-1} Z_{s,h} \|f_h^\star(x_{s,h}) - f_{s,h}(x_{s,h})\|^2 \leq \Psi_\delta^{\ell_\phi}(\mathcal{F}_h, T),$

$(b)$ $\|f_h^\star(x_{t,h}) - f_{t,h}(x_{t,h})\| \leq \Delta_{t,h}(x_{t,h}),$

where $\Psi_\delta^{\ell_\phi}(\mathcal{F}_h, T) = \frac{4}{\lambda}\mathrm{Reg}^{\ell_\phi}(\mathcal{F}_h; T) + \frac{112}{\lambda^2}\log(4H\log^2(T)/\delta).$

*Proof.* $(a)$ We first note that we do not query oracle when $Z_{s,h} = 0$, and thus we can ignore the time steps for which $Z_{s,h} = 0$. Hence, for each $h \in [H]$, applying Lemma 5 along with the fact that $\sup_{x, f \in \mathcal{F}_h} |f(x)| \leq 1$ yields

$$\sum_{s=1}^{t-1} Z_{s,h}\|f_h^\star(x_{s,h}) - f_{s,h}(x_{s,h})\|^2 \leq \frac{4}{\lambda}\mathrm{Reg}^{\ell_\phi}(\mathcal{F}_h; T) + \frac{112}{\lambda^2}\log(4\log^2(T)/\delta)$$

for all $t \leq T$. Then, we take the union bound for all $h \in [H]$, which completes the proof.

$(b)$ The second part follows from using the observation in part-(a) that $f_h^\star$ satisfies the constraint in the definition of $\Delta_{t,h}$ given in (6), and thus $\|f_h^\star(x_{t,h}) - f_{t,h}(x_{t,h})\| \leq \Delta_{t,h}(x_{t,h})$.

$\qquad\square$

The next technical lemma bounds the number of times when $\Delta_{t,h}(x_{t,h}) \geq \zeta$ and we query the expert. Note that Lemma 20 holds even if the sequence $\{x_{t,h}\}_{t \leq T}$ was adversarially generated.

**Lemma 20.** Let $f^\star$ satisfy Lemma 19, and let $\Delta_{t,h}(x_t)$ be defined in (6). Suppose we run Algorithm 2 on data sequence $\{\{x_{t,h}\}_{h \leq H}\}_{t \leq T}$, and let $Z_{t,h}$ be as defined in line 4. Then, for any $\zeta > 0$, with probability at least $1 - H\delta$, for any $h \leq H$,

$$\sum_{t=1}^{T} Z_{t,h}\mathbf{1}\{\Delta_{t,h}(x_{t,h}) \geq \zeta\} \leq \frac{20\Psi_\delta^{\ell_\phi}(\mathcal{F}_h, T)}{\zeta^2} \cdot \mathfrak{E}(\mathcal{F}_h, \frac{\zeta}{2}; f_h^\star),$$

where $\mathfrak{E}$ denotes the eluder dimension is given in Definition 1.

*Proof.* The proof is identical to the proof of Lemma 11 by replacing all $|\cdot|$ with $\|\cdot\|$, and substitute the corresponding bounds for $f_h^\star$ via Lemma 19 (instead of using Lemma 10). We skip the proof for conciseness. $\qquad\square$

### D.3.2. REGRET BOUND

Recall that the trajectory at round $t$ is generated using the dynamics $\{\mathbb{T}_{t,h}\}_{h \le H}$. Define the policy $\pi_t$ and $\pi^\star$ such that for any $h \le H$ and $x \in \mathcal{X}_h$,

$$\pi_t(x_h) = \texttt{SelectAction}(f_{t,h}(x_h)), \quad \text{and,} \quad \pi^\star(x_h) = \texttt{SelectAction}(f_h^\star(x_h)). \tag{44}$$

Furthermore, for any policy $\pi$, let $\tau_t^\pi$ denote the trajectory that one would obtain by running $\pi$ on the deterministic dynamics $\{\mathbb{T}_{t,h}\}_{h \le H}$ with the start state $x_{t,1}$, i.e.

$$\tau_t^\pi = \left\{ x_{t,1}^\pi, \pi(x_{t,1}^\pi), \ldots, x_{t,H}^\pi, \pi(x_{t,H}^\pi) \right\} \tag{45}$$

where $x_{t,1}^\pi = x_{t,1}$ and $x_{t,h+1}^\pi = \mathbb{T}_{t,h}(x_{t,h}^\pi, \pi(x_{t,h}^\pi))$. Note that Algorithm 2 collects trajectories using the policy $\pi_t$ at round $t$. Thus, we have that

$$x_{t,h}^{\pi_t} = x_{t,h}, \tag{46}$$

where $x_{t,h}$ denotes the state at time step $h$ in round $t$ of Algorithm 2. We now have all the notation to proceed to the proof on our regret bound.

**Step 1: Bounding the difference in return at round $t$.** Fix any $t \le T$, and let $\tau_t^{\pi_t}$ and $\tau_t^{\pi^\star}$ denote the trajectories that would have been sampled using the policies $\pi_t$ and the policy $\pi^\star$ at round $t$. Furthermore, define the set $\mathsf{X}_\varepsilon$ as

$$\mathsf{X}_\varepsilon := \bigcup_{h=1}^{H} \{x \in \mathcal{X}_h \mid \texttt{Margin}(f_h^\star(x)) \le \varepsilon\} \tag{47}$$

Using Lemma 18 for the policies $\pi_t$ and $\pi^\star$, and the set $\mathsf{X}_\varepsilon$ defined above, we get that[1]

$$R(\tau_t^{\pi^\star}) - R(\tau_t^{\pi_t}) \le 2H \sum_{h=1}^{H} \mathbf{1}\{x_{t,h}^{\pi^\star} \in \mathsf{X}_\varepsilon\} + 2H \sum_{h=1}^{H} \mathbf{1}\{\pi_t(x_{t,h}^{\pi_t}) \ne \pi^\star(x_{t,h}^{\pi_t}), x_{t,h}^{\pi_t} \notin \mathsf{X}_\varepsilon\} \tag{48}$$

$$= 2H \sum_{h=1}^{H} \mathbf{1}\{x_{t,h}^{\pi^\star} \in \mathsf{X}_\varepsilon\} + 2H \sum_{h=1}^{H} \mathbf{1}\{\pi_t(x_{t,h}) \ne \pi^\star(x_{t,h}), x_{t,h} \notin \mathsf{X}_\varepsilon\}$$

$$= 2H \sum_{h=1}^{H} \mathbf{1}\{x_{t,h}^{\pi^\star} \in \mathsf{X}_\varepsilon\} + 2H \sum_{h=1}^{H} Z_{t,h} \mathbf{1}\{\pi_t(x_{t,h}) \ne \pi^\star(x_{t,h}), x_{t,h} \notin \mathsf{X}_\varepsilon\}$$

$$\qquad\qquad\qquad + 2H \sum_{h=1}^{H} \bar{Z}_{t,h} \mathbf{1}\{\pi_t(x_{t,h}) \ne \pi^\star(x_{t,h}), x_{t,h} \notin \mathsf{X}_\varepsilon\}$$

$$\le 2H \sum_{h=1}^{H} \mathbf{1}\{x_{t,h}^{\pi^\star} \in \mathsf{X}_\varepsilon\} + 2H \sum_{h=1}^{H} Z_{t,h} \mathbf{1}\{\pi_t(x_{t,h}) \ne \pi^\star(x_{t,h}), x_{t,h} \notin \mathsf{X}_\varepsilon\}$$

$$\qquad\qquad\qquad + 2H \sum_{h=1}^{H} \bar{Z}_{t,h} \mathbf{1}\{\pi_t(x_{t,h}) \ne \pi^\star(x_{t,h})\}$$

$$= 2H \sum_{h=1}^{H} \mathbf{1}\{x_{t,h}^{\pi^\star} \in \mathsf{X}_\varepsilon\} + 2H \mathsf{T}_A + 2H \mathsf{T}_B, \tag{49}$$

where the second line is obtained by plugging in (46) and the last line simply defines $\mathsf{T}_A$ and $\mathsf{T}_B$ to be the second and the third terms in the previous line without the $2H$ multiplicative factor.

We bound $\mathsf{T}_A$ and $\mathsf{T}_B$ separately below.

- *Bound on term $\mathsf{T}_A$.* Using the definition of $\mathsf{X}_\varepsilon$ from (47), we note that

$$\mathsf{T}_A = \sum_{h=1}^{H} Z_{t,h} \mathbf{1}\{\pi_t(x_{t,h}) \ne \pi^\star(x_{t,h}), x_{t,h} \notin \mathsf{X}_\varepsilon\}$$

---

[1]The key advantage of using Lemma 18 is that the first term $\sum_{h=1}^{H} \mathbf{1}\{x_{t,h}^{\pi^\star} \in \mathsf{X}_\varepsilon\}$ accounts for the number steps at which a counterfactual trajectory sampled using $\pi^\star$ goes to the state space with margin less than $\varepsilon$. Thus, we only pay for the number of times when the comparator policy $\pi^\star$ would go to states with $\varepsilon$-margin (instead of when $\pi_t$ does to such states).

$$= \sum_{h=1}^{H} Z_{t,h} \mathbf{1}\{\pi_t(x_{t,h}) \neq \pi^\star(x_{t,h}), \mathtt{Margin}(f_h^\star(x_{t,h})) > \varepsilon\}$$

$$\leq \sum_{h=1}^{H} Z_{t,h} \mathbf{1}\{\pi^\star(x_{t,h}) \neq \pi_t(x_{t,h}), \phi(f_h^\star(x_{t,h}))[\pi^\star(x_{t,h})] - \phi(f_h^\star(x_{t,h}))[\pi_t(x_{t,h})] \geq \varepsilon\}$$

$$\leq \sum_{h=1}^{H} Z_{t,h} \mathbf{1}\{\phi(f_h^\star(x_{t,h}))[\pi^\star(x_{t,h})] - \phi(f_h^\star(x_{t,h}))[\pi_t(x_{t,h})] \geq \varepsilon\},$$

where in the second last line we used the definition of $\mathtt{Margin}(f_h^\star(x_{t,h}))$ along with the fact that $\pi^\star(x_{t,h}) \neq \pi_t(x_{t,h})$. Using the relation in Lemma 12 for the term inside the indicator, we can further bound the above as

$$\mathtt{T}_A \leq \sum_{h=1}^{H} Z_{t,h} \mathbf{1}\{2\gamma \|f_h^\star(x_{t,h}) - f_{t,h}(x_{t,h})\| \geq \varepsilon\}$$

$$\leq \frac{4\gamma^2}{\varepsilon^2} \sum_{h=1}^{H} Z_{t,h} \|f_h^\star(x_{t,h}) - f_{t,h}(x_{t,h})\|^2,$$

where in the second inequality we used: $\mathbf{1}\{a \geq b\} \leq a^2/b^2$ for any $a, b \geq 0$.

- *Bound on term $\mathtt{T}_B$.* Before delving into the proof, first note that Lemma 19-$(b)$ implies that

$$\|f_h^\star(x_{t,h}) - f_{t,h}(x_{t,h})\| \leq \Delta_{t,h}(x_{t,h}). \tag{50}$$

Next, note that

$$\mathtt{T}_B = \sum_{h=1}^{H} \bar{Z}_{t,h} \mathbf{1}\{\pi_t(x_{t,h}) \neq \pi^\star(x_{t,h})\}$$

$$= \sum_{h=1}^{H} \mathbf{1}\{\mathtt{Margin}(f_{t,h}(x_{t,h})) > 2\gamma \Delta_{t,h}(x_{t,h}), \pi_t(x_{t,h}) \neq \pi^\star(x_{t,h})\},$$

where in the last line we just plugged in the query condition under which $Z_{t,h} = 0$. However note that the above two conditions inside the indicator imply that

$$2\gamma \Delta_{t,h}(x_{t,h}) < \mathtt{Margin}(f_{t,h}(x_{t,h}))$$
$$\leq \phi(f_{t,h}(x_{t,h}))[\pi_t(x_{t,h})] - \phi(f_{t,h}(x_{t,h}))[\pi^\star(x_{t,h})]$$
$$\leq 2\gamma \|f_{t,h}(x_{t,h}) - f_h^\star(x_{t,h})\|,$$

where the second line uses the definition of $\mathtt{Margin}(\cdot)$ and the fact that $\pi_t(x_{t,h}) \neq \pi^\star(x_{t,h})$, the last line is due to Lemma 12 . Thus,

$$\mathtt{T}_B \leq \sum_{h=1}^{H} \mathbf{1}\{\|f_{t,h}(x_{t,h}) - f_h^\star(x_{t,h})\| > \Delta_{t,h}(x_{t,h})\},$$

but the conditions inside the indicator in the above contradicts (50) (which holds with probability $1 - \delta$). Thus, with probability at least $1 - \delta$,

$$\mathtt{T}_B = 0. \tag{51}$$

Plugging in the bounds on $\mathtt{T}_A$ and $\mathtt{T}_B$ in (49), we get that with probability at least $1 - \delta$,

$$R(\tau_t^{\pi^\star}) - R(\tau_t^{\pi_t}) \leq 2H \sum_{h=1}^{H} \mathbf{1}\{x_{t,h}^{\pi^\star} \in \mathtt{X}_\varepsilon\} + \frac{8H\gamma^2}{\varepsilon^2} \sum_{h=1}^{H} Z_{t,h} \|f_h^\star(x_{t,h}) - f_{t,h}(x_{t,h})\|^2. \tag{52}$$

**Step 2: Bound on total regret.** Using the bound in (52) for each round $t$, we get that

$$\mathrm{Reg}_T = \sum_{t=1}^{T} \left( R(\tau_t^{\pi^\star}) - R(\tau_t^{\pi_t}) \right)$$

$$\leq 2H \sum_{h=1}^{H} \sum_{t=1}^{T} \mathbf{1}\{x_{t,h}^{\pi^\star} \in \mathsf{X}_\varepsilon\} + \frac{8H\gamma^2}{\varepsilon^2} \sum_{h=1}^{H} \sum_{t=1}^{T} Z_{t,h} \|f_h^\star(x_{t,h}) - f_{t,h}(x_{t,h})\|^2$$

$$\leq 2H \sum_{h=1}^{H} T_{\varepsilon,h} + \frac{8H\gamma^2}{\varepsilon^2} \sum_{h=1}^{H} \Psi_\delta^{\ell_\phi}(\mathcal{F}_h, T), \tag{53}$$

where the last line we use the definition of $T_{\varepsilon,h}$, and plug in the bound in Lemma 19. Using the form of $\Psi_\delta^{\ell_\phi}(\mathcal{F}_h, T)$ and ignoring log factors and constants, we get

$$\mathrm{Reg}_T = \widetilde{O}\left(2H \sum_{h=1}^{H} T_{\varepsilon,h} + \frac{8H\gamma^2}{\lambda\varepsilon^2} \sum_{h=1}^{H} \mathrm{Reg}^{\ell_\phi}(\mathcal{F}_h; T) + \log(1/\delta)\right).$$

Notice that $\varepsilon$ is a free parameter above so the final bound follows by taking $\inf$ over all feasible $\varepsilon$.

### D.3.3. TOTAL NUMBER OF QUERIES

Let $N_T$ denote the total number of expert queries made by the learner within $T$ rounds of interaction (with $H$ steps per round). For $t \leq T$, let $h_t$ denote the first timestep at which $Z_{t,h_t} = 1$ at round $t$. Thus, we have that

$$N_T = \sum_{t=1}^{T} \sum_{h=1}^{H} Z_{t,h} \tag{54}$$

$$\leq H \sum_{t=1}^{T} Z_{t,h_t}$$

$$= H \sum_{t=1}^{T} Z_{t,h_t} \mathbf{1}\{x_{t,h_t} \in \mathsf{X}_\varepsilon\} + H \sum_{t=1}^{T} Z_{t,h_t} \mathbf{1}\{x_{t,h_t} \notin \mathsf{X}_\varepsilon\}$$

$$\leq H \sum_{t=1}^{T} Z_{t,h_t} \mathbf{1}\{x_{t,h_t} \in \mathsf{X}_\varepsilon\} + H \sum_{t=1}^{T} \sum_{h=1}^{H} Z_{t,h} \mathbf{1}\{x_{t,h} \notin \mathsf{X}_\varepsilon\}$$

$$= H \sum_{t=1}^{T} Z_{t,h_t} \mathbf{1}\{x_{t,h_t} \in \mathsf{X}_\varepsilon\} + H \sum_{t=1}^{T} \sum_{h=1}^{H} Z_{t,h} \mathbf{1}\{x_{t,h} \notin \mathsf{X}_\varepsilon, \Delta_{t,h}(x_{t,h}) \leq \frac{\varepsilon}{4\gamma}\}$$

$$+ H \sum_{t=1}^{T} \sum_{h=1}^{H} Z_{t,h} \mathbf{1}\{x_{t,h} \notin \mathsf{X}_\varepsilon, \Delta_{t,h}(x_{t,h}) > \frac{\varepsilon}{4\gamma}\}$$

$$= \mathtt{T}_C + H\mathtt{T}_D + H\mathtt{T}_E,$$

where $\mathtt{T}_C$, $\mathtt{T}_D$ and $\mathtt{T}_E$ are the first, second and the third term respectively in the previous line. We bound them separately below:

- *Bound on $\mathtt{T}_C$.* Fix any $t \leq T$, and note that

$$(\mathtt{T}_C)_t = HZ_{t,h_t} \mathbf{1}\{x_{t,h_t} \in \mathsf{X}_\varepsilon\}$$
$$= HZ_{t,h_t} \mathbf{1}\{x_{t,h_t} \in \mathsf{X}_\varepsilon\} \mathbf{1}\{\forall h < h_t : \pi^\star(x_{t,h}) = \pi_t(x_{t,h})\}$$
$$+ HZ_{t,h_t} \mathbf{1}\{x_{t,h_t} \in \mathsf{X}_\varepsilon\} \mathbf{1}\{\exists h < h_t : \pi^\star(x_{t,h}) \neq \pi_t(x_{t,h})\}. \tag{55}$$

For the second term, note that

$$Z_{t,h_t} \mathbf{1}\{x_{t,h_t} \in \mathsf{X}_\varepsilon\} \mathbf{1}\{\exists h < h_t : \pi^\star(x_{t,h}) \neq \pi_t(x_{t,h})\} \leq \sum_{h=1}^{h_t} Z_{t,h_t} \mathbf{1}\{\pi^\star(x_{t,h}) \neq \pi_t(x_{t,h})\}$$

$$\leq \sum_{h=1}^{h_t} Z_{t,h_t} \bar{Z}_{t,h} \mathbf{1}\{\pi^\star(x_{t,h}) \neq \pi_t(x_{t,h})\}$$

$$\leq \sum_{h=1}^{h_t} \bar{Z}_{t,h} \mathbf{1}\{\pi^\star(x_{t,h}) \neq \pi_t(x_{t,h})\}$$

where in second inequality above, we used the fact that $Z_{t,h} = 0$ (and thus $\bar{Z}_{t,h} = 1$) for all $h \leq h_t$, by the definition of $h_t$. However note that the right hand side in the last inequality is equivalent to the term $\mathtt{T}_B$ defined above (where sum is now till $h_t$ instead of $H$). Thus, using the bound in (51) in the above, we immediately get that

$$Z_{t,h_t}\mathbf{1}\{x_{t,h_t} \in \mathtt{X}_\varepsilon\}\mathbf{1}\{\exists h < h_t : \pi^\star(x_{t,h}) \neq \pi_t(x_{t,h})\} = 0.$$

For the first term in (55), using the condition that $\pi^\star(x_{t,h}) = \pi_t(x_{t,h})$ for all $h \leq h_t$, we get that $x_{t,h} = x_{t,h}^{\pi^\star}$ and thus

$$HZ_{t,h_t}\mathbf{1}\{x_{t,h_t} \in \mathtt{X}_\varepsilon\}\mathbf{1}\{\forall h \leq h_t : \pi^\star(x_{t,h}) = \pi_t(x_{t,h})\} \leq HZ_{t,h_t}\mathbf{1}\{x_{t,h_t}^{\pi^\star} \in \mathtt{X}_\varepsilon\}$$
$$\leq H\mathbf{1}\{x_{t,h_t}^{\pi^\star} \in \mathtt{X}_\varepsilon\}$$
$$\leq H\sum_{h=1}^{H}\mathbf{1}\{x_{t,h}^{\pi^\star} \in \mathtt{X}_\varepsilon\}.$$

Gathering the two terms above, and plugging in the definition of $T_{\varepsilon,h}$, we get that

$$\mathtt{T}_C \leq H\sum_{h=1}^{H}\sum_{t=1}^{T}\mathbf{1}\{x_{t,h}^{\pi^\star} \in \mathtt{X}_\varepsilon\} = H\sum_{h=1}^{H}T_{\varepsilon,h}.$$

- *Bound on* $\mathtt{T}_D$. Using the definition of the set $\mathtt{X}_\varepsilon$ and $Z_{t,h}$, we note that

$$(\mathtt{T}_D)_t = \sum_{h=1}^{H}Z_{t,h}\mathbf{1}\{x_{t,h} \notin \mathtt{X}_\varepsilon, \Delta_{t,h}(x_{t,h}) \leq \frac{\varepsilon}{4\gamma}\}$$
$$= \sum_{h=1}^{H}\mathbf{1}\{\mathtt{Margin}(f_{t,h}(x_{t,h})) \leq 2\gamma\Delta_{t,h}(x_{t,h}), \mathtt{Margin}(f_h^\star(x_{t,h})) > \varepsilon, \Delta_{t,h}(x_{t,h}) \leq \frac{\varepsilon}{4\gamma}\} \qquad (56)$$

Recall that Lemma 19 implies that with probability at least $1 - \delta$,

$$\|f_h^\star(x_{t,h}) - f_{t,h}(x_{t,h})\| \leq \Delta_{t,h}(x_{t,h}),$$

using which with Lemma 13 implies that

$$\mathtt{Margin}(f_h^\star(x_{t,h})) \leq \mathtt{Margin}(f_{t,h}(x_{t,h})) + 2\gamma\|f_h^\star(x_{t,h}) - f_{t,h}(x_{t,h})\|$$
$$\leq \mathtt{Margin}(f_{t,h}(x_{t,h})) + 2\gamma\Delta_{t,h}(x_{t,h}).$$

Using the above bound with the conditions in (56) implies that

$$(\mathtt{T}_D)_t = \sum_{h=1}^{H}\mathbf{1}\{\mathtt{Margin}(f_h^\star(x_{t,h})) \leq 4\gamma\Delta_{t,h}(x_{t,h}), \mathtt{Margin}(f_h^\star(x_{t,h})) > \varepsilon, \Delta_{t,h}(x_{t,h}) \leq \frac{\varepsilon}{4\gamma}\}$$
$$= \sum_{h=1}^{H}\mathbf{1}\{\mathtt{Margin}(f_h^\star(x_{t,h})) \leq \varepsilon, \mathtt{Margin}(f_h^\star(x_{t,h})) > \varepsilon\}$$
$$= 0,$$

where the last equality holds because the two conditions in the indicator in the previous line can never occur simultaneously.

- *Bound on* $\mathtt{T}_E$.

$$\mathtt{T}_E = \sum_{t=1}^{T}\sum_{h=1}^{H}Z_{t,h}\mathbf{1}\{x_{t,h} \notin \mathtt{X}_\varepsilon, \Delta_{t,h}(x_{t,h}) > \frac{\varepsilon}{4\gamma}\}$$
$$\leq \sum_{h=1}^{H}\sum_{t=1}^{T}Z_{t,h}\mathbf{1}\{\Delta_{t,h}(x_{t,h}) > \frac{\varepsilon}{4\gamma}\}.$$

An application of Lemma 20 in the above for each $h \leq H$ implies that

$$\mathtt{T}_E \leq \sum_{h=1}^{H}\frac{320\gamma^2\Psi_\delta^{\ell_\phi}(\mathcal{F}_h, T)}{\varepsilon^2} \cdot \mathfrak{E}(\mathcal{F}_h, \frac{\varepsilon}{8\gamma}; f_h^\star).$$

Gathering the bound above, we get that

$$N_T \le H \sum_{h=1}^{H} T_{\varepsilon,h} + \frac{320 H \gamma^2}{\varepsilon^2} \sum_{h=1}^{H} \Psi_\delta^{\ell_\phi}(\mathcal{F}_h, T) \cdot \mathfrak{E}\left(\mathcal{F}_h, \frac{\varepsilon}{8\gamma}; f_h^\star\right).$$

Plugging in the form of $\Psi_\delta^{\ell_\phi}(\mathcal{F}_h, T)$ and ignoring $\log$ factors and constants, we get that

$$N_T \le \widetilde{O}\left( H \sum_{h=1}^{H} T_{\varepsilon,h} + \frac{320 H \gamma^2}{\lambda \varepsilon^2} \sum_{h=1}^{H} \mathrm{Reg}^{\ell_\phi}(\mathcal{F}_h; T) \cdot \mathfrak{E}(\mathcal{F}_h, \varepsilon/8\gamma; f_h^\star) + \log(1/\delta) \right).$$

Notice that $\varepsilon$ is a free parameter above so the final bound follows by taking $\inf$ over all feasible $\varepsilon$. s

## D.4. Proof for the Stochastic Setting

Algorithm 2 considers arbitrary deterministic dynamics $\{\{\mathbb{T}_{t,h}\}_{h \le H}\}_{t \le T}$. When the underlying dynamics $\widetilde{T}$ is stochastic, we can simply simulate Algorithm 2, where we set $\{\mathbb{T}_{t,h}\}_{h \le H} = \widetilde{T}(\cdot; \omega_t)$ where $\omega_t$ is drawn i.i.d. for every $t \le T$. In the following, we provide regret and query complexity bounds for stochastic dynamics.

**Regret bound.** Note that Theorem 4 bounds the difference of cumulative rewards of trajectories drawn using the policies $\pi_t$ and $\pi^\star$ on the adversarially chosen deterministic dynamics $\{\mathbb{T}_{t,h}\}_{h=1}^{H}$ respectively. In particular, we bound

$$\mathrm{Reg}_T = \sum_{t=1}^{T}\left( R(\tau_t^{\pi^\star}) - R(\tau_t^{\pi_t}) \right).$$

On the other hand, when the dynamics is stochastic, we aim to bound the gap between the expected values $V^{\pi^\star} - V^{\pi_t}$ obtained under the stochastic dynamics $\widetilde{T}$. We obtain this bound by pushing all the stochasticity into the choice of random seed $\omega$. Fix any $t \le T$, and consider the deterministic dynamics $(\mathbb{T}_{t,1}, \ldots, \mathbb{T}_{t,H})$ obtained by setting the random seed to be $\omega_t$ in the stochastic dynamics $\widetilde{T}$, i.e. $(\mathbb{T}_{t,1}, \ldots, \mathbb{T}_{t,H}) \coloneqq \widetilde{T}(\ ; \omega_t)$. Thus, for any policy $\pi$

$$V^\pi = \mathbb{E}_{\omega_t}\left[ R(\tau_t^\pi) \mid (\mathbb{T}_{t,1}, \ldots, \mathbb{T}_{t,H}) = \widetilde{T}(\ ; \omega_t) \right].$$

In the following, we will bound the difference in the value function $V^\pi - V^{\pi_t}$, by appealing to the regret bound in the proof of Theorem 4 using appropriate concentration inequalities. First, recall that in Algorithm 2, the dynamics $\{\mathbb{T}_{t,h}\}_{h \le H}$ is chosen before the round $t$, and that the policy $\pi_t$ only depends on the interaction till round $t-1$. Thus,

$$\begin{aligned}
\sum_{t=1}^{T} V^\pi - V^{\pi_t} &= \sum_{t=1}^{T} \mathbb{E}_{\omega_t}\left[ R(\tau_t^\pi) - R(\tau_t^{\pi^\star}) \right] \\
&\le \sum_{t=1}^{T} \mathbb{E}_{\omega_t}\left[ 2H \sum_{h=1}^{H} \mathbf{1}\{x_{t,h}^{\pi^\star} \in \mathsf{X}_\varepsilon\} + 2H \sum_{h=1}^{H} \mathbf{1}\{\pi_t(x_{t,h}^{\pi_t}) \ne \pi^\star(x_{t,h}^{\pi_t}), x_{t,h}^{\pi_t} \notin \mathsf{X}_\varepsilon\} \right],
\end{aligned}$$

where the last holds due to Lemma 18 and the set $\mathsf{X}_\varepsilon$ is defined in (47). An application of Lemma 4 in the above implies that with probability at least $1 - \delta$,

$$\sum_{t=1}^{T} V^\pi - V^{\pi_t} \le 4H \sum_{t=1}^{T} \sum_{h=1}^{H} \mathbf{1}\{x_{t,h}^{\pi^\star} \in \mathsf{X}_\varepsilon\} + 4H \sum_{t=1}^{T} \sum_{h=1}^{H} \mathbf{1}\{\pi_t(x_{t,h}^{\pi_t}) \ne \pi^\star(x_{t,h}^{\pi_t}), x_{t,h}^{\pi_t} \notin \mathsf{X}_\varepsilon\} + 32H^2 \log(2/\delta).$$

The rest of the proof is identical to the proof of Theorem 4 from (48) onwards. They query complexity can be similarly computed.

## D.5. Proof of Theorem 5

---

**Algorithm 4** Imitation learning with M experts, $\mathcal{A} = \{1, 2, \ldots, K\}$

---

**Require:** Parameters $\delta, \gamma, \lambda, T$, function class $\{\mathcal{F}_h^m\}_{h \leq H, m \leq M}$, online oracle $\{\text{Oracle}_h^m\}_{h \leq H, m \leq M}$ w.r.t. $\ell_\phi$.

1: Set $\Psi_\delta^{\ell_\phi}(\mathcal{F}_h^m, T) = \frac{4}{\lambda}\text{Reg}^{\ell_\phi}(\mathcal{F}_h^m; T) + \frac{112}{\lambda^2}\log(4MH\log^2(T)/\delta)$.

2: Compute $f_{1,h}^m = \text{Oracle}_{1,h}(\varnothing)$ for each $h \in [H]$ and $m \in [M]$.

3: **for** $t = 1$ to $T$ **do**

4:     Nature chooses the state $x_{t,1}$.

5:     **for** $h = 1$ to $H$ **do**

6:         Define $F_{t,h}^m(x) := [f_{t,h}^1(x), \ldots, f_{t,h}^M(x)]$.

7:         Learner plays $\widehat{y}_{t,h} = \text{SelectAction}(F_{t,h}(x_{t,h}))$.

8:         Learner transitions to the next state in this round $x_{t,h+1} \leftarrow \mathbb{T}_{t,h}(x_{t,h}, \widehat{y}_{t,h})$.

9:         For each $m \in [M]$, learner computes

$$\Delta_{t,h}^m(x_{t,h}) := \max_{f \in \mathcal{F}_h^m} \|f(x_{t,h}) - f_{t,h}^m(x_{t,h})\|$$

$$\text{s.t.} \quad \sum_{s=1}^{t-1} Z_{s,h}\|f(x_{s,h}) - f_{s,h}^m(x_{s,h})\|^2 \leq \Psi_\delta^{\ell_\phi}(\mathcal{F}_h^m, T). \tag{57}$$

        and defines $\vec{\Delta}_{t,h}(x_{t,h}) = [\Delta_{t,h}^1(x_{t,h}), \ldots, \Delta_{t,h}^M(x_{t,h})]$.

10:         Learner decides whether to query: $Z_{t,h} = \text{MarginQuery}(F_{t,h}(x_{t,h}), \vec{\Delta}_{t,h}(x_{t,h}))$

11:         **if** $Z_{t,h} = 1$ **then**

12:           **for** $m = 1$ to $M$ **do**

13:             Learner queries expert $m$ for its label $y_{t,h}^m$ for $x_{t,h}$.

14:             $f_{t+1,h}^m \leftarrow \text{Oracle}_{t+1,h}^m(\{x_{t,h}, y_{t,h}\})$

15:           **end for**

16:         **else**

17:           $f_{t+1,h}^m \leftarrow f_{t,h}^m$ for each $m \in [M]$.

18:         **end if**

19:     **end for**

20: **end for**

---

Before delving into the proof, we recall the relevant notation. In Algorithm 4, for any round $t \leq T$ and $h \leq H$:

- The aggregation function $\mathscr{A} : \mathbb{R}^{K \times M} \mapsto \mathbb{R}^K$ maps the predictions of twhhe estimated experts to distributions over actions.

- The function $\text{SelectAction}()$ chooses the action to play at round $t$, and is defined as:

$$\text{SelectAction}(F_{t,h}(x_{t,h})) = \arg\max_k \mathscr{A}(\phi(F_{t,h}(x_{t,h})))[k], \tag{58}$$

    where $F_{t,h}(x_{t,h})) = [f_{t,h}^1(x_{t,h})), \ldots, f_{t,h}^M(x_{t,h}))]$, and $\phi$ denotes the link-function given in (2).

- Our goal in Algorithm 4 is to complete with the policy $\pi^\star(x) = \text{SelectAction}(F^\star(x))$.

- On query, the $m$-th expert generates its action (which it returns in the feedback) from the distribution

$$\Pr(y_{t,h}^m = k) = \phi(f_h^{\star,m}(x_{t,h}))[k].$$

- For the ease of notation, we define the operator $\mathbb{Q}$ such that

$$\mathbb{Q}(U; \vec{\varepsilon}) := \sup_V \mathbf{1}\{\text{SelectAction}(U) \neq \text{SelectAction}(V)\}$$

$$\text{s.t.} \quad \|U[:,m] - V[:,m]\|_2 \leq \vec{\varepsilon}[m] \qquad \forall m \leq M. \tag{59}$$

- We decide whether to query the labels (from the $M$ experts) using the query function `MarginQuery()` which is defined as:

$$\texttt{MarginQuery}(F_{t,h}(x_{t,h}), \vec{\Delta}_{t,h}(x_{t,h})) = \texttt{Q}(F_{t,h}(x_{t,h}), \vec{\Delta}_{t,h}(x_{t,h})).$$

At round $t$, the learner interactions with transition dynamics $\{\mathbb{T}_{t,h}\}_{h \leq H}$ and collects data. Without loss of generality, we assume that the learner always starts from the state $x_{t,1}$. We next recall the notation on the interaction at round $t$:

- The learner collects data using the policy $\pi_t$, defined such that

$$\pi_t(x) = \texttt{SelectAction}(f_{t,h}(x_{t,h})).$$

for any $h \leq H$, and state $x \in \mathcal{X}_h$.

- For any policy $\pi$, we use the notation $\tau_t^\pi$ to denote the (counterfactual) trajectory that would have been generated by running $\pi$ on the deterministic dynamics $\{\mathbb{T}_{t,h}\}_{h \leq H}$ with the start state $x_{t,1}$, i.e.

$$\tau_t^\pi = \left\{ x_{t,1}^\pi, \pi(x_{t,1}^\pi), \ldots, x_{t,H}^\pi, \pi(x_{t,H}^\pi) \right\}, \tag{60}$$

where $x_{t,1}^\pi = x_{t,1}$ and $x_{t,h+1}^\pi = \mathbb{T}_{t,h}(x_{t,h}^\pi, \pi(x_{t,h}^\pi))$.

- For any trajectory $\tau = \{x_1, a_1, \ldots, x_H, a_H\}$, we define the total return

$$R(\tau) = \sum_{h=1}^{H} r(x_h, a_h). \tag{61}$$

We finally recall the definition of $T_{\varepsilon,h}$:

- Given a matrix-valued function $F(x) \in \mathbb{R}^{K \times M}$, we define the set $\mathbb{B}_\infty(F(x), \vec{\Delta})$ to denote the set of all matrices $F'$ such that $\|F(x)[:,m] - F'(x)[:,m]\| \leq \vec{\Delta}[m]$ for all $m \leq M$.

- We say that a point $x$ is within $\varepsilon$-margin w.r.t. $F_h^\star$ if

$$\mathbf{1}\{\texttt{Q}(F_h^\star(x), \varepsilon\vec{\mathbb{1}}) = 1\}.$$

Informally, the above implies that there exists an $F'(x) \in \mathbb{B}_\infty(F_h^\star(x), \varepsilon)$ such that $\texttt{SelectAction}(F'(x)) \neq \texttt{SelectAction}(F_h^\star(x))$.

- We define the set $T_{\varepsilon,h} = \sum_{t=1}^{T} \mathbf{1}\{\texttt{Q}(F_h^\star(x_{t,h}^{\pi^\star}), \varepsilon\vec{\mathbb{1}}) = 1\}$ to denote the number of samples within $T$ rounds of interaction, and on the corresponding (counterfactual) trajectory of $\pi^\star$, that are within $\varepsilon$-margin (as defined above).

### D.5.1. SUPPORTING TECHNICAL RESULTS

**Lemma 21.** With probability at least $1 - \delta$, for any $m \leq M$, and $t \leq T$ and $h \leq H$, the function $f^{\star,m}$ satisfies

(a) $\sum_{s=1}^{t-1} Z_{s,h} \|f_h^{\star,m}(x_{s,h}) - f_{s,h}^m(x_{s,h})\|^2 \leq \Psi_\delta^{\ell_\phi}(\mathcal{F}_h^m, T)$,

(b) $\|f_h^{\star,m}(x_{t,h}) - f_{t,h}^m(x_{t,h})\| \leq \Delta_{t,h}^m(x_{t,h})$,

where $\Psi_\delta^{\ell_\phi}(\mathcal{F}_h^m, T) = \frac{4}{\lambda}\text{Reg}^{\ell_\phi}(\mathcal{F}_h^m; T) + \frac{112}{\lambda^2}\log(4MH\log^2(T)/\delta)$.

*Proof.*

(a) We first note that we do not query oracle when $Z_{s,h} = 0$, and thus we can ignore the time steps for which $Z_{s,h} = 0$. Hence, for each $h \in [H]$ and $m \in [M]$, applying Lemma 5 yields

$$\sum_{s=1}^{t-1} Z_{s,h} \|f_h^{\star,m}(x_{s,h}) - f_{s,h}(x_{s,h})\|^2 \leq \frac{4}{\lambda}\text{Reg}^{\ell_\phi}(\mathcal{F}_h^m; T) + \frac{112}{\lambda^2}\log(4\log^2(T)/\delta)$$

for all $t \leq T$. Then, we take the union bound for all $h \in [H]$ and $m \in [M]$, which completes the proof.

($b$) The second part follows from using part-(a) along with the definition in (57).

$\square$

The next lemma bound the number of times when $\Delta_{t,h}^m(x_{t,h}) \geq \zeta$, and we query. Note that Lemma 22 holds even if the sequence $\{x_{t,h}\}_{t \leq T}$ was adversarially generated.

**Lemma 22.** Let $f^{\star,m}$ satisfy Lemma 21, and let $\Delta_{t,h}^m(x_{t,h})$ be defined in Algorithm 4. Suppose Algorithm 4 is run on the data sequence $\{x_{t,h}\}_{t \leq 1}$, and let $Z_{t,h}$ be defined in line D.5. Then, for any $\zeta > 0$, with probability at least $1 - M\delta$, for any $m \in [M]$, and $h \leq H$,

$$\sum_{t=1}^{T} Z_{t,h} \mathbf{1}\{\Delta_{t,h}^m(x_t) \geq \zeta\} \leq \frac{20 \Psi_\delta^{\ell_\phi}(\mathcal{F}_h^m, T)}{\zeta^2} \cdot \mathfrak{E}(\mathcal{F}_h^m, \zeta/2; f_h^{\star,m}),$$

where $\mathfrak{E}$ denotes the eluder dimension given in Definition 1.

*Proof.* The proof is identical to the proof of Lemma 11 where we handle each $m \in [M]$ and $h \in [H]$ separately, and substitute the corresponding bounds for $f_h^{\star,m}$ via Lemma 21 (instead of using Lemma 10). We skip the proof for conciseness.

$\square$

### D.5.2. REGRET BOUND

Suppose the trajectories at round $t$ are generated using the deterministic dynamics $\{\mathbb{T}_{t,1}, \ldots, \mathbb{T}_{t,H}\} = \widetilde{T}(\cdot\,; \omega_t)$ where $\omega_t$ denotes the random seed that captures all of the stochasticity at round $t$. Furthermore, for any policy $\pi$, let $\tau_t^\pi$ denote the trajectory that one would obtain by executing $\pi$ on $\{\mathbb{T}_{t,h}\}_{h \leq H}$ with the start state $x_{t,1}$, i.e.

$$\tau_t^\pi = \{x_{t,1}^\pi, \pi(x_{t,1}^\pi), \ldots, x_{t,H}^\pi, \pi(x_{t,H}^\pi)\} \tag{62}$$

where $x_{t,1}^\pi = x_{t,1}$ and $x_{t,h+1}^\pi = T_{t,h}(x_{t,h}^\pi, \pi(x_{t,h}^\pi))$. Define the policies $\pi^\star$ and $\pi_t$ such that for any $h \leq H$ and $x \in \mathcal{X}_h$,

$$\pi^\star(x) = \texttt{SelectAction}(F_h^\star(x)), \qquad \text{and,} \qquad \pi_t(x) = \texttt{SelectAction}(F_{t,h}(x)).$$

Note that Algorithm 4 collects trajectories using the policy $\pi_t$ at round $t$. Thus, we have that

$$x_{t,h}^{\pi_t} = x_{t,h}, \tag{63}$$

where $x_{t,h}$ denotes the state at time step $h$ in round $t$ of Algorithm 4. We now proceed to the bound on the regret.

**Step 1: Bounding the difference in cumulative return at round $t$.** Fix any $t \leq T$, and let $\tau_t^{\pi_t}$ and $\tau_t^{\pi^\star}$ denote the trajectories that would have been sampled using the policies $\pi_t$ and the policy $\pi^\star$ at round $t$. Furthermore, define the set $\mathsf{X}_\varepsilon$ as

$$\mathsf{X}_\varepsilon := \bigcup_{h=1}^{H} \{x \in \mathcal{X}_h \mid \mathbb{Q}(F_h^\star(x), \varepsilon\vec{\mathbb{1}}) = 1\} \tag{64}$$

Using Lemma 18 for the policies $\pi_t$ and $\pi^\star$, and the set $\mathsf{X}_\varepsilon$ defined above, we get that

$$
\begin{aligned}
R(\tau_t^{\pi^\star}) - R(\tau_t^{\pi_t}) &\leq 2H \sum_{h=1}^{H} \mathbf{1}\{x_{t,h}^{\pi^\star} \in \mathsf{X}_\varepsilon\} + 2H \sum_{h=1}^{H} \mathbf{1}\{\pi_t(x_{t,h}^{\pi_t}) \neq \pi^\star(x_{t,h}^{\pi_t}), x_{t,h}^{\pi_t} \notin \mathsf{X}_\varepsilon\} \\
&= 2H \sum_{h=1}^{H} \mathbf{1}\{x_{t,h}^{\pi^\star} \in \mathsf{X}_\varepsilon\} + 2H \sum_{h=1}^{H} \mathbf{1}\{\pi_t(x_{t,h}) \neq \pi^\star(x_{t,h}), x_{t,h} \notin \mathsf{X}_\varepsilon\} \\
&= 2H \sum_{h=1}^{H} \mathbf{1}\{x_{t,h}^{\pi^\star} \in \mathsf{X}_\varepsilon\} + 2H \sum_{h=1}^{H} Z_{t,h} \mathbf{1}\{\pi_t(x_{t,h}) \neq \pi^\star(x_{t,h}), x_{t,h} \notin \mathsf{X}_\varepsilon\} \\
&\qquad\qquad + 2H \sum_{h=1}^{H} \bar{Z}_{t,h} \mathbf{1}\{\pi_t(x_{t,h}) \neq \pi^\star(x_{t,h}), x_{t,h} \notin \mathsf{X}_\varepsilon\}
\end{aligned}
$$

$$= 2H \sum_{h=1}^{H} \mathbf{1}\{x_{t,h}^{\pi^\star} \in \mathtt{X}_\varepsilon\} + 2H\mathtt{T}_A + 2H\mathtt{T}_B, \tag{65}$$

where the second line is obtained by using the relation (63) in the second line. The last line simply defines $\mathtt{T}_A$ and $\mathtt{T}_B$ to be the second and the third term in the previous line, respectively, without the $2H$ multiplicative factor. We bound these two terms separately below:

- *Bound on* $\mathtt{T}_A$. Using the definition of $\mathtt{X}_\varepsilon$ from (64), we note that

$$
\begin{aligned}
\mathtt{T}_A &= \sum_{h=1}^{H} Z_{t,h} \mathbf{1}\{\pi_t(x_{t,h}) \neq \pi^\star(x_{t,h}), x_{t,h} \notin \mathtt{X}_\varepsilon\} \\
&= \sum_{h=1}^{H} Z_{t,h} \mathbf{1}\{\pi_t(x_{t,h}) \neq \pi^\star(x_{t,h}), \mathtt{Q}(F_h^\star(x_{t,h}), \varepsilon\vec{\mathbb{1}}) = 0\} \\
&= \sum_{h=1}^{H} Z_{t,h} \mathbf{1}\{\exists m \in [m] : \|f_{t,h}^m(x_{t,h}) - f_h^{\star,m}(x_{t,h})\| > \varepsilon\},
\end{aligned}
$$

where the last line follows from the fact that the definition of $\mathtt{Q}$ and the fact that $\pi_t(x_{t,h}) \neq \pi^\star(x_{t,h})$ implies that there exists some $m \in [M]$ for which $\|f_{t,h}^m(x_{t,h}) - f_h^{\star,m}(x_{t,h})\| > \varepsilon$. The above implies that

$$\mathtt{T}_A \leq \sum_{m=1}^{M} \sum_{h=1}^{H} Z_{t,h} \mathbf{1}\{\|f_{t,h}^m(x_{t,h}) - f_h^{\star,m}(x_{t,h})\| > \varepsilon\}.$$

- *Bound on* $\mathtt{T}_B$. First note that Lemma 21 implies that with probability at least $1 - \delta$, for all $m \leq M$ and $h \leq H$,

$$\|f_h^{\star,m}(x_{t,h}) - f_{t,h}^m(x_{t,h})\| \leq \Delta_{t,h}^m(x_{t,h}). \tag{66}$$

Next, note that

$$
\begin{aligned}
\mathtt{T}_B &\leq \sum_{h=1}^{H} \bar{Z}_{t,h} \mathbf{1}\{\pi_t(x_{t,h}) \neq \pi^\star(x_{t,h})\} \tag{67} \\
&= \sum_{h=1}^{H} \mathbf{1}\{\mathtt{Q}(F_{t,h}(x_{t,h}), \vec{\Delta}_{t,h}(x_{t,h})) = 0, \pi_t(x_{t,h}) \neq \pi^\star(x_{t,h})\},
\end{aligned}
$$

where in the last line follows from plugging in the query condition under which $Z_{t,h} = 0$. However note that for any $h \leq H$ for which $\mathtt{Q}(F_{t,h}(x_{t,h}), \vec{\Delta}_{t,h}(x_{t,h})) = 0$, by the definition of $\mathtt{Q}$ and the fact that $\pi_t(x_{t,h}) \neq \pi^\star(x_{t,h})$, there must exist some $m \in [M]$ such that

$$\|f_h^{\star,m}(x_{t,h}) - f_{t,h}^m(x_{t,h})\| > \Delta_{t,h}^m(x_{t,h}).$$

However, the above contradicts (66), and thus with probability at least $1 - \delta$,

$$\mathtt{T}_B = 0. \tag{68}$$

Plugging the above bounds on $\mathtt{T}_A$ and $\mathtt{T}_B$ in (65), we get that

$$R(\tau_t^{\pi^\star}) - R(\tau_t^{\pi_t}) \leq 2H \sum_{h=1}^{H} \mathbf{1}\{x_{t,h}^{\pi^\star} \in \mathtt{X}_\varepsilon\} + 2H \sum_{m=1}^{M} \sum_{h=1}^{H} Z_{t,h} \mathbf{1}\{\|f_{t,h}^m(x_{t,h}) - f_h^{\star,m}(x_{t,h})\| > \varepsilon\}. \tag{69}$$

**Step 2: Aggregating over all time steps.** Using the bound in (69) for each round $t$, we get that

$$\mathrm{Reg}_T = \sum_{t=1}^{T} \left( R(\tau_t^{\pi^\star}) - R(\tau_t^{\pi_t}) \right)$$

$$\leq 2H \sum_{h=1}^{H} \sum_{t=1}^{T} \mathbf{1}\{x_{t,h}^{\pi^\star} \in \mathsf{X}_\varepsilon\} + 2H \sum_{t=1}^{T} \sum_{m=1}^{M} \sum_{h=1}^{H} Z_{t,h} \mathbf{1}\{\|f_{t,h}^m(x_{t,h}) - f_h^{\star,m}(x_{t,h})\| > \varepsilon\}.$$

Using the fact that $\mathbf{1}\{a \geq b\} \leq a^2/b^2$ for any $a, b \geq 0$, and the definition of $T_{\varepsilon,h}$ in the above, we get that

$$\mathrm{Reg}_T \leq 2H \sum_{h=1}^{H} T_{\varepsilon,h} + 2H \sum_{t=1}^{T} \sum_{m=1}^{M} \sum_{h=1}^{H} Z_{t,h} \frac{\|f_{t,h}^m(x_{t,h}) - f_h^{\star,m}(x_{t,h})\|^2}{\varepsilon^2}$$

$$\leq 2H \sum_{h=1}^{H} T_{\varepsilon,h} + \frac{2H}{\varepsilon^2} \sum_{m=1}^{M} \sum_{h=1}^{H} \Psi_\delta^{\ell_\phi}(\mathcal{F}_h^m, T). \tag{70}$$

where the last line follows from using the bound in Lemma 21.

Plugging in the form of $\Psi_\delta^{\ell_\phi}(\mathcal{F}_h^m, T)$ and ignoring log factors and constants, we get that

$$\mathrm{Reg}_T \lesssim 2H \sum_{h=1}^{H} T_{\varepsilon,h} + \frac{2H}{\lambda\varepsilon^2} \sum_{m=1}^{M} \sum_{h=1}^{H} \mathrm{Reg}^{\ell_\phi}(\mathcal{F}_h^m; T) + \log(1/\delta).$$

Notice that $\varepsilon$ is a free parameter above so the final bound follows by taking $\inf$ over all feasible $\varepsilon$.

### D.5.3. TOTAL NUMBER OF QUERIES

Next, we bound $N_T$-the total number of queries made by the learner within $T$ rounds of interaction. We first define additional notation. Fix any $t \leq T$, and let $h_t$ denote the first time step at round $t$ for which $Z_{t,h_t} = 1$, if such a time-step exists (and is set to be $H + 1$ otherwise). We first observe that for all $h \leq h_t$, we have $\pi^\star(x_{t,h}) = \pi_t(x_{t,h})$. To see this, note that

$$Z_{t,h_t} \mathbf{1}\{\exists h < h_t : \pi^\star(x_{t,h}) \neq \pi_t(x_{t,h})\} \leq \sum_{h=1}^{h_t-1} Z_{t,h_t} \mathbf{1}\{\pi^\star(x_{t,h}) \neq \pi_t(x_{t,h})\}$$

$$\leq \sum_{h=1}^{h_t-1} Z_{t,h_t} \bar{Z}_{t,h} \mathbf{1}\{\pi^\star(x_{t,h}) \neq \pi_t(x_{t,h})\}$$

$$\leq \sum_{h=1}^{h_t-1} \bar{Z}_{t,h} \mathbf{1}\{\pi^\star(x_{t,h}) \neq \pi_t(x_{t,h})\}$$

where in second inequality above, we used the fact that $Z_{t,h} = 0$ (and thus $\bar{Z}_{t,h} = 1$) for all $h < h_t$, by the definition of $h_t$. Observe that the right hand side in the last inequality above is equivalent to the term (67) in the bound on $\mathrm{T}_B$ above (where sum is now till $h_t$ instead of $H$). Thus, using the bound in (68), we get that

$$Z_{t,h_t} \mathbf{1}\{\exists h < h_t : \pi^\star(x_{t,h}) \neq \pi_t(x_{t,h})\} = 0,$$

and thus

$$\pi^\star(x_{t,h}) = \pi_t(x_{t,h}) \qquad \text{for all } h \leq h_t. \tag{71}$$

Next, note that plugging in the definition of $h_t$, we get that

$$N_T = \sum_{t=1}^{T} \sum_{h=1}^{H} Z_{t,h}$$

$$\leq H \sum_{t=1}^{T} Z_{t,h_t}$$

$$= H \sum_{t=1}^{T} Z_{t,h_t} \mathbf{1}\{x_{t,h_t} \in \mathsf{X}_\varepsilon\} + H \sum_{t=1}^{T} Z_{t,h_t} \mathbf{1}\{x_{t,h_t} \notin \mathsf{X}_\varepsilon\}$$

$$= H \sum_{t=1}^{T} Z_{t,h_t} \mathbf{1}\{x_{t,h_t} \in \mathsf{X}_\varepsilon\} + H \sum_{t=1}^{T} Z_{t,h_t} \mathbf{1}\{x_{t,h_t} \notin \mathsf{X}_\varepsilon, \|\Delta_{t,h_t}^m(x_{t,h_t})\|_\infty \leq \frac{\varepsilon}{4}\}$$

$$+ H \sum_{t=1}^{T} Z_{t,h_t} \mathbf{1}\{x_{t,h_t} \notin \mathsf{X}_\varepsilon, \|\Delta_{t,h_t}^m(x_{t,h_t})\|_\infty > \frac{\varepsilon}{4}\}$$

$$= \mathsf{T}_C + \mathsf{T}_D + \mathsf{T}_E,$$

where $\mathsf{T}_C$, $\mathsf{T}_D$ and $\mathsf{T}_E$ are the first, second and the third term respectively in the previous line. We bound them separately below.

- *Bound on* $\mathsf{T}_C$. Fix any $t \le T$. Using the relation in (71), note that $\pi^\star(x_{t,h}) = \pi_t(x_{t,h})$ for all $h < h_t$. Thus, the corresponding trajectories would be identical till time step $h_t$, which implies that $x_{t,h_t} = x_{t,h_t}^{\pi^\star}$. Using this property in the $\mathsf{T}_C$, we get that

$$
\begin{aligned}
(\mathsf{T}_C)_t &= H \sum_{t=1}^{T} Z_{t,h_t} \mathbf{1}\{x_{t,h_t} \in \mathsf{X}_\varepsilon\} \\
&= H \sum_{t=1}^{T} Z_{t,h_t} \mathbf{1}\{x_{t,h_t}^{\pi^\star} \in \mathsf{X}_\varepsilon\} \\
&\le H \sum_{t=1}^{T} \sum_{h=1}^{H} Z_{t,h} \mathbf{1}\{x_{t,h}^{\pi^\star} \in \mathsf{X}_\varepsilon\} \\
&= H \sum_{h=1}^{H} T_{\varepsilon,h},
\end{aligned}
$$

where the last line plugs in the definition of $T_{\varepsilon,h}$.

- *Bound on* $\mathsf{T}_D$. First note that

$$(\mathsf{T}_D)_t = H\mathbf{1}\{\mathbb{Q}\big(F_{t,h_t}(x_{t,h_t}), \vec{\Delta}_{t,h_t}(x_{t,h_t})\big) = 1, \mathbb{Q}(F^\star(x_{t,h_t}), \varepsilon\vec{\mathbb{1}}) = 0, \sup_{m \in [M]} \Delta_t^m(x_{t,h_t}) \le \varepsilon/4\}.$$

In the following, we will show that all the conditions in the above indicator can not hold simultaneously. First note that since $\mathbb{Q}\big(F_{t,h_t}(x_{t,h_t}), \vec{\Delta}_{t,h_t}(x_{t,h_t})\big) = 1$, there exists an $\widetilde{F}$ such that

$$\mathtt{SelectAction}(\widetilde{F}(x_{t,h_t}))) \ne \mathtt{SelectAction}(F_{t,h_t}(x_{t,h_t})) \tag{72}$$

and

$$\forall m \in [M]: \qquad \|\widetilde{F}(x_{t,h_t})[:,m] - F_{t,h_t}(x_{t,h_t})[:,m]\| \le \Delta_{t,h_t}^m(x_{t,h_t}). \tag{73}$$

On the other hand, recall that Lemma 21 implies that

$$\forall m \in [M]: \qquad \|F^\star(x_{t,h_t})[:,m] - F_{t,h_t}(x_{t,h_t})[:,m]\| \le \Delta_{t,h_t}^m(x_{t,h_t}). \tag{74}$$

Since, $\sup_m \Delta_{t,h_t}^m(x_{t,h_t}) \le \varepsilon/4$, an application of Triangle inequality along with the bounds (73) and (74) imply that

$$\forall m \in [M]: \qquad \|F^\star(x_{t,h_t})[:,m] - \widetilde{F}(x_{t,h_t})[:,m]\| \le 2\Delta_{t,h_t}^m(x_{t,h_t}) < \varepsilon. \tag{75}$$

But the above contradicts the fact that $\mathbb{Q}(F^\star(x_{t,h_t}), \varepsilon\vec{\mathbb{1}}) = 0$ since both $\widetilde{F}$ and $F_t$ satisfy the norm constraints in the definition of $\mathbb{Q}$, but we can not simultaneously have that

$$\mathtt{SelectAction}(F^\star(x_{t,h_t}))) = \mathtt{SelectAction}(F_{t,h_t}(x_{t,h_t})) = \mathtt{SelectAction}(\widetilde{F}(x_{t,h_t})),$$

due to (72). Thus, we must have that

$$(\mathsf{T}_D)_t = 0.$$

- *Bound on* $\mathtt{T}_E$. We note that

$$\mathtt{T}_E \le H \sum_{t=1}^{T} Z_{t,h_t} \mathbf{1}\{\|\vec{\Delta}_{t,h_t}(x_{t,h_t})\|_\infty > \varepsilon/4\}$$

$$= H \sum_{t=1}^{T} Z_{t,h_t} \mathbf{1}\{\exists m \in [M] : \Delta_{t,h_t}^m(x_{t,h_t}) > \varepsilon/4\}$$

$$\le H \sum_{m=1}^{M} \sum_{t=1}^{T} Z_{t,h_t} \mathbf{1}\{\Delta_{t,h_t}^m(x_{t,h_t}) > \varepsilon/4\}$$

$$\le H \sum_{h=1}^{H} \sum_{m=1}^{M} \sum_{t=1}^{T} Z_{t,h} \mathbf{1}\{\Delta_{t,h}^m(x_{t,h}) > \varepsilon/4\},$$

where the last line simply upper bound the term for $h_t$ by the corresponding terms for all $h \le H$.

Using Lemma 22 to bound the term in the right hand side for each $m \in [M]$ and $h \le H$, we get that

$$\mathtt{T}_E \le \sum_{h=1}^{H} \sum_{m=1}^{M} \frac{320 H \Psi_\delta^{\ell_\phi}(\mathcal{F}_h^m, T)}{\varepsilon^2} \cdot \mathfrak{E}(\mathcal{F}_h^m, \frac{\varepsilon}{8}; f_h^{\star,m}).$$

Gathering the bound above, we get that

$$N_T \le H \sum_{h=1}^{H} T_{\varepsilon,h} + \frac{320 H}{\varepsilon^2} \sum_{h=1}^{H} \sum_{m=1}^{M} \Psi_\delta^{\ell_\phi}(\mathcal{F}_h^m, T) \cdot \mathfrak{E}(\mathcal{F}_h^m, \frac{\varepsilon}{8}; f_h^{\star,m}).$$

Plugging in the form of $\Psi_\delta^{\ell_\phi}(\mathcal{F}_h^m, T)$ and ignoring $\log$ factors and constants, we get that

$$N_T \lesssim H \sum_{h=1}^{H} T_{\varepsilon,h} + \frac{320 H}{\lambda \varepsilon^2} \sum_{h=1}^{H} \sum_{m=1}^{M} \mathrm{Reg}^{\ell_\phi}(\mathcal{F}_h^m; T) \cdot \mathfrak{E}(\mathcal{F}_h^m, \varepsilon/8; f_h^{\star,m}) + \log(1/\delta).$$

Notice that $\varepsilon$ is a free parameter above so the final bound follows by taking $\inf$ over all feasible $\varepsilon$.

