# OpenReview forum: "Selective Sampling and Imitation Learning via Online Regression"
_ICML.cc/2023/Workshop/ILHF — ILHF Workshop ICML 2023_

### Official Review · Reviewer_QCPM · 2023-06-09
**Novel Contribution but with Detrimental Flaws**

**Rating:** 6
**Confidence:** 5

**Review:**

## **Strong Points**

* Despite some minor language inconsistencies and typographical errors, the paper presents its content clearly and explains the proposed approach in an easily understandable manner. However, there are some areas of concern, which I address separately below.
* The approach proposed in this paper exhibits novelty, and its empirical setup, along with the theoretical and experimental results, are extensively explored and discussed. The authors might consider enhancing the content further and refining the presentation. Also, I would aim for a journal submission with this paper, given the substance of their research.

## **Weak Points, Suggestions, and Areas for Improvement**

### **Abstract**
* The current abstract is quite lengthy. The same information could be effectively communicated in fewer sentences. For instance, the second paragraph of the abstract—which arguably doesn't need to be there—could be eliminated as its points are already distributed throughout the paper.

### **Introduction**

* The first sentence of the Introduction should be revised for clarity. Its current phrasing makes the intended message unclear.
* Instead of merely stating that active learning and selective sampling have been extensively studied, it would be beneficial to provide a brief explanation of these concepts. This would serve to establish context and relevance.
* The authors mention a "lower bound indicating that dependence on the eluder dimension is unavoidable in the query complexity in general" and their "new selective sampling algorithm can operate under the most general setting where we have multiple labels (i.e., multiclass classification) with general model class." The implications and importance of these statements, however, remain unclarified. Specifically, what is the significance of the dependence on the eluder dimension? And what constitutes 'the most general setting' and 'general model classes'? Further elaboration would improve understanding.
* Lastly, the claim that "Such an assumption is realistic and has been considered in robotics applications in (Beliaev et al., 2022)" is unsubstantiated. The authors should provide the reasoning behind this statement's realism.

### **Presentation and Methodology**
* While a couple of prior studies are discussed in the Introduction section, it would be better to include a "Related Work" section at least in the Appendix.
* The first two sections seem a bit protracted and could be presented more concisely. It takes roughly 2.5 pages to transition into the methodology, and this is done without the inclusion of a 'Related Work' section. A considerable number of statements are also reiterated throughout.
* I would argue that Section 2 is superfluous. However, if you wish to summarize your contributions, it's unnecessary to delve into technical details within this section. Rather, a high-level summary of your algorithm's improvements over previous methods would suffice. You could then highlight the technical advantages once the method's description is complete. This would avoid potential reader confusion and improve the overall presentation. It could also contribute to space conservation.
* In Section 3, the first sentence makes reference to "nature," a term that is left undefined. For the sake of the general readership, please clarify this term's meaning.
* The first paragraph of the second column on page 6 contains the statement, "However, in practice, human demonstrators are far from being optimal and suggestions from experts should be modeled as noisy suggestions that only correlate with $\pi^{*}$." Is there a justification or reference to support this claim?
* The authors frame the problem of selective sampling in the Imitation Learning (IL) setting as an MDP. While this is technically sound, no justification or explanation is provided as to whether there are temporal correlations between subsequent states. If none exist, why not consider a bandit setting for this specific problem?
* Finally, I would suggest the authors include at least some of their most significant empirical results. Space limitations are understood, but this could be accommodated by removing Section 2, as previously suggested.

### **Language and Grammar**

* The language used in the paper could benefit from refinement. For example, instead of using a phrase such as "we care about" on page 2, consider adopting a more academic tone with phrases like "we only consider..." or "we are concerned with..."
* In Section 3, the last sentence of the first paragraph appears incomplete. It reads, "We assume that a link function $\phi: \mathbb{R}^{K} \mapsto \Delta(K)$ maps scores to distributions and assume that the noisy label" This sentence needs to be completed to convey the intended meaning.
* Similarly, the following sentence, "In this work, we assume that the link function $\Phi: \mathbb{R}^{K} \mapsto \mathbb{R}$ (see (Agarwal, 2013) for more details) which satisfies the following assumption:" ends with an unnecessary colon. The sentence structure should be revised for proper punctuation.

### **Minor Comments**

* In Section 2, it would be beneficial to label the paragraphs related to "Selective Sampling" and "Imitation Learning" as subsections since their content aligns with the corresponding section headers.
* For the first bullet point on page 2, it would be best to reference Algorithm 1 only after fully elucidating the method. Doing so would improve the flow and comprehension of the text.
* The use of references could use some refinement. Citations should not be made in the first person with parentheses. For instance, it's more appropriate to say "by Ross et al. (2011)" rather than "by (Ross et al., 2011)" and "Dekel et al. (2012) showed that..." instead of “In (Dekel et al., 2023)". Moreover, try to avoid nested parentheses when giving references within parentheses, such as “(e.g., (Laskey et al., 2016; Brantley et al., 2020))” in the first paragraph on page 2.
* The term 'Markov decision process' is defined multiple times, which is unnecessary and disrupts the text's flow. Additionally, it should be written as "Markov decision process" instead of "Markov Decision Process" in line with standard academic writing conventions.

---

### Official Review · Reviewer_QrkX · 2023-06-09
**A Mathematical Solid Paper**

**Rating:** 10
**Confidence:** 2

**Review:**

Disclaimer: The reviewer only scans through the equations, as this submission is 51 pages (including the appendix). But the assumptions and results seem to be reasonable at first glance.

This paper studies the imitation learning problem with access to noisy expert feedback. It proposes a novel selective sampling algorithm for general function classes and multiple actions. The authors also extended the algorithm that accommodates noisy expert data. This paper also provides extensive theoretical analysis to justify the proposed algorithm.

Given the condition that the reviewer does not fully understand the theoretical contributions, they look legit and solid (which should go into a top conference or journal in the reviewer’s opinion). Hence the reviewer would like to recommend the highest rating.

---

### Decision · Program_Chairs · 2023-06-20

Accept